# Electrically modulated photothermal force microscopy for revealing molecular conformation changes during polarization switching at the nanoscale

Songyou Yao [1,2,3], He Jiang[1,2,3], Jiaxuan Wen[1,2,3], Da Shu[1,2,3], Chang Xu[1], Wenpeng Zhu [1,2,3], Xiaoyue Zhang [1,2,3] ✉ & Yue Zheng [1,2,3] ✉

Organic ferroelectrics have attracted extensive attention because of the broad tunability of polarization via chemical and structural modifications. However, simultaneous analysis of the evolution of chemical and polarization properties at the nanoscale remains a challenge, impeding the understanding of their origin. In this work, we report electrically modulated photothermal force microscopy (ePTFM), an atomic force microscopy (AFM)-based technique that integrates nanoscale analysis of polarization with chemical specificity. By characterizing electrodriven ferroelectric switching in edge-on poly(vinylidene fluoride-co-trifluoroethylene) (P(VDF-TrFE)) lamellae, ePTFM reveals not only the evolution of polarization but also in situ chemical correlations. The results show that ePTFM has less electrostatic interference than conventional AFM techniques do, allowing intrinsic characterization of polar evolution under bias. Furthermore, via multi-wavenumber analysis, we propose a conformational mechanism for chain-direction ferroelectric switching in face-on P(VDF-TrFE). The proposed ePTFM provides fresh insight into polarization evolution and paves the way for mechanistic studies of polar organics.

Organic ferroelectrics have diverse chemical constituents and microstructures, allowing a wide range of controllability with regard to polarization behavior and thus diverse functionalities[1,2]. They have been the subject of extensive research in cutting-edge applications ranging from high-performance actuators[3,4] to multifunctional sensors[5,6], nonvolatile memories[7,8] and artificial synaptic devices[9,10]. However, the inherent complexity of the structure and chemical constituents poses challenges in fully comprehending the intricate relationship between the polarization behavior and molecular structure. This gap in understanding has hindered further exploration in evaluating and tailoring these materials for desired physical functionalities[11–13]. To bridge this gap, an integrating analysis of

nanoscale chemical properties and polar evolution is needed. This integration is crucial for understanding the fundamental mechanisms that drive polarization switching. Only when nanoscale polarization and chemical component analysis are integrated can the origins and controllability principles of organic ferroelectrics be established with certainty, other than relying on empiricism or speculation[14,15].

To understand polarization evolution, numerous experimental techniques have been proposed[16]. However, there are still constraints in employing these methods to monitor chemical components during nanoscale polarization evolution. While macroscale and nanoscale polarization analyses have been successfully realized via electrical

[1]Guangdong Provincial Key Laboratory of Magnetoelectric Physics and Devices, School of Physics, Sun Yat-sen University, Guangzhou, China. [2]State Key Laboratory of Optoelectronic Materials and Technologies, Sun Yat-sen University, Guangzhou, China. [3]Centre for Physical Mechanics and Biophysics, School of Physics, Sun Yat-sen University, Guangzhou, China. ✉e-mail: zhangxy26@mail.sysu.edu.cn; zhengy35@mail.sysu.edu.cn

techniques like Sawyer-Tower circuit[17] and piezoresponse force microscopy (PFM)[18], there remains a lack of chemical specificity and the movement or deformation of specific molecular structure during polarization reversal. Chemical analysis techniques, such as Fourier transform infrared spectroscopy (FTIR), have enabled the analysis of chemical information during polar evolution[19–21]. Nevertheless, a restriction in spatial resolution occurs due to the diffraction limit[22]. For other approaches, although transmission electron microscopy (TEM) may offer an alternative method for nanoscale chemical and polar analyses, radiation damage to organics can be a concern[23,24]. The preparation procedure for thin-film samples, which involves slicing and ion thinning, can potentially alter the original structure of the organic film[25], even when using cryo-TEM. The widely used atomic force microscopy-infrared spectroscopy (AFM-IR) has established itself as a powerful tool for nanoscale chemical imaging, utilizing a combination of mechanical and optical techniques[26–28]. Several notable attempts have been made to modify AFM-IR to expand its performance and applications[29–33]. Among these, linearly polarized AFM-IR stands out as a method used to achieve nanoscale analysis of molecular orientation[34]. However, the lack of electric modulation poses a challenge in correlating the polarization behavior with the evolution of specific chemical conformations at the nanoscale.

Here, electrically modulated photothermal force microscopy (ePTFM), an AFM-IR-based technique, is designed and proposed to achieve in situ polarization analysis with nanoscale capability and chemical specificity. With this method, the evolution of the molecular structure during ferroelectric switching in edge-on poly(vinylidene fluoride-co-trifluoroethylene) (P(VDF-TrFE)) is observed. The results reveal the capability of ePTFM to integrate chemical and polar evolution at the nanoscale and confirm its robustness under an electric bias. Furthermore, using ePTFM, the ferroelectric mechanism along the chain direction in toroidal face-on P(VDF-TrFE) was subsequently explored. We believe that the proposed method will provide fresh perspectives and encourage the discovery of emerging phenomena for organic ferroelectrics and even polar organics.

## Results

### Configuration and principles of the ePTFM system

The main configuration of the ePTFM is shown in Fig. 1a. Briefly, our technique integrates optical radiation, modulated electric bias and photothermal effect into an AFM system to achieve chemically specific characterization during polarization evolution at the nanoscale. First, nanoscale chemical characterization is realized by detecting pulsed infrared-induced photothermal expansion of the sample through an AFM tip. The amplitude of cantilever oscillation allows for the determination of infrared absorption[35]. To manipulate polarization evolution simultaneously, an electric module consisting of a signal generator and a voltage amplifier is used to enable a programmable electric bias on the sample under the AFM tip. Moreover, a tuning circuit is established to handle the probe's response and control pulse frequency through an oscillator (OSC), as shown in Fig. 1a. The circuit first divides the deflection of the probe into two equivalent parts and multiplies them with their reference signals, which are produced by the OSC with a 0° or 90° phase shift. Following lock-in amplifier, the results yield the amplitude $R$ and phase $\phi$. Phase $\phi$ is fed to a phase-lockedloop (PLL) to ensure the IR pulse frequency tracks and reaches the AFM probe-sample contact resonance frequency. The amplitude is synchronously recorded with the modulated bias via multichannel data acquisition (DAQ). Finally, ePTFM captures a switching spectrum that shows case selective IR absorption evolving with a modulated electric bias (Fig. 1b, Supplementary Fig. 1). A comparison between IR spectra acquired via ePTFM and the original AFM-IR instrument is presented in Supplementary Fig. 2, confirming that such modifications do not affect the basic functionalities of the AFM-IR instrument.

Because a linearly polarized laser source is used, the variation in infrared absorption serves as indicators of both the changes in chemical concentration and molecular orientation during electric modulation. The photothermal expansion $\mu_0$ of a sample can be given as[27]

$$\mu_0(\sigma, \theta) = G\alpha \frac{I_{\text{inc}}}{\rho C_{\text{p}}} t_P I_{\text{abs}}(\sigma, \theta), \tag{1}$$

where G contains all optical and geometrical parameters of the sample, $\alpha$ is the thermal expansion coefficient, $I_{\text{inc}}$ is the incident laser energy, $\rho$ is the sample density, $C_{\text{p}}$ is the sample heat capacity, $t_P$ is the duration of the laser pulse, $I_{\text{abs}}(\sigma, \theta)$ is the IR absorption coefficient, $\sigma$ is the wavenumber and $\theta$ is the relative angle between the direction of the IR electric field and the transition dipole moment. Since the cantilever amplitude is directly proportional to the magnitude of thermal expansion, it is also related to the angle $\theta$. In particular, if no chemical reaction occurs, the cantilever amplitude indicates the orientation of a specific polar structure. The ePTFM enables the identification of these two cases by observing variations in polarization-dependent infrared absorption at multiple wavenumbers. As shown in Fig. 1c, when a certain molecular structure is rotated to align its transition dipole moment parallel to IR electric field, the infrared absorption and thus cantilever amplitude reach the maximum value. In contrast, when the two are perpendicular, the amplitude is minimized. Therefore, for orthogonal transition dipole moments, changes in the molecular orientation produce opposite absorption variations at the corresponding wavenumbers. Alternatively, changes in chemical concentration only affect the related characteristic absorption, producing the same absorption variations for all transition dipole moments.

In comparison, PFM characterizes nanoscale polarization through the local electromechanical response. However, it cannot directly reveal the origin and mechanism of polarization evolution. The ePTFM method addresses this limitation. By selecting a specific IR wavenumber during polarization switching, the technique enables the tracking of particular molecular structure, thereby revealing the underlying chemical origins.

### Chemically specific characterization of electrodriven polarization evolution

To demonstrate the chemical specificity of ePTFM and its relevant performance, we characterized the electrodriven polarization evolution of a ferroelectric P(VDF-TrFE) thin film with an edge-on structure. Edge-on P(VDF-TrFE) has polymer chains aligned parallel to the substrate, as schematically shown in Fig. 2a[36]. The polarization reversal of edge-on P(VDF-TrFE) exhibits hysteresis characteristics of ferroelectric switching. When subjected to an out-of-plane (OOP) electric bias, the dipolar moment originating from the -CH$_2$ and -CF$_2$ groups will undergo crankshaft rotation about the polymer chain axis[37] (Fig. 2b). The resulting dipolar hysteresis serves as the underlying mechanism for ferroelectrics in edge-on P(VDF-TrFE). Using a self-assembly method assisted by poly(tetrafluoroethylene) (PTFE) template, we fabricated edge-on P(VDF-TrFE) lamellae with high ferroelectric crystallinity and uniform orientation of the polymer chains[38,39]. Ferroelectric β-phase crystallization was confirmed by the IR spectrum in Supplementary Fig. 3. Both the AFM image (Fig. 2c) and the cross-sectional SEM image (Supplementary Fig. 4) show that the morphology of the edge-on P(VDF-TrFE) lamellae is consistent with that of the reference[38].

To investigate the polarization evolution of the edge-on P(VDF-TrFE) films, we used a triangle–square electric bias for both the PFM and ePTFM measurements. The signal can be obtained during the direct current (DC) pulse or immediately after each DC pulse. These two situations are referred to as "on-field" and "off-field", respectively[40]. Figure 2d shows that both on-field and off-field PFM loops exhibit typical ferroelectric butterfly patterns, indicating the

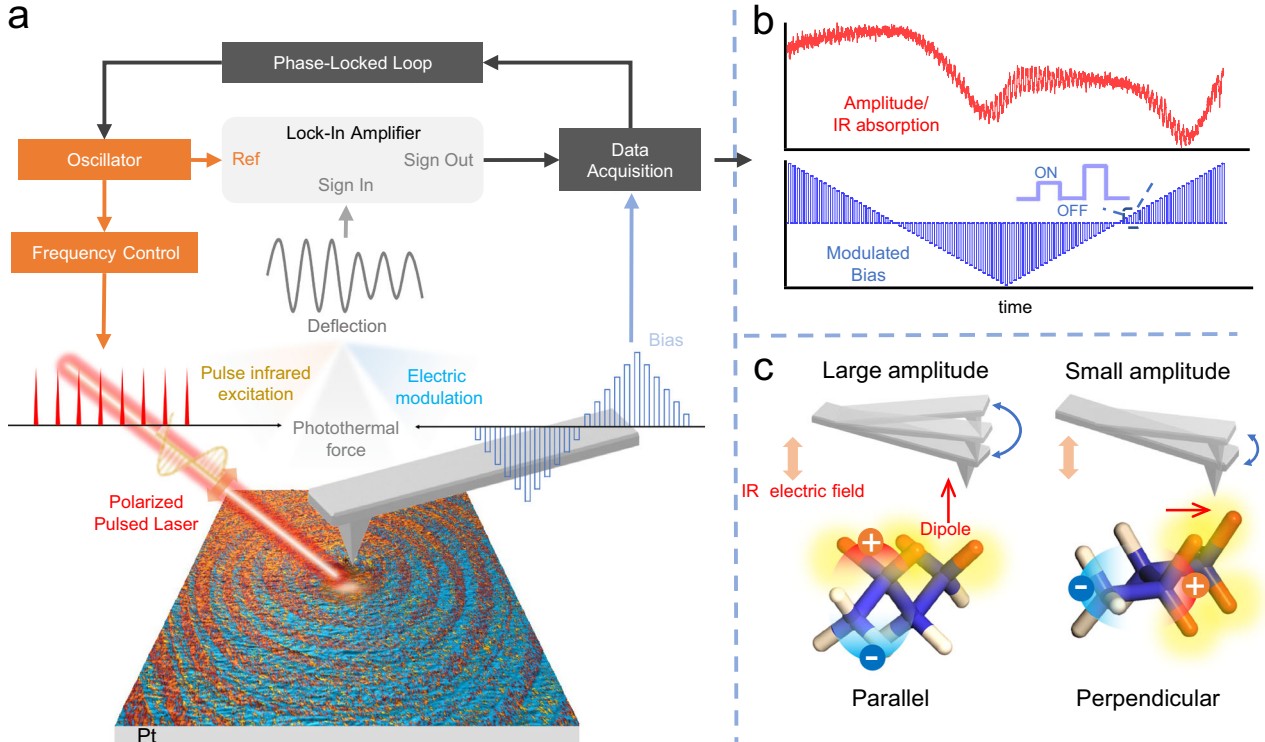

**Fig. 1 | Setup and working principle of electrically modulated photothermal force microscopy. a** Schematic of electrically modulated photothermal force microscopy. A polarized pulsed laser illuminates the sample below the AFM tip, generating photothermal expansion. A modulated bias is applied through the AFM tip for manipulation of the polarization. The cantilever deflection signal is obtained under the influence of pulsed infrared excitation, photothermal force and electric modulation. A tuning circuit is used to collect and analyze the cantilever deflection signal. **b** Time-dependent variation in the amplitude and electric bias. A more specific result is shown in Supplementary Fig. 1. IR absorption is recorded under a triangle–square bias. The amplitude trace (red), which indicates the IR absorption, varies with the modulated bias (blue), reflecting the corresponding electrodriven evolution. **c** Schematic showing the correlation between the IR absorption and rotation of the molecular structure under a vertically polarized IR beam. The red and blue regions represent the polarizations of the entire species. The yellow glow indicates the molecule that is selectively observed. The specific molecular dipolar moment can be tracked by selecting the IR wavenumber. The amplitude reaches the maximum or minimum value if the selective transition dipolar moment is parallel or perpendicular to the IR electric field, respectively.

ferroelectric reversal of polarization. Then, we monitored the IR absorption at 1289 cm$^{-1}$ to explore the conformation changes during ferroelectric switching. Typical results are shown in Fig. 2e. Both the on-field and off-field 1289 cm$^{-1}$ IR-E loops of edge-on P(VDF-TrFE) lamellae exhibit butterfly patterns of ferroelectrics. It's noteworthy that the IR-E loops not only reflect ferroelectric switching but also simultaneously contain chemical correlations that are absent in the PFM loops. This means that by observing the alteration in IR absorption, we can monitor the behavior of specific molecular conformation during ferroelectric switching. We also conducted a negative control experiment using a non-ferroelectric polymer, specifically PMMA (poly(methyl methacrylate)). The IR-E loop results reveal a flat evolutionary trend without the characteristic butterfly shape (Supplementary Fig. 5).

The IR absorption peak at 1289 cm$^{-1}$ is attributed to the -CF$_2$ symmetric stretching vibration ($v_s$ (CF$_2$)) of the β phase crystal[41]. The corresponding transition dipole moment of it is almost parallel to the macroscopic polarization direction of the edge-on P(VDF-TrFE) lamellae. Therefore, Fig. 2e indicates the direct correspondence between the -CF$_2$ rotation and ferroelectric switching. To elucidate the correlation, three critical stages of ferroelectric switching are marked in the IR-E loop (Fig. 2b, e, f). At stage (I), the OOP orientation of -CF$_2$ contributes to OOP remnant polarization, as indicated by the relatively larger absorption intensity. As the electric bias increases to the coercive voltage at stage (II), the -CF$_2$ is forced to align nearly perpendicular to the IR electric field, resulting in minimal OOP polarization and IR absorption (Fig. 2d, e). After the switching is completed at stage (III),

-CF$_2$ turns over and realigns parallel to the IR electric field, thereby restoring the cantilever amplitude and OOP polarization. Owing to the ferroelectric hysteresis, the flip-back rotation occurs at a negative coercive voltage, resulting in a characteristic butterfly pattern. These results are consistent with previous research on dipolar switching dynamics in P(VDF-TrFE)[20].

Moreover, we can observe correlations among different chemical components by selecting various IR absorption peaks in Supplementary Fig. 3. Specifically, the IR absorption peak located at 1183 cm$^{-1}$ is chosen, which corresponds to the asymmetric stretching vibration of -CF$_2$ ($v_{as}$ (CF$_2$))[41]. Its transition dipole moment aligns orthogonally to that at 1289 cm$^{-1}$ and is perpendicular to the polymer chains. Thus, while $v_s$(CF$_2$) exhibits strong absorption, $v_{as}$(CF$_2$) should show weak absorption, resulting in the opposite trend. The IR-E loops at 1183 cm$^{-1}$ shown in Fig. 2f are consistent with the above scenario, which exhibits an upside-down butterfly pattern. On the other hand, the transition dipole moment at 1076 cm$^{-1}$ aligns parallel to the polymer chain of edge-on P(VDF-TrFE) lamellae[41], which exhibits minimal variation during OOP ferroelectric switching. As expected, no butterfly pattern emerges in the 1076 cm$^{-1}$ IR-E loops (Supplementary Fig. 6). Moreover, Supplementary Fig. 7 illustrates additional IR-E loops that consistently support our findings. To ensure accuracy, we have carefully excluded artifacts resulting from the shift of the absorption peak under an electric bias, as evidenced in Supplementary Fig. 8. Furthermore, we also conducted on-field spectrum characterizations (Supplementary Fig. 9), which demonstrate consistency of the IR-E loops results.

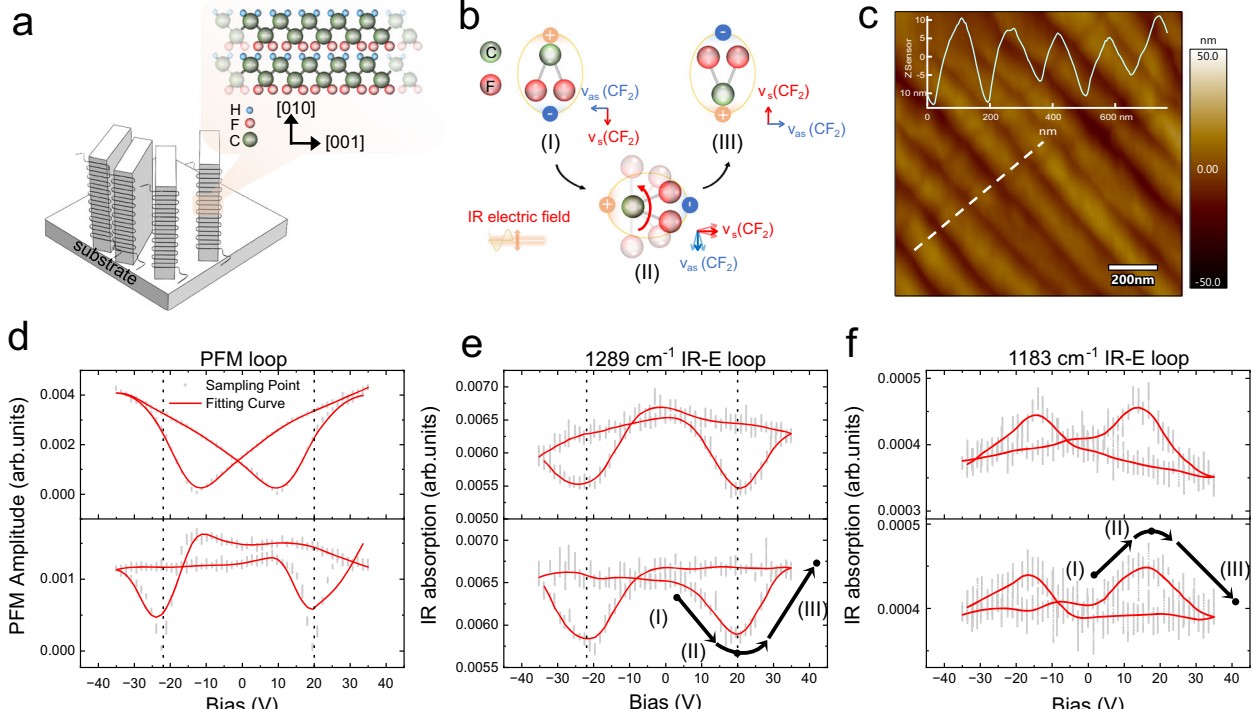

**Fig. 2 | Electrodriven evolution of -CF₂ in ferroelectric edge-on P(VDF-TrFE) lamellae. a** Schematic diagram showing the microstructure of the edge-on lamellae. The polymer chain orientation is mainly parallel to the substrate. **b** Schematic of -CF₂ molecular rotation during ferroelectric polarization switching. The red arrow indicates the transition dipole of the -CF₂ symmetric stretching vibration (vₛ (CF₂)). The blue arrow indicates the transition dipole of the -CF₂ asymmetric stretching vibration (vₛ (CF₂)). The mechanism of ferroelectricity in edge-on P(VDF-TrFE) originates from the rotation of -CF₂ about the chain axis. The IR electric field

is in the vertical direction. **c** AFM morphology of the fabricated edge-on P(VDF-TrFE) lamellae. The inset is the cross-sectional profile at the white dashed line. **d** PFM hysteresis loops in the on-field and off-field stages. The gray dots correspond to the distribution of 64 sampling data points of cantilever amplitude at each on/off-field stage. The red solid line corresponds to the fitting curve according to the average value of the sampling points. **e** IR-E loops at 1289 cm⁻¹ with on- and off-fields. Stages (I) - (III) correspond to (**b**). **f** IR-E loop of 1183 cm⁻¹ with on- and off-field.

When comparing the on-field and off-field loops, two important differences emerge between ePTFM and PFM. As shown in Fig. 2d, the on-field PFM loop clearly exhibits a lower coercive voltage, and an unsaturated amplitude compared to off-field loop. Conversely, both the on-field and off-field IR-E loops at 1289 cm⁻¹ in Fig. 2e maintain consistent coercive voltages and saturable amplitudes. The differences between the on-field and off-field PFM amplitude loops are the results of the DC-induced phenomena. In fact, the PFM signal contains additional contributions, including electrostatic response, charge injection and Vegard strain, giving a misleading picture of the ferroelectricity of the sample[42,43]. However, the utilization of pulsed infrared radiation can eliminate the electrostatic interference induced by DC electric bias. These interferences do not exhibit the same infrared absorption characteristic frequency as the observed polar bonds and groups. Thus, they should not affect the IR absorption. A comprehensive discussion on the mechanisms and bias-induced phenomena in PFM and ePTFM is provided in Supplementary Note 1.

**Trans-to-gauche conformational transition in face-on P(VDF-TrFE) with toroidal domains**

Toroidal polar topology has never been observed in ferroelectric polymers until its discovery in the biaxially strained P(VDF–TrFE) lamellae with face-on structure (Fig. 3a). These developments have led to emerging functionalities[44,45]. Although polarization extinction has been reported in face-on P(VDF-TrFE)[46], unexpected ferroelectricity along the chain direction was still observed and confirmed in face-on P(VDF-TrFE) lamellae with toroidal domains[44]. For brevity, it is referred to as toroidal P(VDF-TrFE). The mechanism of its ferroelectric polarization reversal remains an unsettled issue of current investigation,

which may inspire new design principles for materials. To understand the electrodriven mechanism of ferroelectric switching, toroidal P(VDF-TrFE) lamellae were fabricated, and IR-E analysis was conducted via ePTFM.

The toroidal domain and chain-direction ferroelectricity are confirmed by in-plane PFM (IP-PFM), as shown in Fig. 3b, and OOP domain switching is successfully conducted in toroidal P(VDF-TrFE) films (Supplementary Fig. 10). The face-on structure was further confirmed with the IR spectrum (Fig. 3c), where the appearance of a peak at 1400 cm⁻¹ under a vertically polarized IR beam represents the vertical orientation of the polymer chain[47,48]. Through a comparison of the IR spectra of edge-on and face-on P(VDF-TrFE) in Fig. 3c, we revealed that 1120 cm⁻¹ is significantly distinguished from that of edge-on P(VDF-TrFE). It is mentioned in the literature that the peak at 1120 cm⁻¹ is related to the T₃GT₃G′ (TTTG) conformations[20,49]. Here, trans(T) and gauche(G) feature torsional bond angles[50]. T represents the arrangement with substituents at 180° to each other, and G/G′ represents that at ±60°. Correspondingly, the absorption bands characteristic of TTTT at 1289 cm⁻¹ and 1183 cm⁻¹ decrease, indicating an increased amount of TTTG in face-on P(VDF-TrFE) with toroidal domains. To further verify the structural conformation, we performed FTIR for both edge-on and toroidal P(VDF-TrFE). As shown in Supplementary Fig. 11, several peaks assigned to TTTG are observed[49,51,52], including those at 833 cm⁻¹, 800 cm⁻¹ and 772 cm⁻¹. The results also show a double band at 614 cm⁻¹ and 605 cm⁻¹ in face-on lamellae, which indicates a greater content of the TG conformation[52]. The appearance of more gauche conformations is understandable, as relaxor behavior has been confirmed in toroidal P(VDF-TrFE)[44]. One of the reasons for the appearance of relaxor properties in polymers is conformational disorder[53] or

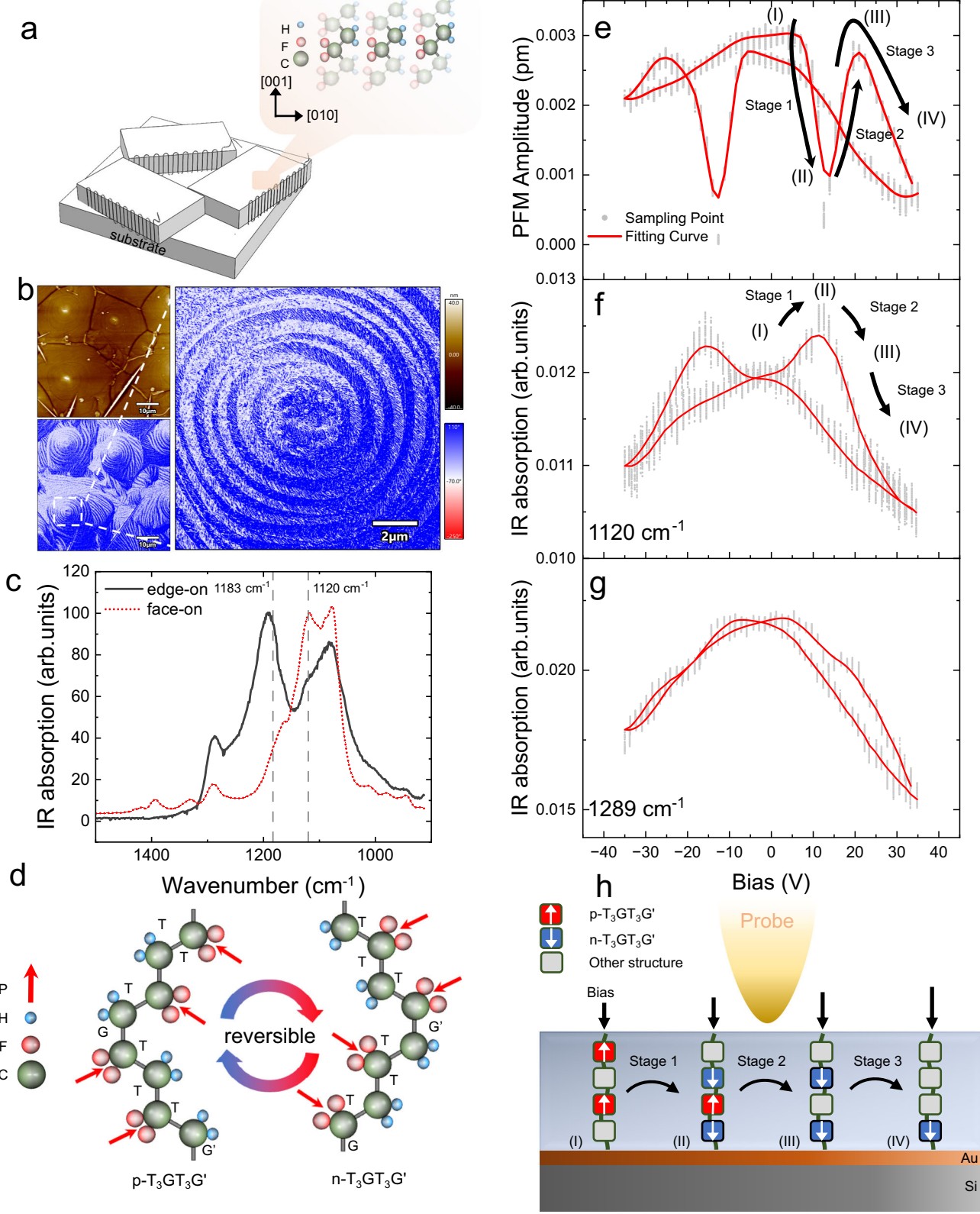

defects[54]. As the introduction of gauche conformation will cause chain distortion[50], both TTTG and TGTG can be considered to be disordered conformations compared to TTTT.

The direction of the chain-parallel dipole of $T_3GT_3G'$ can be determined by where the gauche defects occur at the polymer chains. As shown in Fig. 3d, $T_3GT_3G'$ with a gauche defect at -$CH_2$-$CF_2$- results in upwards polarization and it is termed p- $T_3GT_3G'$ (positive). $T_3GT_3G'$

with a gauche defect at -$CF_2$-$CH_2$- results in the opposite outcome and it is termed n-$T_3GT_3G'$ (negative). Biaxial strain in face-on P(VDF-TrFE) flattens the energy landscape of polarization states and is crucial for the development of toroidal polar topology[44]. Analogously, we believe that such strain also reduces the energy barrier between T and G, promoting a trans-to-gauche transformation under an electric bias. On the basis of these observations, we postulated that OOP polarization

**Fig. 3 | Characterizations and mechanism of polarization reversal in toroidal face-on P(VDF-TrFE) lamellae. a** Schematic diagram showing the microstructure of face-on lamellae. The chain orientation is perpendicular to the substrate. **b** AFM morphology and IP-PFM images of the lamellae showing the topology of the toroidal domain. The upper color box applies to the upper-left image, while the lower color box applies to the other two images. The white dashed square denotes a detailed IP-PFM phase image of a zoomed-in region. **c** Normalized IR spectrum of edge-on and face-on P(VDF-TrFE) lamellae showing the appearance of the γ phase. **d** The schematic of negative and positive $T_3GT_3G'$ states, which are reversible depending on the external modulation. **e** OOP-PFM amplitude loop showing the reversible OOP plane polarization of lamellae. The gray dots correspond to the distribution of 64 sampling data points of cantilever amplitude at each off-field stage. The red solid line corresponds to the fitting curve according to the average value of the sampling points. **f** Off-field IR-E loop at 1120 cm$^{-1}$ with an obvious butterfly pattern. **g** Off-field IR-E loop at 1289 cm$^{-1}$. The on-field IR-E loop results are shown in Supplementary Fig. 12. **h** Schematic of unconventional structural evolution under a modulated bias. The conversion of different chain structures is responsible for the out-of-plane polarization. The evolution is proposed to experience three stages denoted in (**e** and **f**).

reversal is caused by electric bias-induced changes in trans and gauche populations. To verify this scenario, via ePTFM, PFM amplitude loops and IR-E loops of the emerged 1120 cm$^{-1}$ absorption peak were obtained and displayed in Fig. 3e, f. A notable decrease of the PFM amplitude loop was observed when the electric bias surpassed the coercive voltage, which has been previously reported[44] in face-on P(VDF-TrFE) but rarely encountered in edge-on P(VDF-TrFE). A ferroelectricity-related inverse butterfly pattern is observed in IR-E loops at 1120 cm$^{-1}$. This pattern interconnects the evolution of the conformation and polarization, confirming our deduction. Furthermore, the IR-E loop at 1289 cm$^{-1}$ (Fig. 3g) was also measured. The butterfly pattern becomes indistinct, and the coercive electric field differs from that of PFM loops. The corresponding on-field IR-E loops are shown in Supplementary Fig. 12. This finding indicates that the -CF$_2$ rotation of the all-trans phase no longer contributes to OOP ferroelectric switching. However, the possibility that the rotation of -CF$_2$ in other conformations could contribute to polarization reversal cannot be ruled out.

A model for conformational evolution and corresponding polarization evolution is proposed in Fig. 3h. Initially, the sample was polarized by negative voltage. At the beginning of the OOP ferroelectric switching (stage 1 in Fig. 3f), there is a notable increase in IR absorption at 1120 cm$^{-1}$ with an increase in tip electric bias. This observation stands contrary to the OOP polarization shown in Fig. 3e. The synthesis of ePTFM and PFM data suggests that n-$T_3GT_3G'$ evolves from its previous conformations to provide a downwards dipole (Fig. 3h). It simultaneously suppresses the original upwards polarization, resulting in a decrease in the PFM amplitude. Moreover, pristine p-$T_3GT_3G'$ becomes unstable under the exposure of opposite electric bias, triggering its degradation. In such competition between the generation and degradation of n and p-$T_3GT_3G'$, the 1120 cm$^{-1}$ IR absorption peak reaches its maximum value, whereas the OOP polarization peak reaches its minimum value, as shown in condition (II). When the reversal process reaches stage 2, n-$T_3GT_3G'$ becomes saturated. The subsequent increase in downward polarization is mainly caused by the degradation of p-$T_3GT_3G'$. This result decreases in the IR absorption intensity at 1120 cm$^{-1}$ but increases in the PFM amplitude. In stage 3, a further increase in the electric bias results in an even lower IR absorption intensity at 1120 cm$^{-1}$, implying the onset of n-$T_3GT_3G'$ degradation at a higher bias. The degradation ultimately results in a decrease in OOP polarization, which corresponds to the unusual reduction in PFM amplitude loops (Fig. 3e). The above IR-E analysis indicates that the transformation between the trans and gauche conformations contributes to the OOP ferroelectric switching of toroidal P(VDF-TrFE) lamellae, which is significantly different from the -CF$_2$ rotation in edge-on P(VDF-TrFE).

### Nanoscale mapping for studying electrodriven evolution using ePTFM

In addition to straightforward correlations of polar evolution and molecular structure at the nanoscale, ePTFM also provides a more distinct and unambiguous nanoscale mapping of electrodriven evolution under the influence of electric bias. To provide a comprehensive understanding of the conformational evolution during ferroelectric

reversal, on-field ePTFM mapping of the IR absorption at 1120 cm$^{-1}$ was conducted on toroidal P(VDF-TrFE). As shown in Fig. 4a, the absorption at 1120 cm$^{-1}$ varies periodically along the radius direction of the toroidal domain, which is in accordance with previous report[44]. Correspondingly, two maps are constructed for vibrational modes with orthogonal dipole moments, along with local IR spectra (Supplementary Fig. 13). These results confirm that the distribution of the IR mapping was not dominated by noise or mechanical effects. To delve deeper into these observations, the 2.5 × 2.5 μm region is then selected for ePTFM mapping under an incremental electric bias. The spatial resolution reached 10 nm (Supplementary Fig. 14). The histograms derived from these mappings are displayed in Fig. 4b, and the most probable values from Fig. 4a are shown in Fig. 4c.

As the results show, nonuniform IR absorption with a mottled pattern is observed under all voltage levels. The mottled pattern denotes the coexistence of $T_3GT_3G'$ and other conformations, which affirms the aforementioned hypothesis of conformational transition. As the electric bias increases, the uniformity and intensity of the 1120 cm$^{-1}$ absorption increase, indicating the formation of $T_3GT_3G'$ conformation. A further increase in the electric bias results in a decrease in the IR absorption, aligning well with the aforementioned IR-E analysis. The histograms show that it remains nearly unchanged in peak value between 15 V and 30 V, whereas the distribution in Fig. 4a is quite inconsistent. These distributional variations imply a conformational difference between the two states, which can be explained by the conformational transition model. A repetitive experiment has been conducted to show the reproducibility of this evolution (Supplementary Fig. 15). The results exhibit the same variation. The PLL frequency maps and flat IR phase maps demonstrated a good quality of PLL tracking (Supplementary Fig. 16). Owing to the chemical specificity of on-field mapping, ePTFM provides a complementary perspective for investigating the nanoscale polar evolution of organics. The unique evolutionary characteristics of the ePTFM signal at the nanoscale are extracted and are clearly displayed in Fig. 4d, providing insights into its distinctive behavior. In comparison, the on-field PFM mappings and histogram analysis of the toroidal P(VDF-TrFE) exhibit certain limitations (Fig. 5a–c). The PFM signal contains overall polarization information, including DC-induced polarization and artifacts. Consequently, the distribution patterns and intricate details of polarization evolution can easily be buried under DC bias interference, even if electrostatic artifacts can be excluded.

## Discussion

In summary, we propose a technique named ePTFM, which is based on AFM-IR and aimed at the electrodriven evolution of polarization and molecular structure. By integrating pulse infrared excitation, electric bias modulation and photothermal force within an AFM system, the technique simultaneously enables nanoscale resolution and chemical specificity in polarization characterization, thereby complementing existing methodologies. The ePTFM method allows in situ IR-E analysis for specific polarization evolution with a spatial resolution of 10 nm. With this method, we observed the hysteretic rotation of -CF$_2$ during ferroelectric switching in edge-on P(VDF-TrFE) lamellae. The approach shows great robustness under electric bias because of the decoupling

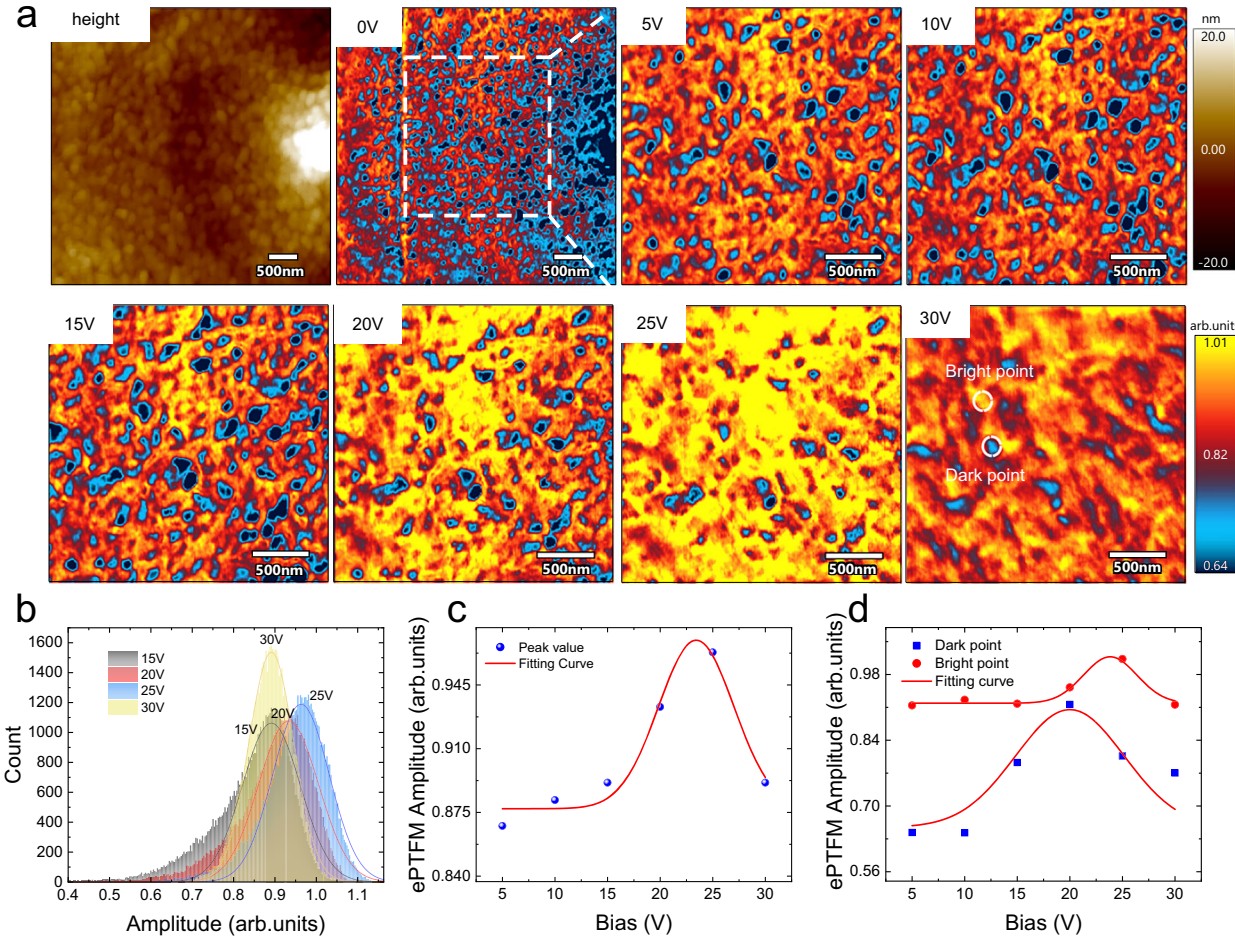

**Fig. 4 | Nanoscale mapping of electrodriven evolution via ePTFM.**
**a** Corresponding nanoscale ePTFM maps of 1120 cm⁻¹ absorption at 0 V, 5 V, 10 V, 15 V, 20 V, 25 V, and 30 V. To demonstrate the intrinsic evolution, the mapping results use the original data without flattening or further processing. The upper color box applies to height image, whereas the lower color box applies to the other images. **b** Histograms of the 15 V, 20 V, 25 V, and 30 V ePTFM mappings. The solid line represents the fitting curve of each histogram. **c** Corresponding peak value of the ePTFM signal. **d** Corresponding local ePFTM signal evolution of the dark point and bright point denoted in the 30 V ePTFM mappings of (**a**). These two points are tracked according to their contour feature and relative position.

between the DC-modulated electric bias and the probing excitation source. Moreover, through multi-wavenumber IR-E analysis, the conformational mechanism of OOP ferroelectric switching in face-on P(VDF-TrFE) with toroidal domains is revealed. By utilizing on-field mapping of ePTFM, we can obtain insights into the information related to the nanoscale morphology and molecular structure evolution under electric modulation. Owing to the penetration of IR radiation, ePTFM is also suitable for films with thicknesses ranging from several nanometers to micrometers. We believe that the proposed ePTFM approach provides fresh insight into polarization evolution, which is valuable for in-depth studies on the mechanism and controllability of ferroelectric organics and even polar organics. By incorporating an in-plane IR electric field, the analysis of dipole behavior can be elevated to a three-dimensional level. Further improvements in quantitative IR analysis and the expansion of diverse light sources may unlock even more possibilities.

## Methods

### Setup and implementation of ePTFM
The ePTFM was conducted on a modified AFM-IR platform (NanoIR-2s, Anasys Instrument, now Bruker Corporation). An electric module consisting of a signal generator (AWG3390, Keithley) and a high-voltage amplifier (ATA-2021, Aigtek) was used to apply a modulated electric bias through an AFM probe (ContE-G, BudgetSensors). A

triangular square waveform was used to acquire IR-E loops. The period of the square was set to 50 ms with a 50% duty ratio, whereas the period of the triangular square wave was set to 5 s. An optical module with a QCL laser source (MIRcat-QT, Daylight Solutions) was used to generate a polarized pulsed IR. A polarization control module was installed, and the light was p-polarized. The pulse frequency was controlled by an oscillator in a lock-in amplifier with a feedback loop (MFLI with PID, Zurich Instruments). The demodulated cantilever deflection was recorded with the DAQ in the lock-in amplifier by connecting the high-speed deflection voltage channel on the back panel of the NanoIR-2s to the input of the lock-in amplifier via the BNC connector. The bias was also recorded with DAQ by connecting the monitoring channel of the electric module to the input of the lock-in amplifier. The resulting cantilever amplitude and bias data were analyzed via a self-coded MATLAB script and producing on and off-field IR-E loops, respectively. The cantilever amplitude was sampled, and 64 sampling data points were selected equidistantly at each bias stage for analysis of the IR-E loops. Curve fitting of the IR-E loops was conducted via local regression methods in MATLAB on the basis of the average value of these 64 sampling data points.

### Fabrication and characterization of edge-on P(VDF-TrFE)
The P(VDF-TrFE) 70/30 mol% copolymer was obtained from Piezotech and dissolved in methyl ethyl ketone (MEK) at 2% w/v. For edge-on

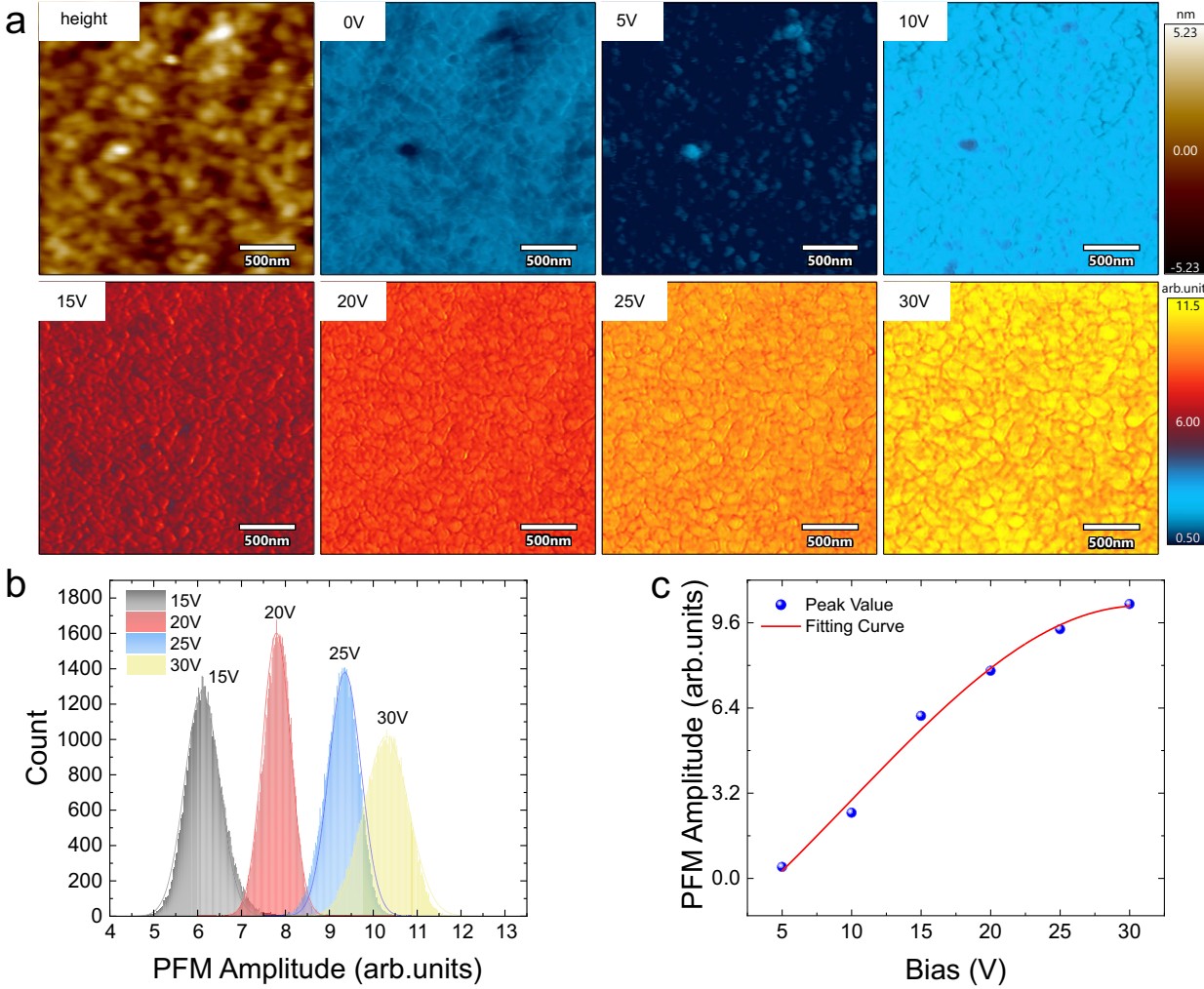

**Fig. 5 | Nanoscale mapping of electrodriven evolution using PFM.**
**a** Corresponding nanoscale PFM maps at 0 V, 5 V,10 V, 15 V, 20 V, 25 V and 30 V. To demonstrate the intrinsic evolution, the mapping result uses the original data without flattening or further processing. The upper color box applies to height image, whereas the lower color box applies to the other images. **b** Histograms of the 15 V, 20 V, 25 V and 30 V mapping. The solid line represents the fitting curve of each histogram. **c** Corresponding peak value of the PFM signal.

P(VDF-TrFE), random P(VDF-TrFE) films were first spin-coated onto silicon substrates with a Pt coating. Then, random P(VDF-TrFE) films were dried at 30 °C under vacuum for 1 h. The dried films were cut into areas of nearly 5 mm × 5 mm and covered with commercially available PTFE films. The sample was hot-pressed with a 4 kg weight at 150 °C for 30 min and then annealed at 150 °C, and the cooling rate was controlled at nearly 1 °C/min.

The AFM morphology and AFM-IR spectra were obtained via a commercial AFM instrument (NanoIR-2s, Analys Instrument, now Bruker Corporation) with an Au-coated silicon cantilever (PR-EX-nIR-10, Bruker). The morphology was obtained in contact mode with 256 points/line.

The PFM loop and IR-E loop measurements were all conducted with an ePTFM platform with a Pt-coated silicon AFM probe (ContE-G, BudgetSensors). PFM loops were acquired on the ePTFM platform by replacing the infrared pulsed excitation with an alternating current (AC) bias supplied by a signal generator. During PFM loop measurements, the tip was modulated at ~60 kHz, with the drive AC voltage set at 3 V. During the IR-E loop measurements, the repetition rate of the IR laser was in the range of 187.815 ± 14.1 kHz, matching the second-order contact resonance mode of the cantilever. The precise frequency is shown in Supplementary Table 1. The matching condition was ensured by the phase-locked loop.

For the cross-sectional morphology, an edge-on P(VDF-TrFE) film was immersed in liquid nitrogen and broken with a glass cutter. The fragment was characterized via commercial SEM (Quanta 250, FEI).

### Fabrication and characterization of toroidal face-on P(VDF-TrFE)

For face-on P(VDF-TrFE), a 1% w/v P(VDF-TrFE) solution was used and spin-coated onto Pt-coated silicon substrates at a spin speed of 6000 rpm (the film thickness was controlled at ~100 nm). After drying at 30 °C under vacuum for 1 h, the film was subsequently annealed at 200 °C for 10 min, and the cooling rate was controlled at ~0.5 °C/min.

PFM imaging was conducted via commercial scanning probe microscopy (MFP-3D Infinity, Asylum Research). A silicon cantilever with a Pt conductive coating layer (ContE-G, BudgetSensors) was used. The images were obtained in AC resonance tracking (DART) mode for both in-plane (IP) and out-of-plane (OOP) PFM measurements. During the IP-PFM measurements, the tip was modulated at ~240 kHz, with the drive AC voltage set at 3 V. The scan rate was set at 1 Hz, and the resolution of the image set was 512 points/lines. During OOP-PFM domain writing, the tip bias was set at −40 V and +40 V. For OOP domain reading, the tip was modulated at ~60 kHz, with the drive AC voltage set at 3 V. The domain remained for at least 3 days. For electrodriven PFM mapping, the image resolution was set at 256 points/

lines. The other settings were the same as above. The PFM loops and IR-E loops were conducted on the ePTFM platform.

For the ePTFM loops and mapping, a silicon cantilever with a conductive Pt-coated layer (ContE-G, BudgetSensors) was used. An electric DC bias was applied through the electric module of the ePTFM platform. The repetition rate of the IR laser was in the range of $187 \pm 14.1$ kHz, matching the second-order contact resonance mode of the cantilever. The precise frequency is shown in Supplementary Table 1. The matching condition was ensured by the phase-locked loop. The scan rate was set at 1 Hz, with the image resolution set at 256 points/lines.

## Data availability
The source data generated in this study have been deposited in the figshare database under accession code: https://doi.org/10.6084/m9. figshare.29376998.

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

## Acknowledgements

This work is supported by the National Natural Science Foundation of China (Grant No. 12472190) to X.Z., National Natural Science Foundation of China (Grant No. 12427803 and 12132020) to Y.Z., the National Natural Science Foundation of China (Grant No. 11972383 and 81827802) to W. Z., the Open Fund of Key Laboratory for Intelligent Nano Materials and Devices of the Ministry of Education NJ2022002(Grant INMD-2022M07) to H.J. The experiments reported were conducted on Research Center for Magnetoelectric Physics of Guangdong Province (No. 2024B0303390001), Physical Research Platform in School of Physics, Sun Yat-sen University (PRPSP, SYSU) and the Instrumental Analysis & Research Center, Sun Yat-sen University.

## Author contributions

S.Y. and X.Z. designed the method. S.Y. built the experimental platform and carried out the experiments, performed data acquisition and analysis. J.W. assisted with the PFM measurements and sample fabrication. H.J. and D.S. assisted with the AFM-IR measurements. And C.X. assisted in toroidal sample fabrication. W.Z. and H.J. participated in the analysis of the results and the writing of the manuscript. X.Z. and Y.Z. guided the overall research. All authors discussed the results and commented on the manuscript.

## Competing interests

The authors declare no competing interests.
