## [Transparent Peer Review file · Nature Communications]

Electrically Modulated Photothermal Force Microscopy for Revealing Molecular Configuration Changes During Polarization Switching at the Nanoscale

Corresponding Author: Professor Xiaoyue Zhang

Version 0:

Reviewer comments:

Reviewer #1

(Remarks to the Author)

1. Summary

This work developed electric modulated photothermal force microscopy (ePTFM) based on the modified AFM-IR platform, realizing nanoscale in-situ polarization and chemical analysis under electric fields. With this technique, the authors analyzed the out-of-plane polarization reversal process of P(VDF-TrFE) edge-on and face-on lamellae films at nanoscale, and proposed potential mechanism of c-axis ferroelectricity of P(VDF-TrFE).

All told, compared with FTIR measurements under electric fields, ePTFM achieves local chemical evolution analysis under electric fields with high spatial resolution. This technique is conducive to investigating the relationship between polarization and lattice in ferroelectric polymers, and it has potential applications in the electric-induced evolution of other polar organics. However, I have some concerns about this technique as well as the mechanism of the polarization reversal in face-on lamellae. Thus, I think the publication of this paper may be considered after addressing these concerns. Detailed comments are as follows.

2. Major issues

2.1. This study developed ePTFM based on the modified AFM-IR platform. Thus, it is necessary to simultaneously provide and compare the FTIR and AFM-IR spectrum of edge-on and face-on P(VDF-TrFE) lamellae to prove that such modification has not affected the basic functionalities of AFM-IR.

2.2. P(VDF-TrFE) mainly consists of two phases at room temperature, trans-planar phase and 3/1 helical phase, as indicated in the phase diagram of P(VDF-TrFE)¹. When TrFE content is 30 mol%, the polymer chains favor all-trans conformation due to the steric hindrance effect^{2,3}. In this study, the authors confirmed the γ phase by the IR peak at 1120 cm⁻¹. However, according to the FTIR spectrum in P(VDF-TrFE) edge-on lamellae films⁴ and unoriented films¹ (please refer to their Supplementary Information), obvious IR peak around 1120 cm⁻¹ can be observed. But the phase structure of these films is considered as trans-planar phase, which further suggests that the IR peak around 1120 cm⁻¹ might not be attributed to the γ phase. Kim et al. reported that IR peak around 1120 cm⁻¹ is related to ordered crystalline and disordered phases^{5,6}, not assigning to the γ phase. Additional structural characterizations should be provided to confirm the existence of γ phase in face-on lamellae films.

2.3. If γ phase in face-on lamellae films is confirmed, some issues still need to be addressed.

2.3.1. I am wondering about the underlying reasons for the formation of γ phase in face-on P(VDF-TrFE) lamellae.

2.3.2. The polarization reversal in toroidal face-on P(VDF-TrFE) lamellae still could not exclude the -CF₂ rotation in γ phase. The IR absorption peak at 1289 cm⁻¹ is related to the -CF₂ symmetric stretching vibration of trans-planar phase with conformation T_m (m>4). Therefore, the different pattern of IR-E loop at 1289 cm⁻¹ (Fig. 3g) might exclude -CF₂ rotation in trans-planar phase, but cannot exclude the -CF₂ rotation in γ phase. This means that the polarization reversal mechanism may still contain the dipole rotations.

2.3.3. If the out-of-plane polarization reversal is related to conformation transition as depicted in Fig. 3h, then what conformation might be transformed into? Are there any IR-E loops at certain IR peaks that exhibit reverse patterns compared to Fig. 3f? If yes, this may provide more hints on the polarization reversal mechanism.

2.3.4. Are the on-field and off-field IR-E loops similar under the circumstance of conformation transition? In other words, can the conformation generated under high electric fields be maintained after the removal of electric field? And why?

2.4. Could it be possible to conduct on-field AFM-IR spectrums? If it is possible, it can provide direct evidence on whether the

absorption peak shifts during the measurement. Besides, Supplementary Fig. 5 reported that the butterfly-shaped IR-E loops maintained near the IR peak around 1183 cm⁻¹. However, the shape of IR-E loops at 1183 cm⁻¹ and 1289 cm⁻¹ are reversed. I am wondering if there is a critical IR wavenumber that is related to the shape reversal? If yes, is there any implication? It is suggested to provide more IR-E loops between 1183 cm⁻¹ and 1289 cm⁻¹.

2.5. Could the authors compare the probing depth of AFM-IR and the effective depth of electric fields implemented by tips? In other words, does ePTFM suitable for ferroelectric films with a wide range of thickness?

2.6. The variation of IR absorption is quite significant in Supplementary Fig. 3 (0.020~0.028). What is the possible reason? Such relative variation (0.020~0.028) is comparable to the relative variation at 1289 cm⁻¹ (0.00555~0.00655, off-field, Fig. 2e) and 1183 cm⁻¹ (0.000375~0.000450, off-field, Fig. 2f). Does it have an adverse effect on the IR-E loop measurement, making it difficult to analysis the real signal of polarization reversal?

2.7. Could it be possible to conduct IR-E loop measurements with in-plane IR electric field? If yes, it may provide more evidence for the polarization reversal process, and may show more advantages than the Switching Spectroscopy PFM measurement.

3. Minor issues

3.1. There are several references that have not been properly cited. It is suggested to further examine this issue. Details are listed below.

3.1.1. Lines 174-175: This sentence mainly talks about the IR absorption peak of P(VDF-TrFE), while ref. 46 is about Nylon random copolymers and terpolymers.

3.1.2. Lines 235-237: The crystal structure of γ phase are shown in several articles^{7,8}. How to understand that it has more disordered T3GT3G chain conformation? Besides, it seems that ref. 47 did not discuss about whether the chain conformation of γ phase is disordered, as 'disordered' only appeared once.

3.1.3. Lines 259-260: This sentence is about the contribution of tensile strain to toroidal polar topology which was reported in ref. 40 instead of ref. 47.

3.1.4. Lines 267-269: This sentence is about the drop of amplitude in Switching Spectroscopy PFM measurement which was reported in ref. 40 instead of ref. 47.

3.1.5. Lines 297-299: This sentence is about the periodical absorption of toroidal domain which is in accordance with ref. 40 instead of ref. 47.

3.2. Lines 191-192: It is also important to point out that the transition dipole moment at 1183 cm⁻¹ is perpendicular to polymer chains.

3.3. Lines: 270-271: The pattern of Fig. 3g is still butterfly-shaped, and thus it might be inaccurate to state that butterfly pattern is not obvious. It is suggested to report the difference in coercive field instead.

3.4. Figure 2b: Figure 2b and its caption is not clear enough to readers. It is suggested to include IR electric field direction in Figure 2b, and the meaning of arrows with difference colors in the caption of Figure 2b.

3.5. Figure 3f, g: It is suggested to point out whether the IR-E loop is on-field or off-field in the caption even if they may be similar in pattern.

3.6. Figures 4: It seems that electric-induced conformation evolution behaviors in different regions are different. If yes, it is suggested to demonstrate such different behaviors to better show the advantages of such nanoscale characterization over the FTIR measurements.

4. Reference

1. Liu, Y. et al. Ferroelectric polymers exhibiting behaviour reminiscent of a morphotropic phase boundary. *Nature* 562, 96-100 (2018).
2. Furukawa, T. Ferroelectric properties of vinylidene fluoride copolymers. *Phase Transitions* 18, 143-211 (1989).
3. Costa, C. M. et al. Smart and Multifunctional Materials Based on Electroactive Poly(vinylidene fluoride): Recent Advances and Opportunities in Sensors, Actuators, Energy, Environmental, and Biomedical Applications. *Chemical Reviews* 123, 11392-11487 (2023).
4. Guo, M. et al. Flexible Robust and High-Density FeRAM from Array of Organic Ferroelectric Nano-Lamellae by Self-Assembly. *Adv Sci* 6, 1801931 (2019).
5. Kim, K. J., Reynolds, N. M. & Hsu, S. L. Spectroscopic Analysis of the Crystalline and Amorphous Phases in a Vinylidene Fluoride/Trifluoroethylene Copolymer. *Macromolecules* 22, 4395-4401 (1989).
6. Prabu, A. A., Lee, J. S., Kim, K. J. & Lee, H. S. Infrared spectroscopic studies on crystallization and Curie transition behavior of ultrathin films of P(VDF/TrFE) (72/28). *Vibrational Spectroscopy* 41, 1-13 (2006).
7. Cui, Z., Hassankiadeh, N. T., Zhuang, Y., Drioli, E. & Lee, Y. M. Crystalline polymorphism in poly(vinylidene fluoride) membranes. *Progress in Polymer Science* 51, 94-126 (2015).
8. Zhang, L. et al. Recent Progress on Structure Manipulation of Poly(vinylidene fluoride)-Based Ferroelectric Polymers for Enhanced Piezoelectricity and Applications. *Advanced Functional Materials*, 2301302 (2023).

Reviewer #2

(Remarks to the Author)

In "Electric Modulated Photothermal Force Microscopy for Revealing Chemical Origins of Polarization Switching at Nanoscale" Yao and coworkers merge two AFM based techniques (piezoresponse force microscopy – PFM - and AFM-IR (an infrared nano spectroscopic method). The resulting method, named by the authors electric modulated photothermal force microscopy (ePTFM), leverages both infrared optical pulses (as in AFM-IR) and electrical bias (as in PFM). In this work the authors use ePTFM to characterize electro-driven ferroelectric switching in edge-on poly(vinylidene fluoride-ran-trifluoroethylene [P(VDF-TrFE)] lamellae and in toroidal face-on P(VDF-TrFE).

The proposed technique is in principle quite interesting. However, the manuscript will benefit from restructuring some of the

technical information and from additional technical details. In the current form, the manuscript is a bit hard to read and of difficult assessment since the technical details are spread between main text, experimental and SI, requiring much back and forth by the reader. Since the technique has not been reported before, the bulk of technical details shall be presented and explained in the main text. In summary, I find the work interesting, sufficiently novel, and it could become publishable, provided the authors can make major changes to address the comments below:

1) There are many timescales, frequencies and polarization directions (i.e. of the sample, of the IR light, of the applied bias) contributing to the measurement, but not all details are specified. Is the AFM tip gold coated? Or what is it made of? What is the repetition rate of the IR laser in the context of the measurement? Is the IR laser matching one of the cantilever mechanical resonances (like in resonance enhanced AFM-IR) or not? If it is matching, how is the matching condition ensured?

It would be useful to expand and define the timescale of Fig 1b to see how the IR signal changes within the 50 ms intervals of the applied bias. The authors should specify the direction of the linearly polarized IR light. In the NanoIR-2 tool used by the authors the IR polarization is not typically controlled unless an additional polarization controller is used. Was the polarization control module installed and used? If so, was the light polarization s- or p- polarized? (Given the side illumination, if s-polarization is used the light pulses are linearly polarized in the sample plane, i.e. parallel to the substrate, but if p-polarization is used the light polarization has both in plane and out of plane components with respect to the substrate/sample plane. If the polarization control module is not used, then the light polarization at the sample is typically not controlled nor known). Please clarify.

2) The main claim that ePTFM, adds chemical specificity to PFM is spectroscopically unconvincing and thus far unsubstantiated since all ePTFM experiments are obtained looking at the signal intensity at fixed wavelength, which is not sufficient proof. In fact, several effects unrelated to composition or orientation can/could influence the signal intensity in AFM-IR detection such as the sample mechanical properties or the applied bias. To claim chemical specificity and determination of the sample electrical polarization direction the authors shall spectroscopic evidence plotting the evolution of the IR spectra as a function of the applied bias, not just at selected wavelengths.

3) For face-on P(VDF-TrFE) with toroidal domain is nicely characterized by PFM (Fig 3b) which show the hierarchical structure of the toroidal pattern over an area of several micrometers wide, instead the ePTFM maps in Fig 4 are shown over a much smaller area. For the ePTFM study to be convincing the authors should provide at least 2 maps for vibrational modes with orthogonal dipole moment at the same scale of Fig 3b and corroborate the domain assignment with local IR spectra. In this context, the data in Fig 4 are not convincing and appear to be dominated by noise or by properties extrinsic to the sample electrical polarization, perhaps due to mechanical effects.

4) A very similar infrared photocurrent nano-spectroscopy technique operating at zero bias and providing nanoscale spectra (instead of just single wavelength measurements) was reported recently using the same AFM platform (see Venanzi et al. Appl. Phys. Lett. 123, 153509 (2023)) and should be cited. The same work (page 3) also reports that “the signal to noise ratio at finite DC voltage bias (i.e. 40mV) between tip and substrate is not as good since it is dominated by the mechanical noise”. The ePTFM data reported by Yao et al. (Fig 2, 3 and 4) at much larger AC voltages (i.e. -35V/+35 V in Fig 2) appear to be noisy (perhaps to the point of becoming untrustworthy). Can the authors comment how the applied bias affects the noise in their measurements?

5) Another electrically detected photothermal nanospectroscopy method (Katzenmeyer et al. Nanoscale, 2015,7, 17637-17641) reported several years ago should be cited as it is relevant in the context of the method developed in this work.

Reviewer #3

(Remarks to the Author)

Yao et al. Electric Modulated Photothermal Force Microscopy for Revealing Chemical Origins of Polarization Switching at Nanoscale

Overview:

Yao et al. claim to report the simultaneous analysis of ferroelectric polarization and chemical specificity, using modified piezoresponse force microscopy on an organic ferroelectric. The underlying hypothesis—that understanding the chemistry is essential while altering the polarization—is not immediately clear to the general reader, nor is it explained why these aspects cannot be measured separately. Given that the novelty of in-situ analysis is pivotal to the paper, this needs to be articulated more clearly.

While some data sets appear robust, the manuscript is hindered by poor writing quality and structure, alongside numerous technical challenges and misleading sections, making it incredibly difficult to comprehend. The inaccessibility to a general audience, combined with misleading content and ambiguity regarding the real impact of the work, leads me to conclude that I cannot recommend its publication in Nature Communications.

General points:

The manuscript is challenging to read, primarily due to the frequent use of specialized terminology without proper introduction, such as the term “gauche defects”. For a broad readership, it is essential to introduce and explain such terms. Additionally, the poor structure of the paragraphs contributes to the difficulty in reading. This lack of clear organization makes

it hard to distinguish between new findings and established knowledge.

“However, the lack of electric modulation makes it still difficult to correlate the polarization behavior with the evolution of specific molecular structure in nanoscale.”

The authors emphasize the importance of chemical specificity and polarization, but its significance may not be immediately clear to a general audience. One might question, for instance, whether the chemical composition could be readily mapped using techniques such as EDX (Energy Dispersive X-ray Spectroscopy). Additionally, the manuscript should address whether the authors believe that the chemistry changes with ferroelectric switching. If so, it is pertinent to question whether the material still conforms to the definition of a ferroelectric, traditionally characterized as a material exhibiting switchable net remanent polarization arising from a symmetry-lowering phase transition.

“For other approaches, transmission electron microscopy (TEM) may be able to achieve nanoscale chemical and polar analyses, whereas it may cause radiation damage for organics.”

I believe this is readily solved by using cryo TEM. This would also give much higher resolution imaging.

The manuscript does not clearly articulate the extent of information this approach can provide without preexisting knowledge. For instance, in Figure 1(c), while explaining the technique, the authors mention a significant response when the dipole is aligned with the AFM (Atomic Force Microscopy) tip. Although this is similar to vertical PFM (Piezoresponse Force Microscopy), as discussed in specific points below, it remains unclear whether this technique would yield new insights on an unknown sample.

I have a specific question regarding the response time of materials to thermal effects. How does this response time correlate with the scan speed and other related phenomena in ferroelectrics?

Specific points

“In brief, our technique integrates optical radiation, modulated electric bias and photothermal effect into an AFM system to achieve chemically specific characterization of polarization evolution in nanoscale.”

The claim about the effectiveness of your approach is reiterated often, yet it remains unclear to me as a reviewer whether you have conclusively established this. To substantiate your claim, I suggest demonstrating PFM (Piezoresponse Force Microscopy) independently, alongside a separate chemically specific characterization. Following this, it would be necessary to establish that these two methods measure different aspects (as referenced in another question). Only then, by combining both approaches, could you convincingly demonstrate the generation of new information.

“However, the lack of electric modulation makes it still difficult to correlate the polarization behavior with the evolution of specific molecular structure in nanoscale.”

Do the authors believe the structure changes during switching? Perhaps they mean reorientation of a molecule with the same structure?

Figure 1(c), the smaller amplitude of tip oscillation when the polar moment is in-plane rather than out of plane: is the same information not included in the PFM? Dipole moments out of plane will show up in the vertical PFM while dipole moments in plane will show in plane PFM. where a parallel

In Figure 1(c), you note a smaller amplitude of tip oscillation when the polar moment is in-plane, as opposed to out-of-plane. However, isn't this information already captured in Piezoresponse Force Microscopy (PFM)? Specifically, dipole moments oriented out-of-plane would be evident in vertical PFM, while in-plane dipole moments would be observable in lateral PFM.

The introduction is written for general cases, but the references focus on very specific studies. The authors would be encouraged to include some reviews or more general articles.

“Only when nanoscale polarization and chemical component analysis is integrated, the mechanism and controllability principles of organic ferroelectrics can be established with certainty, other than empiricism or speculation^{14,15}.”

Do the authors propose that organic ferroelectrics adhere to different physical principles compared to conventional ferroelectrics? If this is the case, I would appreciate a detailed justification, supported by relevant references. From my perspective as a referee, while the origin of the polar component in organic ferroelectrics might differ, the ferroelectric properties of that component are anticipated to be analogous to those in conventional materials. This similarity is one of the reasons Piezoresponse Force Microscopy (PFM) is applicable to organic ferroelectrics.

“To understand the polarization evolution, lots of experimental techniques have been proposed, in which the use of these techniques still have constraints in monitoring chemical components during polarization evolution in nanoscale.”

Could the authors provide references supporting the proposed experimental techniques? Are these techniques comparable to those long-established for measuring ferroelectrics in inorganic materials, such as Piezoresponse Force Microscopy (PFM)? For conventional ferroelectrics, there are numerous recent publications that offer overviews of these techniques. An example can be found at <https://doi.org/10.1515/psr-2019-0067>, which could be a useful reference in this context.

“Its signal often includes the unwanted contributions of bias-induced polarization and electrostatic response caused by cantilever-sample capacitance, which is detailedly described in Supplementary Note 3.”

This representation is highly misleading. While such effects may occur, they are not intrinsic to the technique. Proper operation by the user can effectively eliminate these issues. Please refer to the specific comments below for further details.

SI, "As a result, in addition to remanent polarization, the on-field PFM amplitude also includes the contributions of polarization induced by DC bias (PDC=0VDC) and electrostatic response caused by cantilever-sample capacitance. And because DC modulated bias is coupling 60 with AC probing bias, it is also unable to separate the contribution in frequency domain."

I would like to request clarification on a specific point from the authors:

- 1) The statement regarding the use of PFM (Piezoresponse Force Microscopy) is somewhat ambiguous. Typically, PFM operates without a DC bias; for switching experiments, poling can be conducted (a DC voltage applied and removed) followed by scanning with an AC voltage and zero DC voltage. Could the authors elaborate on what they mean in this context?
- 2) Regarding the inability to separate contributions in the frequency domain: could the authors clarify this point? Variations in resonant frequency during scanning are precisely the basis of Dual AC Resonance Tracking (DART) mode PFM.

SI "Therefore, electrostatic response appears as artifact, and the influence of PDC blurs the intrinsic information in PFM when analyzing polar evolution."

Could you specify which electrostatic response is being referred to? Piezoresponse Force Microscopy (PFM) is widely used in inorganic materials, often characterized by substantial polarization (e.g., PZT, BaTiO₃, LiNbO₃), without encountering significant issues. Please detail the specific problem you are addressing. Additionally, consider the recent pivotal work by Killgore et al., which effectively demonstrates a simple way to mitigate such effects when they are present [DOI: <https://doi.org/10.1039/D2NA00046F>].

SI, "Specifically, infrared absorption is directly correlated with the absorptive chemical group whose characteristic frequency reaching about terahertz in the experiment which is far away the DC-induced dipolar movement. Thus, DC-induced polarization are unable to contribute IR absorption signal and the influence is excluded."

This section may not be convincing to a general audience. Could you please elaborate on how the infrared (IR) absorption changes as a function of an applied electric field? Specifically, if electric fields induce changes in the unit cell through electret effects, would these not also lead to alterations in IR absorption?

As an additional point, the term "DC-induced polarization" appears to describe a paraelectric effect, or possibly an electret effect, depending on the specifics. How do the authors perceive this contributing to Piezoresponse Force Microscopy (PFM)? To a general audience, it might seem more plausible that such effects would influence infrared (IR) absorption to a greater extent than the relative piezo response, considering that the former is an absolute measurement while the latter is relative.

Going further, in the main text you contradict this sentence "The ePTFM enables the identification of the two cases by observing variations of polarization-dependence infrared absorption at multiple wavenumbers." I.e. that the DC polarisation does change the absorption. This lack of consistency is very concerning.

SI "Consequently, the on-field and off-field IR-E loops are consistent."

Please show experimental evidence to back up this claim.

"Because the use of linear polarized laser source, the variation of IR absorptions can reflect both the changes in chemical concentration and molecular orientation during electric modulation"

This is a nice point, if you can show that you can use certain wavelengths to track specific molecules it would add significantly to your story.

"Due to the coupling of AC probing electric bias and DC modulated electric bias, it is difficult to separate these contributions."

No it isn't. They are at different frequencies and lockin amplifiers do this trivially.

"However, ePTFM signal is directly determined by the IR absorption of chemical group, which is decoupling with DC modulated electric bias."

"The ePTFM enables the identification of the two cases by observing variations of polarization-dependence infrared absorption at multiple wavenumbers."

The manuscript contains two statements that seem to directly contradict each other: Does the polarization change the infrared (IR) absorption or not? Intuitively, I am inclined to agree with the statement suggesting that the absorption is likely polarization-dependent, and that DC modulation is intended to specifically alter the polarization.

"The utilization of pulsed infrared radiation is also able to get rid of electrostatic interference or ionic artifact induced by direct current (DC) electric bias, which is usually encountered in PFM^{33,34}."

These issues are not typically encountered in standard operation. They may arise if the Piezoresponse Force Microscopy (PFM) is not operated correctly or if the sample is leaky. However, such issues are not inherent to the proper operation of PFM.

"...but also simultaneously contain chemical correlations which is absent in PFM loops."

Could the authors clarify their statement? Are they suggesting that during ferroelectric switching, they can specifically monitor a molecule (such as CF₂) to observe its switching? Since this mechanism appears to be known, its significance in this context is not immediately apparent. Could the authors please elaborate on the benefits or novel insights this approach provides? While there may be interesting observations in this research, for publication in a general journal, the results and their implications need to be clearly understandable to non-specialists, which, at present, they are not.

"In addition, the repeatability and the exclusion of artifacts arising from shifting of absorption peak under electric bias was conducted and shown in Supplementary Figure S4 and Supplementary Figure S5."

The authors mention repeating the loops once, but it's not clear if attempts were made to repeat them more than once. The lack of clarity is compounded by Figures S4 and S5, which seem to show no shift in absorption peak under bias. This casts further doubt on the statement, especially considering that other data (Figure 3 f and g) appear to indicate an absorption shift. To conclusively demonstrate this, one diagnostic approach could be to measure the absorption spectra at different field strengths, for example, at 0 V, 10 V, 20 V, and 30 V.

"The ePTFM not only enables chemical correlation, but also produces a signal with more certainty."

Could you specify what the "more" in "more certainty" is relative to? The term "more" implies a comparison, so it would be helpful to know what the baseline or reference point is that you are comparing against.

"...two set of data are sorted and processed 167 according to whether the DC electric bias is applied, yielding the on-field/off-field PFM loops³⁸."

Could the authors clarify what they mean by "on and off field PFM loops"? The referenced citation pertains to switching spectroscopy PFM, which involves observing hysteresis loops with a small AC voltage superimposed on a DC voltage. However, a search of the manuscript reveals no mention of "on-field" or "on-bias." Understanding these terms is crucial for comprehending the manuscript.

"...in Fig 2d, the on-field PFM amplitude loop exhibits obviously lower coercive voltage and unsaturable amplitude than off-field."

The interpretation of these loops is challenging, as the manuscript does not clearly define what is meant by "on and off field." However, if the authors are suggesting that loops obtained using AC+DC voltages differ from those obtained with a large AC voltage alone, this observation is indeed correct. They also note that the 'coercive field' varies with the frequency of the AC voltage. This phenomenon is well-known and observable by taking bulk P(E) loops at different frequencies. Consequently, quantifying the significance of their findings in relation to these loops is difficult.

"In fact, the differences between on-field and off-field PFM amplitude loops are the results of the coupling between DC modulation and AC excitation electric bias, and the replacement of AC electric bias with pulsed infrared laser in ePTFM successfully alleviated this influence as discussed above and in Supplementary Note 3."

This sentence makes no sense, and the information in note 3 is subject to critique (see above).

"These results demonstrated that ePTFM will be a powerful tool for in-situ studies of nanoscale polarization evolution and will become more scientifically valuable in ferroelectric organics."

As evident from the detailed comments provided in the preceding seven pages, I, as the referee, do not concur with this position.

"(d) PFM phase distribution of the same region. The square pattern in phase and amplitude was still exist after several day confirming the out of plane ferroelectricity."

The duration of stability does not confirm ferroelectricity. While some ferroelectric materials may back switch within hours, certain surface charges can remain stable for days. It is important to note that your Figures e and f do provide strong evidence of ferroelectricity. However, using time as a defining criterion for ferroelectricity is not particularly useful or definitive.

Regarding Fig S6, I'm curious as to why a 'box-in-a-box-in-a-box' poling approach was not employed? This method typically involves poling a large area positively, a smaller area negatively, and an even smaller area positively again.

For a general journal please explain what a gauche defect is.

"Analogous to this, we believe such strain also facilitates the transformation of chain conformation under electric bias. And it is a nature deduction that the OOP polarization reversal is essentially the result of electric bias inducing conformational change of polymer chains."

This section is quite difficult to read and seems to contain speculative content. Could you please clarify what is meant by "strain facilitate"? The term is ambiguous and needs further explanation.

"Meanwhile, the pristine p-T3GT3G' are unstable under the opposite electric bias and begin to degrade."

If you cannot switch it, can you really call it a ferroelectric?

"The mottled pattern denotes the co-existence of T3GT3G' and other configurations, which affirms the aforementioned hypothesis of configurations transition."

Could the authors explain how they have ruled out the possibility that the observed pattern is merely a result of strain gradients arising from topographical interactions? The detection of ferroelastic domains using PT IR has already been established in the literature [DOI: 10.1126/sciadv.1602165].

Version 1:

Reviewer comments:

Reviewer #1

(Remarks to the Author)

with all my concerns addressed, this paper can be accepted for publication in Nature Communications.

Reviewer #2

(Remarks to the Author)

see attached file

Reviewer #3

(Remarks to the Author)

The authors have conducted an extensive amount of work and have provided detailed responses to all questions raised. This additional effort, particularly the high-quality new data sets, has significantly improved the quality of the manuscript. While I still disagree with the authors on several points in the review process, this disagreement is on the level of scholarly debate, and the authors have justified their position well.

Therefore, I see no reason why this work cannot be published as it stands.

Version 2:

Reviewer comments:

Reviewer #3

(Remarks to the Author)

I have been asked to assess the authors' responses to Reviewer 2's comments and determine whether they have been satisfactorily addressed. I will address each point in detail below; however, in my opinion, none of the responses adequately address Reviewer 2's concerns.

Comment 1:

Reviewer 2 has asked for improvements in the writing quality, making this the second request for the same thing. To offer an objective perspective on the updated writing quality, I have utilized a large language model for feedback. The following are the five key issues identified by the model (all of which I agree with):

- 1) Clarity and Precision: Some sentences are overly complex, with nested clauses or passive constructions that obscure the meaning. Simplifying sentence structure and using more active voice could improve the clarity.
- 2) Consistency in Terminology: The manuscript uses some terms interchangeably (e.g., polarization switching, ferroelectric reversal) without clear definitions for each. It would be helpful to define these terms early on and use them consistently.
- 3) Redundancy and Repetition: There are instances of repetition, particularly in the Results and Discussion sections, where similar points are reiterated. Eliminating redundancy could make the paper more concise and easier to follow.
- 4) Transitional Phrases: The flow between paragraphs and sections can be improved by using more effective transitional phrases. Some paragraphs feel disjointed because they jump from one concept to another without a clear link.
- 5) Jargon and Technical Language: While the manuscript is meant for a technical audience, it sometimes relies heavily on jargon without sufficient explanation. Providing brief explanations or definitions for highly specialized terms (e.g., "toroidal face-on P(VDF-TrFE)") could help make the manuscript more accessible.

Comment 2:

Information requested: "...What is the frequency of the cantilever contact resonant frequency?"

Information given: "about 200 kHz".

The term "about" is not a precise number. This is a fundamental and important question. If the authors are unable to provide an exact frequency (which should be recorded in both their lab book and AFM data files), it raises concerns about their experimental proficiency and casts doubt on the reliability of the data as a whole.

Comment 3:

The reviewer requested the full cycle data sets, which the authors have declined to provide, citing sample damage.

However, I expect this is precisely why the reviewer asked for both spectra—to assess the extent of the damage.

If the authors believe the sample degrades at 10V, it seems counterintuitive to then apply voltages of 20V and 30V. Wouldn't it be more prudent to experiment with voltages below 10V in order to avoid damaging (changing) the sample?

Additionally, in the next section, the authors claim that $\pm 20V$ switches their sample, raising the question: does this switching voltage cause damage to the sample? Could this suggest that the PFM signal originates from an electret effect?

In my opinion, if the measurement process itself alters the sample, the resulting data becomes problematic and does not provide meaningful insight into the true properties of the sample. For instance, does the data reflect the initial state, a transient state, or a final, degraded state?

Furthermore, when introducing a new technique, it is essential to demonstrate its reliability. At a minimum, this requires strict control over variables and so the authors statement that "...properly controlling the sample quality, testing bias, laser parameters, and contact force between the tip... is beyond the scope of our work", is tough to accept.

Comment 4:

The reviewer requested additional ePTFM data and suggested that it showed a contradiction with the existing data. While I understand the authors' argument that strong out-of-plane contrast may not be expected in this particular orientation, I find the response unconvincing. In my opinion, further investigation and additional data are necessary to adequately address this issue.

Comment 5:

The reviewer rightfully pointed out the incorrect statement: "The probing depth of AFM-IR is determinate by experiment setting." Although you acknowledge in your response that "the probed depth is still under investigation and not yet fully understood," the suggestion that increased power simply results in greater depth is problematic. While higher power may indeed increase the volume of the sample being probed, this is not equivalent to having precise control over the probing depth. This distinction is crucial.

I recognize that this might stem from a linguistic issue; however, for a published paper, the language must be both technically accurate and precise. Misstatements like this can lead to confusion and cast doubt on the overall rigor of the manuscript. It also raises concerns about whether other such inaccuracies may have gone unnoticed.

Comment 6:

The reviewer expressed concerns over the use of the term "chemical structure" in the manuscript. While the authors have made changes to the wording in many places, which is commendable, the term "chemical origins" remains in the title. In my view, this creates a misleading title, as it suggests a deeper exploration of chemical mechanisms than is actually present in the study. Going forward, I suggest that "molecular configuration" might be a better term.

Version 3:

Reviewer comments:

Reviewer #4

(Remarks to the Author)

Reviewer #5

(Remarks to the Author)

As I understand, I am asked to assess authors' Response to the original reviewer as well as a followup referee. I have read the Response carefully, and it appears to me that the authors Response is adequate. One of the key issues is whether the technique damages the sample, and the authors articulated that the technique they proposed, cycling under a fixed wavenumber, does not damage the sample, while the suggested checkup, cycling under the full spectrum, could damage the sample. The response is quite convincing. Other issues raised are pretty minor to me, and the authors response is satisfactory.

The technique proposed is quite promising to me. As the authors explained in their motivation, STEM does damage the organic, or organic-inorganic sample, even under low doses; see for example Nature Communications volume 9, Article number: 4807 (2018); and the sample preparation for STEM could alter polar topology as well; see for example Nature Communications volume 12, Article number: 4620 (2021). The technique proposed thus provide a valuable alternative to probe both polar and chemical structures at nanoscale.

Version 4:

Reviewer comments:

Reviewer #6

(Remarks to the Author)

I believe this manuscript represents a significant advance and is appropriate in its current form for publication in Nature Communications. The authors demonstrate the ability to dynamically modulate the polarization of ferroelectric materials while measuring corresponding changes in photothermally detected IR absorption with nanoscale spatial resolution. In particular, the ability to perform chemically specific measurements of ferroelectric hysteresis loops at multiple IR bands is very compelling. This enables the authors to track specific molecular vibrational modes during polarization switching, offering insight into the molecular mechanisms behind ferroelectric behavior—including conformational transitions that are otherwise difficult to resolve.

The latest revision of the manuscript has satisfactorily addressed previous reviewer concerns. In particular, the negative control experiment using non-ferroelectric PMMA is well designed and convincingly rules out spurious contributions to the IR-E signals.

I expect the publication of this manuscript will attract broad interest, particularly among researchers in ferroelectric materials, nanoscale spectroscopy, and organic electronics. I strongly recommend publication, either as-is or with attention to the following minor suggestions for clarity:

Minor Suggestions for Improvement:

1. Figure 1a (Lock-in Amplifier Detail):

The diagram includes internal lock-in amplifier components such as mixers, low-pass filters, and X/Y-to-R conversion, which are standard and well understood. These details may distract from the core technical innovation being illustrated. I suggest simplifying this part of the figure to a single "Lock-in Amplifier" block, allowing the reader to focus on the ePTFM configuration and its conceptual novelty.

2. Figure 1c (Molecular Representation Clarity):

a. The yellow glow around one molecule is unexplained and somewhat obscures the "+" charge marker. Consider either removing the glow or clarifying its intended meaning (if any).

b. The charge indicators (+/-) are subtle and could be made more prominent.

c. Adding a small vector arrow to indicate the dipole orientation of the selected molecule may help communicate the principle more intuitively.

d. The "IR electric field" label and arrow currently appear only on the right-hand panel. Initially, this caused some confusion—was IR illumination present only in that frame? To clarify, I suggest either centering the label between the two panels or moving it to Figure 1a, where the system schematic could establish the IR field's consistent direction across all panels.

3. Terminology: "Recently Developed" AFM-IR

The manuscript describes AFM-IR as "recently developed" but the first seminal paper on AFM-IR was published 20 years ago (citation below) and AFM-IR has been commercialized for at least 15 years. At this point AFM-IR is more accurately described as an established and widely used technique. I suggest modifying the wording accordingly.

A. Dazzi, R. Prazeres, F. Glotin, and J. M. Ortega, "Local infrared microspectroscopy with subwavelength spatial resolution with an atomic force microscope tip used as a photothermal sensor," *Opt. Lett.* 30, 2388-2390 (2005)

Response to the comments of reviewer #1

1. Summary

This work developed electric modulated photothermal force microscopy (ePTFM) based on the modified AFM-IR platform, realizing nanoscale in-situ polarization and chemical analysis under electric fields. With this technique, the authors analyzed the out-of-plane polarization reversal process of P(VDF-TrFE) edge-on and face-on lamellae films at nanoscale, and proposed potential mechanism of c-axis ferroelectricity of P(VDF-TrFE).

All told, compared with FTIR measurements under electric fields, ePTFM achieves local chemical evolution analysis under electric fields with high spatial resolution. This technique is conducive to investigating the relationship between polarization and lattice in ferroelectric polymers, and it has potential applications in the electric-induced evolution of other polar organics. However, I have some concerns about this technique as well as the mechanism of the polarization reversal in face-on lamellae. Thus, I think the publication of this paper may be considered after addressing these concerns. Detailed comments are as follows.

Response:

We sincerely thank the reviewer for his/her time and effort in carefully reviewing our manuscript. And we also appreciated reviewer's valuable comments and suggestions. We will respond to these comments one by one below. The comments, responses and revisions are highlighted in blue, black and red respectively.

2. Major issues

Comment 2.1 of Reviewer #1: "This study developed ePTFM based on the modified AFM-IR platform. Thus, it is necessary to simultaneously provide and compare the FTIR and AFM-IR spectrum of edge-on and face-on P(VDF-TrFE) lamellae to prove that such modification has not affected the basic functionalities of AFM-IR."

Response:

Thanks for the reviewer's comment and suggestion. The IR spectra on both modified AFM-IR platform and original AFM-IR system have been conducted under the same engaged point. The results are shown in Fig. R1. The spectra show the same IR absorption peak position and shape, which prove that the basic functionalities are not affected. In fact, the ePTFM replaces the signal processing pathway of original AFM-IR system and uses modified AFM tip to applied bias, which will not influence the functionalities of AFM-IR. The comparison of FTIR spectrum also shows that AFM-IR

share the same absorption peak position with FTIR. It has been broadly discussed and recognized (ref. 1,2) that AFM-IR is directly correlated to FTIR spectra. The slight difference may be caused by the different characterization scale between AFM-IR and FTIR. The AFM-IR characterizes nanoscale IR absorption instead of bulk FTIR absorption.

Fig. R1. Edge-on P(VDF-TrFE) IR Spectrum acquired in different platforms. a Comparison between modified ePTFM platform and original AFM-IR. **b** Comparison between FTIR and original AFM-IR platform.

Reference:

1. Mathurin, J. et al. Photothermal AFM-IR spectroscopy and imaging: Status, challenges, and trends. *Journal of Applied Physics* 131, 010901 (2022).
2. V. D. dos Santos, A. C., Lendl, B. & Ramer, G. Systematic analysis and nanoscale chemical imaging of polymers using photothermal-induced resonance (AFM-IR) infrared spectroscopy. *Polymer Testing* 106, 107443 (2022).

Revisions:

According to these comments, a statement on the affection of such modification had been supplemented.

Please refer to line 119-121 in manuscript highlighted in red and Supplementary Fig. 2 in Supplementary Materials.

“The IR spectrum acquired using ePTFM and original AFM-IR is shown in Supplementary Fig. 2, which shows that such modification has not affected the basic functionalities of AFM-IR.”

Comment 2.2 of Reviewer #1: “P(VDF-TrFE) mainly consists of two phases at room temperature, trans-planar phase and 3/1 helical phase, as indicated in the phase diagram of P(VDF-TrFE)¹. When TrFE content is 30 mol%, the polymer chains favor all-trans conformation due to the steric hindrance effect^{2,3}. In this study, the authors confirmed the γ phase by the IR peak at 1120 cm⁻¹. However, according to the FTIR spectrum in P(VDF-TrFE) edge-on lamellae films⁴ and unoriented films¹ (please refer to their Supplementary Information), obvious IR peak around 1120 cm⁻¹ can be observed. But

the phase structure of these films is considered as trans-planar phase, which further suggest that the IR peak around 1120 cm^{-1} might not be attributed to the γ phase. Kim et al. reported that IR peak around 1120 cm^{-1} is related to ordered crystalline and disordered phases^{5,6}, not assigning to the γ phase. Additional structural characterizations should be provided to confirm the existence of γ phase in face-on lamellae films.”

Reference:

1. Liu, Y. et al. Ferroelectric polymers exhibiting behaviour reminiscent of a morphotropic phase boundary. *Nature* 562, 96-100 (2018).
2. Furukawa, T. Ferroelectric properties of vinylidene fluoride copolymers. *Phase Transitions* 18, 143-211 (1989).
3. Costa, C. M. et al. Smart and Multifunctional Materials Based on Electroactive Poly(vinylidene fluoride): Recent Advances and Opportunities in Sensors, Actuators, Energy, Environmental, and Biomedical Applications. *Chemical Reviews* 123, 11392-11487 (2023).
4. Guo, M. et al. Flexible Robust and High-Density FeRAM from Array of Organic Ferroelectric Nano-Lamellae by Self-Assembly. *Adv Sci* 6, 1801931 (2019).
5. Kim, K. J., Reynolds, N. M. & Hsu, S. L. Spectroscopic Analysis of the Crystalline and Amorphous Phases in a Vinylidene Fluoride/Trifluoroethylene Copolymer. *Macromolecules* 22, 4395-4401 (1989).
6. Prabu, A. A., Lee, J. S., Kim, K. J. & Lee, H. S. Infrared spectroscopic studies on crystallization and Curie transition behavior of ultrathin films of P(VDF/TrFE) (72/28). *Vibrational Spectroscopy* 41, 1-13 (2006).

Response:

The reviewer’s comment and suggestion are appreciated. Additional structural characterization has been conducted (Fig. R2), which shows more content of TTTG in face-on lamellae. Moreover, additional literatures are provided to support that 1120 cm^{-1} is attributed to TTTG structure. Further, we have read reviewer’s references in detail. We found it is not contradicted to our results either. The detail response is shown below. In revised manuscript, we will provide more discussion and the assignment has been toned down.

Considering the complexity of chain conformation in P(VDF-TrFE), we would like to use T and G sequence to describe the chain arrangement, which might be clearer for the discussion. The crystalline phase is actually the aggregation of identical chain conformations with long-range ordering. For examples, trans-planar phase features long-range ordering TTTT chain conformation. And the γ phase features long-range ordering TTTG conformation. The amorphous regions exhibit irregular mix of TG, TT and TG’, resulting TTTG, TGTG’ and 3/1 helical conformations.

In our opinion, even when TrFE content is 30 mol%, there still may exist non-negligible amount of other conformation, rather than just TTTT. P(VDF-TrFE) was semicrystalline, which unavoidably consists amorphous regions and thus contains TTTG, TGTG’ and 3/1 helical conformations. Therefore, it is reasonable that P(VDF-

TrFE) edge-on lamellae in Guo, M. et al.'s report and unoriented films in Liu, Y. et al.'s report have 1120 cm^{-1} IR peak (ref. 1, 2). Therefore, it cannot rule out the contribution of 1120 cm^{-1} to TTTG.

In fact, Liu, Y. et al.'s report supports our opinion (ref.1). One of the main conclusions is that the TrFE monomers lead to order-to-disorder evolution. With the increase of TrFE content, not only the absorption peak of 3/1 helix increase (please refer to their Extended Data Fig. 3 or Supplementary Information), but also the intensity of 1120 cm^{-1} dramatically increase. This phenomenon indicated that 1120 cm^{-1} is related to disorder structure. Since the introduction of gauche will cause chain distortion, both TTTG and TGTG can be considered to be more disordered structures comparing to TTTT arrangement. Moreover, theoretical analysis has revealed that the energy barrier between TTTG and long T sequence is smaller than that of all other conformations (ref 3). Therefore, it is a reasonable inference that the increase of 1120 cm^{-1} intensity in ref. 1 actually reflects the increase of TTTG conformation.

Kim et al.'s reports are not contradictory to our results either. They use factor analysis to reveal that 1120 cm^{-1} is related to disorder phase. (please refer to "Its intensity in the annealed sample spectrum decreased with increasing crystallinity because the intensity decrease for the disordered-phase band at 1122 cm^{-1} was much greater."). As discuss above, the disorder-phase means structures consisting of a statistical combination of TG and TTTG, and TTTG has a higher probability of presence.

It has been mentioned in many literatures that 1120 cm^{-1} is attributed to TTTG structure. Kohki et al. reported that 1125 cm^{-1} in FTIR spectrum was assigned to γ phase, which is TTTG structure (please refer to table 1) (ref. 4). J. L. Koenig et al reported that weak band including 1118 cm^{-1} and 1133 cm^{-1} are useful in characterizing phase III (γ phase with TTTG) (ref. 5).

Additional structural characterization are shown in Fig. R2. Although both face-on and edge-on lamellae show absorption peak at 1120 cm^{-1} , face-on lamellae have higher absorption intensity. In addition, several peaks attributed to TTTG can only be observed in face-on lamellae, including 833 cm^{-1} (ref. 6), 800 cm^{-1} and 772 cm^{-1} (ref. 3 and ref. 5). These results confirm that there are more TTTG structure in toroidal face-on lamellae. The results also show 614 cm^{-1} and 605 cm^{-1} peaks in face-on lamellae, which indicate the higher content of TG structure (ref. 2, 3).

Noteworthy, AFM-IR characterizes the local infrared response with a nanoscale resolution. Thus, it may enhance the signal of local disorder structure and yielding a different absorption intensity compared to macroscopic characterization method like FTIR.

Fig. R2. Reflectance FTIR spectrum comparison of face-on and edge-on P(VDF-TrFE). The detailed views of the spectral change are shown below. The red arrows show the trends of change of the characteristic peaks. The intensities of the characteristic bands corresponding to the TTTG conformation near 833 cm^{-1} (ref. 6), 800 cm^{-1} and 772 cm^{-1} (ref. 3, 5) was observed. And TG short sequence in 614/604 cm^{-1} double band was observed as well (ref. 2, 3).

Reference:

1. Liu, Y. et al. Ferroelectric polymers exhibiting behaviour reminiscent of a morphotropic phase boundary. *Nature* 562, 96-100 (2018).
2. Guo, M. et al. Flexible Robust and High-Density FeRAM from Array of Organic Ferroelectric Nano-Lamellae by Self-Assembly. *Adv. Sci.* 6, 1801931 (2019).
3. Zhu, Y. et al. Operando Investigation of the Molecular Origins of Dipole Switching in P(VDF-TrFE-CFE) Terpolymer for Large Adiabatic Temperature Change. *Adv Funct Materials* 2314705 (2024) doi:10.1002/adfm.202314705.
4. Uneda, K., Horike, S., Koshiba, Y. & Ishida, K. Dipole switching dynamics in P(VDF-TrFE) film revealed by in-situ polarization switching and infrared spectroscopy

measurements with high-time resolution. *Polymer* 249, 124822 (2022).

5. Bachmann, M. A., Gordon, W. L., Koenig, J. L. & Lando, J. B. An infrared study of phase-III poly(vinylidene fluoride). *Journal of Applied Physics* 50, 6106–6112 (1979).

6. Martins, P., Lopes, A. C. & Lanceros-Mendez, S. Electroactive phases of poly(vinylidene fluoride): Determination, processing and applications. *Progress in Polymer Science* 39, 683–706 (2014).

Revisions:

According to this comment, we have supplemented more details about the verification of TTTG structure. And we also added Fig. R2 to the Supplementary Materials as Supplementary Fig. 10.

Please refer to line 274-292 in manuscript highlighted in red and Supplementary Fig. 10 in Supplementary Materials.

“Comparing IR spectrum of edge-on and face-on P(VDF-TrFE) in Fig 3c, we found that 1120 cm^{-1} significantly distinguished from that of edge-on P(VDF-TrFE). It is mentioned in literatures that 1120 cm^{-1} is related to T_3GT_3G' (TTTG) conformations (*Polymer* 249, 124822, 2022; *Journal of Applied Physics* 50, 6106–6112, 1979). Here, Trans(T) and gauche(G) feature torsional bond angle (Ferroelectric Polymers. *Science* 220, (1983)). T represents arrangement with substituents at 180° to each other, and G/G' represents for that at $\pm 60^\circ$ respectively. Correspondingly, the absorption band characterizing TTTT at 1289 cm^{-1} and 1183 cm^{-1} diminish, indicating increase amount of TTTG in toroidal face-on P(VDF-TrFE). To further verify the structural conformation, we performed FTIR in both edge-on and face-on P(VDF-TrFE). As shown in Supplementary Fig. 10, several peaks assigned to TTTG have been observed, including 833 cm^{-1} (*Progress in Polymer Science* 39, 683–706, 2014), 800 cm^{-1} and 772 cm^{-1} (*Adv Funct Materials* 2314705, 2024 and *Journal of Applied Physics* 50, 6106–6112, 1979). The results also show double band of 614 cm^{-1} and 605 cm^{-1} peaks in face-on lamellae, which indicates the higher content of TG structure (*Adv Funct Materials* 2314705, 2024). The appearance of more gauche conformations was understandable. It has been reported that relaxor behavior was confirmed in toroidal face-on P(VDF-TrFE) (*Science* 371, 1050–1056, 2021). One of the reasons for the appearance of the relaxor properties in polymer is conformational disorder (*Nat. Mater.* **19**, 1169–1174, 2020) or defects (*Macromolecules* **50**, 9360–9372, 2017). Compared to TTTT, both TTTG and TGTG can be consider as disordered structure since the introduction of gauche will cause chain distortion. Thus, it induces relaxor behavior in toroidal face-on P(VDF-TrFE).”

Comment 2.3.1 of Reviewer #1: “If γ phase in face-on lamellae films is confirmed, some issues still need to be addressed. I am wondering about the underlying reasons for the formation of γ phase in face-on P(VDF-TrFE) lamellae.”

Response:

Thanks for the reviewer's comment. We suggest that the formation of TTTG conformation was induced by the biaxial tensile stress and electric field. The biaxial tensile strain, which is perpendicular to the chain axis, tends to distort the polymer chain. And electric field, which is parallel to the chain axis, promotes this process. It has been reported that TTTG has a smaller torsion energy than long T sequences (ref.1, please refer to Table S1). Thus, it's not a stable energy state to maintain TTTT conformation in this condition. The introduction of gauche can cause a distortion in polymer chain and reduced the energy (ref. 2). As a result, polymer chain undergoes trans-to-gauche transition and formed TTTG to fit with the tensile stress, even producing more gauche under high electric field. In other words, the biaxial strain reduced the energy barrier between trans and gauche. A similar mechanism of T-G transition under electric field was proposed by Zhu et al. recently (ref. 1), which supports our model.

References:

1. Zhu, Y. et al. Operando Investigation of the Molecular Origins of Dipole Switching in P(VDF-TrFE-CFE) Terpolymer for Large Adiabatic Temperature Change. *Adv Funct Materials* 2314705 (2024) doi:10.1002/adfm.202314705.
2. Su, H., Strachan, A. & Goddard, W. A. Density functional theory and molecular dynamics studies of the energetics and kinetics of electroactive polymers: PVDF and P(VDF-TrFE). *Phys. Rev. B* 70, 064101 (2004).

Revisions:

According to this comment, we have revised the description of the underlying reason.

Please refer to line 315-320 in manuscript highlighted in red.

“Biaxial strain in face-on P(VDF-TrFE) flattened the energy landscape of polarization states and it is crucial for the development of toroidal polar topology⁴⁴. Analogously, we believe such strain also reduced the energy barrier between T and G, promoting a trans-to-gauche transformation under electric bias.”

Comment 2.3.2 of Reviewer #1: “The polarization reversal in toroidal face-on P(VDF-TrFE) lamellae still could not exclude the -CF₂ rotation in γ phase. The IR absorption peak at 1289 cm⁻¹ is related to the -CF₂ symmetric stretching vibration of trans-planar phase with conformation T_m (m>4). Therefore, the different pattern of IR-E loop at 1289 cm⁻¹ (Fig. 3g) might exclude -CF₂ rotation in trans-planar phase, but cannot exclude the -CF₂ rotation in γ phase. This means that the polarization reversal mechanism may still contain the dipole rotations.”

Response:

Thanks for the reviewer's useful comment. We agree to the reviewer that it cannot exclude the rotation of -CF₂ in TTTG during polarization reversal. According to our investigation (ref. 1), the bending mode of TTTG is beyond our QCL laser wavenumber

limitation. Thus, we currently cannot confirm the $-CF_2$ rotation in TTTG. It is also true that our current results (hysteresis IR-E loops of 1120 cm^{-1}) support that conformation transition between TTTT and TTTG is involved in the polarization reversal of face-on lamellae. We will tone down the statement of the exclusion in the manuscript.

Reference:

1. Bachmann, M. A., Gordon, W. L., Koenig, J. L. & Lando, J. B. An infrared study of phase-III poly(vinylidene fluoride). *Journal of Applied Physics* 50, 6106–6112 (1979).

Revision:

According to this comment, we revised the statement of the proposed polarization mechanism to make it more acceptable.

Please refer to line 329-331 in manuscript highlighted in red.

“Even though, it still cannot rule out the possibility that CF_2 rotation from other conformations may contributed to the polarization reversal.”

Please refer to line 318-319 in manuscript highlighted in red.

“Based on foregoing observations, we postulated that the OOP polarization reversal is caused by an electric bias induced T-to-G conformational.”

Comment 2.3.3 of Reviewer #1: “If the out-of-plane polarization reversal is related to conformation transition as depicted in Fig. 3h, then what conformation might be transformed into? Are there any IR-E loops at certain IR peaks that exhibit reverse patterns compared to Fig. 3f? If yes, this may provide more hints on the polarization reversal mechanism.”

Response:

Thanks for the reviewer’s comment and suggestion. According to our model, the conformation might transform into structure containing more G. Firstly, polymer chain experiences T-to-G transition, producing TTTG. As the electric field increase, More T structure is converted to G, resulting TGTG’ or even 3/1 helical conformation $(TG)_3$. Thus, conformation of both TTTT and TTTG reduced. That’s why under high electric both 1289 cm^{-1} and 1120 cm^{-1} reduced as shown in Fig. 3.

No reverse patterns compared to Fig. 3f can be observed in our subsequent experiments. Perhaps the wavenumber was not included in our QCL laser. It may not exist any reverse patterns. The reason is that the IR-E loops of 1120 cm^{-1} in Fig 3f indicating structural transition. The changes in molecular orientation will produce opposite absorption variations at corresponding wavenumber. But under the transition model, the content of TTTT and other disorder structure seem to change monotonously. Thus, only the intermediate structure TTTG will experience non-monotonic change during the T-to-G transition and result butterfly pattern.

Comment 2.3.4 of Reviewer #1: “Are the on-field and off-field IR-E loops similar

under the circumstance of conformation transition? In other words, can the conformation generated under high electric fields be maintained after the removal of electric field? And why?"

Response:

Thanks a lot for the reviewer's question. The on-field and off-field IR-E loops are similar under the circumstance of conformational transition. Please refer to Fig. R3. However, we think the gauche structure generated by the electric fields will partially vanishes back to trans. And some TG units will remain and forming sequences like TTTG. As the bias decrease, both IR-E loops of 1289 cm^{-1} and 1120 cm^{-1} was back to the original point.

Fig. R3. On-field IR-E loop of 1120 cm^{-1} and 1289 cm^{-1} in toroidal P(VDF-TrFE). (a) Off-field IR-E loop of 1120 cm^{-1} with obvious butterfly pattern. (b) Off-field IR-E loop of 1289 cm^{-1} . The gray dot corresponds to 64 sampling data points of cantilever amplitude at each on-field stage. Red solid line corresponds to fitting curve according to average value of sampling points.

Revisions:

According to this comment, we have added Fig. R3 to the Supplementary Materials. Please refer to Supplementary Fig. 11 in Supplementary Materials. And please refer to line 305-306 in manuscript highlighted in red.

“The on-field IR-E loops result was shown in Supplementary Fig. 11.”

Comment 2.4 of Reviewer #1: “Could it be possible to conduct on-field AFM-IR spectrums? If it is possible, it can provide direct evidence on whether the absorption peak shifts during the measurement. Besides, Supplementary Fig. 5 reported that the butterfly-shaped IR-E loops maintained near the IR peak around 1183 cm^{-1} . However, the shape of IR-E loops at 1183 cm^{-1} and 1289 cm^{-1} are reversed. I am wondering if there is a critical IR wavenumber that is related to the shape reversal? If yes, is there any implication? It is suggested to provide more IR-E loops between 1183 cm^{-1} and

1289 cm^{-1} .”

Response:

Thanks for the reviewer’s comment and suggestion. The on-field AFM-IR spectrums was conducted in edge-on P(VDF-TrFE) and shown in Fig. R4. There has no clearly peak shifts under different bias.

Fig. R4. On-field AFM-IR spectrums at edge-on P(VDF-TrFE). The AFM-IR spectrum was obtained without notarization, which show the variation of IR abortion at 0V, 10V, 20V and 30V electric filed. And there has no clearly peak shifts under different bias.

We have not found a certain critical wavenumber which seems not to be in a fixed position. It will change refer to sample and measuring condition. We suggested that the critical wavenumber was dependent by the peak width of 1183 cm^{-1} and 1289 cm^{-1} . Perhaps it can be further investigated what the critical means about and what can it used for.

Revisions:

According to this comment, we added Fig. R4 to the Supplementary Materials as Supplementary Fig. 8.

Please refer to line 238-239 in manuscript highlighted in red.

“And the exclusion of artifact rising from shifting of absorption peak under electric bias was conducted and shown in Supplementary Fig. 7 and Supplementary Fig. 8.”

Comment 2.5 of Reviewer #1: “Could the authors compare the probing depth of AFM-IR and the effective depth of electric fields implemented by tips? In other words, does ePTFM suitable for ferroelectric films with a wide range of thickness?”

Response:

Thanks for the reviewer's comment. The probing depth of ePTFM is the same as AFM-IR, which is suitable for a wide range of thickness from several nanometers to micrometers. The probing depth of AFM-IR is determined by experiment setting. By setting different laser power, the depth sensitivity can be altered. Infrared radiation can penetrate several micrometers for organic sample. Thus, the probing depth can vary from several nanometers to micrometer under appropriate experiment settings (ref. 1). The effective depth of electric field also can cover from nanometer to micrometer, which is similar to piezoresponse force microscopy.

Reference:

1. Schwartz, J. J., Jakob, D. S. & Centrone, A. A guide to nanoscale IR spectroscopy: resonance enhanced transduction in contact and tapping mode AFM-IR. *Chem. Soc. Rev.* 51, 5248–5267 (2022).

Revision:

A description related to the probing depth of AFM-IR has been added to the manuscript. *Please refer to line 420-422 in the manuscript highlighted in red.*

“Owing to the penetration of IR radiation, the ePTFM is also suitable for films with a wide range of thickness from several nanometers to micrometers.”

Comment 2.6 of Reviewer #1: “The variation of IR absorption is quite significant in Supplementary Fig. 3 (0.020~0.028). What is the possible reason? Such relative variation (0.020~0.028) is comparable to the relative variation at 1289 cm^{-1} (0.00555~0.00655, off-field, Fig. 2e) and 1183 cm^{-1} (0.000375~0.000450, off-field, Fig. 2f). Does it have an adverse effect on the IR-E loop measurement, making it difficult to analysis the real signal of polarization reversal?”

Response:

Thanks for the reviewer's comment. In fact, IR-E loops in 1289 cm^{-1} and 1183 cm^{-1} can also exhibit high relative variation, as shown in Fig. R5. Similar to many AFM technologies such as PFM, the difference in absolute signal strength does not affect the observation of polarization reversal, and the relative IR absorption variation is our concern. As the probe moving, retuning and optimizing, the testing parameters are changed between each testing point. Especially, the difference in matching of resonance frequency and spot position would result in different IR signal strength.

Fig. R5. IR-E loops of 1289 cm^{-1} and 1183 cm^{-1} with on and off-field. (a) IR-E loops of 1289 cm^{-1} with on and off-field. (b) IR-E loop of 1183 cm^{-1} with on and off-field. The gray dot corresponds to 64 sampling data points of cantilever amplitude at each on/off-field stage. Red solid line corresponds to fitting curve according to average value of sampling points.

Comment 2.7 of Reviewer #1: “Could it be possible to conduct IR-E loop measurements with in-plane IR electric field? If yes, it may provide more evidence for the polarization reversal process, and may show more advantages than the Switching Spectroscopy PFM measurement.”

Response:

Thanks for the insightful comment. It is possible to conduct IR-E loops with in-plane IR electric field. As shown in Fig. R6, when the IR electric field tune to in-plane direction, the 1289 cm^{-1} IR-E loops will exhibit an upside-down butterfly pattern. The result shows difference compared to out-of-plane results in Fig. 2e, which is in consistent with the ferroelectric switching mechanism of edge-on sample. With the use of in-plane IR, the analysis of dipole behavior has escalated to a three-dimensional level. It contains a lot more information than just using out-of-plane IR.

Fig. R6. In-plane IR-E loops of 1289 cm^{-1} . The gray dot corresponds to 64 sampling data points of cantilever amplitude at each off-field stage. Red solid line corresponds

to fitting curve according to average value of sampling points.

Revision:

According to this comment, we have supplemented more details about the in-plane IR-E loops.

Please refer to line 425-426 in the manuscript highlighted in red.

“With the combination of in-plane IR electric field, it is able to escalate the analysis of dipole behavior to a three-dimensional level.”

3. Minor issues

Comment 3.1.1 of Reviewer #1: “There are several references that have not been properly cited. It is suggested to further examine this issue. Details are listed below.

Lines 174-175: This sentence mainly talks about the IR absorption peak of P(VDF-TrFE), while ref. 46 is about Nylon random copolymers and terpolymers.”

Response:

Thanks for the detail comment and suggestion. It has a mistake in the citation here.

Revisions:

It had been corrected in the main text.

Comment 3.1.2 of Reviewer #1: “Lines 235-237: The crystal structure of γ phase is shown in several articles^{7,8}. How to understand that it has more disordered T₃GT₃G chain conformation? Besides, it seems that ref. 47 did not discuss about whether the chain conformation of γ phase is disordered, as ‘disordered’ only appeared once.”

Response:

Thanks for the detail comment and suggestion. The statement in lines 235-237 may have misguidance in understanding. The disordered TTTG conformations was compared to the all-trans phase with conformation TTTT. Because the participation of gauche structure will alter the TTTT conformations to a more distortion chain arrangement sequence (ref. 1). The ref. 47 was aiming to show the structure of TTTG and it did not discuss about disorder conformation. According to this comment, we have revised the description and supplemented related reference in this sentence.

References:

1. Lovinger, A. J. Ferroelectric Polymers. Science 220, (1983).

Revisions:

Please refer to line 289-291 in the manuscript highlighted in red.

“Compared to TTTT arrangement, both TTTG is more disordered since the introduction of gauche will cause chain distortion (Lovinger, A. J. Ferroelectric Polymers. Science 220, 1983)”. Thus induce relaxor behavior in toroidal face-on P(VDF-TrFE).

Comment 3.1.3-3.1.5 of Reviewer #1: “Lines 259-260: This sentence is about the contribution of tensile strain to toroidal polar topology which was reported in ref. 40 instead of ref. 47.

Lines 267-269: This sentence is about the drop of amplitude in Switching Spectroscopy PFM measurement which was reported in ref. 40 instead of ref. 47.

Lines 297-299: This sentence is about the periodical absorption of toroidal domain which is in accordance with ref. 40 instead of ref. 47.”

Response:

Thanks for the detail comment and suggestion again. We apologize for mistake in the citation here.

Revisions:

All the reference had been corrected in the manuscript.

Comment 3.2 of Reviewer #1: “Lines 191-192: It is also important to point out that the transition dipole moment at 1183 cm^{-1} is perpendicular to polymer chains.”

Response:

Thanks for reviewer’s suggestion. It is very helpful for us to improve the readability of our manuscript.

Revisions:

According to this comment, we have supplemented the direction of transition dipole moment.

Please refer to line 229-230 in the manuscript highlighted in red.

“Its transition dipole moment is orthogonal to that of 1289 cm^{-1} and is perpendicular to polymer chains.”

Comment 3.3 of Reviewer #1: “Lines: 270-271: The pattern of Fig. 3g is still butterfly-shaped, and thus it might be inaccurate to state that butterfly pattern is not obvious. It is suggested to report the difference in coercive field instead.”

Response:

Thanks for reviewer’s suggestion. It is a very helpful suggestion and the reviewer’s opinion is very much appreciated. It has been revised in a more suitable description.

Revisions:

Please refer to lines 327-331 in the manuscript highlighted in red.

“The butterfly pattern become indistinct and the coercive electric field was different compared to PFM loops, which indicates $-\text{CF}_2$ rotation of all-trans phase no longer contributed to OOP polarization reversal.”

Comment 3.4 of Reviewer #1: “Figure 2b: Figure 2b and its caption is not clear enough

to readers. It is suggested to include IR electric field direction in Figure 2b, and the meaning of arrows with difference colors in the caption of Figure 2b.”

Response:

Thanks for reviewer’s suggestion. It’s very useful for improving clarity of the manuscript. We have revised the caption of Fig. 2 and included the IR electric field direction in Fig. 2b.

Revisions:

Please refer to Fig. 2 in the manuscript highlighted in red.

“Fig. 2. Electro-driven evolution of -CF₂ in ferroelectric edge-on P(VDF-TrFE) lamellae. ...**(b)** Schematic of -CF₂ molecular rotation during ferroelectric polarization switching. **The red arrow means transition dipole of -CF₂ symmetric stretching vibration (ν_s(CF₂)). The blue arrow means transition dipole of -CF₂ asymmetric stretching vibration (ν_{as}(CF₂)).** The mechanism of ferroelectricity in edge-on P(VDF-TrFE) was originated from the rotation of -CF₂ about the chain axis. The IR electric field was at vertical direction.”

Comment 3.5 of Reviewer #1: “Figure 3f, g: It is suggested to point out whether the IR-E loop is on-field or off-field in the caption even if they may be similar in pattern.”

Response:

Thanks for reviewer’s suggestion. It’s very useful for improving clarity of the manuscript. We have revised Fig. 3f, g and pointed out the off-field in the caption.

Revisions:

Please refer to Fig. 3, line 304-305 in the manuscript highlighted in red.

“(f) **Off-field** IR-E loop of 1120 cm⁻¹ with obvious butterfly pattern. (g) **Off-field** IR-E loop at 1289 cm⁻¹.”

Comment 3.6 of Reviewer #1: “Figures 4: It seems that electric-induced conformation evolution behaviors in different regions are different. If yes, it is suggested to demonstrate such different behaviors to better show the advantages of such nanoscale

characterization over the FTIR measurements”

Response:

Thanks for reviewer’s suggestion. It is a very good point for us. The local 1120cm^{-1} IR response under each voltage level in Fig. 4 are extracted and shown in Fig. R7. The results indeed demonstrate the advantages of ePTFM over the FTIR measurements. It seems that the attenuation occurs in dark region earlier. Based on current results, it needs further investigation and combination with other experiments to ensure that. Considering that the manuscript was aiming at proposing an approach, we suggest to discussed as an outlook in the manuscript.

Fig. R7. Local ePTFM amplitude evolution with electric bias. The scale bar was $0.5\mu\text{m}$.

Revisions:

The Fig. R7 includes the local 1120cm^{-1} IR response under each voltage level has been added as Fig. 4d in the manuscript.

Please refer to line 391-392 in the manuscript.

“The unique evolution characteristics of the ePTFM signal at the nanoscale have been extracted and displayed clearly in Fig. 4d, providing insights into its distinctive behavior.”

Response to the comments of reviewer #2

In “Electric Modulated Photothermal Force Microscopy for Revealing Chemical Origins of Polarization Switching “at Nanoscale Yao and coworkers merge two AFM based techniques PFM (piezoresponse force microscopy) and AFM-IR (an infrared nano spectroscopic method). The resulting method, named by the authors electric modulated photothermal force microscopy (ePTFM), leverages both infrared optical pulses (as in AFM-IR) and electrical bias (as in PFM). In this work the authors use ePTFM to characterize electro-driven ferroelectric switching in edge-on poly(vinylidene fluoride-ran-trifluoroethylene [P(VDF-TrFE)] lamellae and in toroidal face-on P(VDF-TrFE).

The proposed technique is in principle quite interesting. However, the manuscript will benefit from restructuring some of the technical information and from additional technical details. In the current form, the manuscript is a bit hard to read and of difficult assessment since the technical details are spread between main text, experimental and SI, requiring much back and forth by the reader. Since the technique has not been reported before, the bulk of technical details shall be presented and explained in the main text. In summary, I find the work interesting, sufficiently novel, and it could become publishable, provided the authors can make major changes to address the comments below:

Response:

We thank the reviewer’s recognition of the novelty and interest of our work. We also appreciate the reviewer’s valuable comments and suggestions on the manuscript. We have reconstructed the paragraphs to concentrate and present technical details in main text, including moving the technique detail from supplementary materials to the main text. We will respond to these comments one by one below. The comments, responses and revisions are highlighted in blue, black and red respectively.

Comment 1 of Reviewer #2: “There are many timescales, frequencies and polarization directions (i.e. of the sample, of the IR light, of the applied bias) contributing to the measurement, but not all details are specified. Is the AFM tip gold coated? Or what is it made off? What is the repetition rate of the IR laser in the context of the measurement? Is the IR laser matching one of the cantilever mechanical resonances (like in resonance enhanced AFM-IR) or not? If it is matching, how is the matching condition ensured? It would be useful to expand and define the timescale of Fig 1b to see how the IR signal changes within the 50 ms intervals of the applied bias. The authors should specify the direction of the linearly polarized IR light. In the NanoIR-2 tool used by the authors the IR polarization is not typically controlled unless an add on polarization controller is used. Was the polarization control module installed and used? If so, was the light

polarization s- or p- polarized? (Given the side illumination, if s-polarization is used the light pulses are linearly polarized in the sample plane, i.e. parallel to the substrate, but if p-polarization is used the light polarization has both in plane and out of plane components with respect to the substrate/sample plane. If the polarization control module is not used, then the light polarization at the sample is typically not controlled nor known). Please clarify.”

Response:

Thank for the reviewer’s comments and suggestions. It is very useful for us to improve the clarity of our manuscript. The detail of our measurement has been supplemented in the methods. The AFM tip is platinum coated and made of silicon. The repetition rates of IR laser is matching the second resonance frequency of cantilever. The matching condition is ensured by the phase-lock-loop.

The suggestion of Fig. 1b is very useful. Fig. R8 shows the expand and define timescale of Fig 1b. The IR polarization controller is used. And p-polarization IR light is used.

Fig. R8. Detail of time-dependent variation of amplitude and electric bias.

Revisions:

According to these comments, relative correction and experiments details have been made. We have added the measurement detail in the method. Meanwhile, Fig. R8 has been added as Supplementary Fig. 1 to show the detail of Fig 1b.

Please refer to line 441 in the manuscript highlighted in red.

“Polarization control module was installed and light polarization was p-polarized.”

Please refer to line 466-477 in the manuscript highlighted in red.

“The PFM loops and IR-E loops was all conducted with ePTFM platform with an Pt coating silicon AFM probe (ContE-G, BudgetSensors).”

Please refer to line 469-472 in the manuscript highlighted in red.

“During PFM loops measurement, the tip was modulated at ~60 kHz, with drive AC

voltage set at 3V. During IR-E loops measurement, repetition rates of IR laser were matching the second resonance frequency of cantilever. The matching condition was ensured by the phase-lock-loop.”

Please refer to line 497-499 in the manuscript highlighted in red.

“Repetition rates of IR laser were matching the second resonance frequency of cantilever. The matching condition was ensured by the phase-lock-loop.”

Comment 2 of Reviewer #2: “The main claim that ePTFM, adds chemical specificity to PFM is spectroscopically unconvincing and thus far unsubstantiated since all ePTFM experiments are obtained looking at the signal intensity at fixed wavelength, which is not sufficient proof. In fact, several effects unrelated to composition or orientation can/could influence the signal intensity in AFM-IR detection such as the sample mechanical properties or the applied bias. To claim chemical specificity and determination of the sample electrical polarization direction the authors shall spectroscopic evidence plotting the evolution of the IR spectra as a function of the applied bias, not just at selected wavelengths.”

Response:

Thanks for the reviewer’s comment and suggestions. The IR spectrum under 0 V, 10 V, 20 V, and 30 V bias have been conducted and shown in Fig. R9. The results show that there is no obvious peak shift under different bias. To confirm that the IR-E loops variations is mainly contributed from orientation or composition, we have presented several experiment results in the manuscript (please refer to the manuscript line 222-225 and Supplementary Fig. 7). The mechanical properties of sample might cause the shift of resonance frequency and result artifact of cantilever amplitude variations. Thus, we used phase-lock-loop for tracking the resonance frequency to reduce such influence. Meanwhile, the different pattern of Fig. 2e and Fig. 2f also indicate that unrelated effects have been suppressed. Otherwise, it cannot result a reverse pattern in the case of only changing the wavenumber.

In practice, there are several advantages for conducting ePTFM at fixed wavenumber. By doing this, the measure time as well as damage from long-term bias and IR pulsed can be significantly reduced. Meanwhile, unnecessary information from other wavenumber that has no significance on the experiments is excluded, which improves the observation of IR variation. For example, the IR-E loops in our experiments contain 200 bias stages. To gather the same quantity of information, 200 IR spectra must be collected, which may cause irreversible damage to the test sample point. Therefore, we believe the IR-E loop is suitable for revealing the evolution of composition and orientation with a fixed wavenumber.

Fig. R9. On-field AFM-IR spectrums at edge-on P(VDF-TrFE). The AFM-IR spectrum was obtained without notarization, which show the variation of IR abortion at 0V, 10V, 20V and 30V electric filed. And there has no clearly peak shifts under different bias.

Revisions:

According to this comment, Fig. R9 has been added as Supplementary Fig. 8.

Comment 3 of Reviewer #2: “For face-on P(VDF-TrFE) with toroidal domain is nicely characterized by PFM (Fig. 3b) which show the hierarchical structure of the toroidal pattern over an area of several micrometers wide, instead the ePTFM maps in Fig 4 are shown over a much smaller area. For the ePTFM study to be convincing the authors should provide at least 2 maps for vibrational modes with orthogonal dipole moment at the same scale of Fig 3b and corroborate the domain assignment with local IR spectra. In this context, the data in Fig 4 are not convincing and appear to be dominated by noise or by properties extrinsic to the sample electrical polarization, perhaps due to mechanical effects.”

Response:

Thanks for your appreciation in Fig. 3b and thanks for the helpful comments and suggestions. We have conducted a larger sale of ePTFM mapping, as shown in Fig. R10. The two maps of orthogonal dipole moment confirm that there exists hierarchical structure related to the toroidal domain. The local IR spectra result was also shown in Fig. R10, which shown that the variation was not dominated by noise or mechanical effects. The results of ePTFM show different information compared to PFM. It is aimed to reveal the chemical distribution, while PFM results shows the in-plane polar domain

structure. Thus, ePTFM maps may not completely as same as in-plane PFM results.

Fig. R10. The ePTFM maps of toroidal domain face-on P(VDF-TrFE). **a** the topography of face-on P(VDF-TrFE). **b** The ePTFM mapping of 1289 cm^{-1} absorption. **c** The ePTFM mapping of 1183 cm^{-1} absorption. **d** comparison of local IR spectrum of different point marked in **e**, which show that the distribution of IR mapping was not owing to the noise or mechanical effect. **e** Zoom in 1289 cm^{-1} IR mapping of **b**. **f** Zoom in 1183 cm^{-1} IR mapping of **c**

Revisions:

According to this comment, Fig. R10 showing IR mapping for vibrational modes with orthogonal dipole moment has been added to the supplementary materials as Supplementary Fig. 12.

Please refer to line 363-365 in the manuscript highlighted in red.

“Correspondingly, two maps for vibrational modes with orthogonal dipole moment and local IR spectra were conducted (Supplementary Fig. 12) confirming the toroidal domain.”

Comment 4 of Reviewer #2: “A very similar infrared photocurrent nano-spectroscopy technique operating at zero bias and providing nanoscale spectra (instead of just single

wavelength measurements) was reported recently using the same AFM platform (see Venanzi et al. Appl. Phys. Lett. 123, 153509 (2023)) and should be cited. The same work (page 3) also reports that “the signal to noise ratio at finite DC voltage bias (i.e. 40mV) between tip and substrate is not as good since it is dominated by the mechanical noise”. The ePTFM data reported by Yao et al. (Fig 2, 3 and 4) at much larger AC voltages (i.e. -35V/+35 V in Fig 2) appear to be noisy (perhaps to the point of becoming untrustworthy). Can the authors comment how the applied bias affects the noise in their measurements?”

Response: Thanks for the comments and suggestions. We have read thoroughly about this work (Venanzi et al. Appl. Phys. Lett. 123, 153509 (2023)). And a verification has been conducted, as shown in Fig. R11, the noise of the demodulated cantilever amplitude with and without applied electric field are monitored. No obvious noise increase can be observed under different bias. We would like to point out that the physical quantity measured in our technique is different with the reference’s approach, which results in the different extent of bias-induced noise. The reference use current as signal, which is more easily impacted by changes in the contact state between tip and sample. In our work, the cantilever’s mechanical vibration is measured, which is more robust under bias. The reference is very useful and has been cited in revised manuscript.

Fig. R11. The ePTFM signal before and after applied 30V electric bias. No obvious noise increase can be observed under different bias.

Revision:

According to these comments, we have cited this work (Venanzi et al. Appl. Phys. Lett. 123, 153509 (2023)) as ref.32 in the main text. And Fig. R11 has been added in Supplementary Fig. 1.

Please refer to line 81-82 in the manuscript highlighted in red.

“Several notable attempts have been made for modifying AFM-IR to expand the performance and applications²⁹⁻³³”

Comment 5 of Reviewer #2: “Another electrically detected photothermal nanospectroscopy method (Katzenmeyer et al. Nanoscale, 2015,7, 17637-17641) reported several years ago should be cited as it is relevant in the context of the method developed in this work.”

Response:

Thanks for the comments and suggestions. The reference is useful for our research.

Revisions:

We have cited this paper as ref.33 in the main text.

Please refer to line 81-82 in the manuscript highlighted in red.

“Several notable attempts have been made for modifying AFM-IR to expand the performance and applications²⁹⁻³³”

Response to the comments of reviewer #3

Overviews

Yao et al. claim to report the simultaneous analysis of ferroelectric polarization and chemical specificity, using modified piezoresponse force microscopy on an organic ferroelectric. The underlying hypothesis—that understanding the chemistry is essential while altering the polarization—is not immediately clear to the general reader, nor is it explained why these aspects cannot be measured separately. Given that the novelty of in-situ analysis is pivotal to the paper, this needs to be articulated more clearly.

While some data sets appear robust, the manuscript is hindered by poor writing quality and structure, alongside numerous technical challenges and misleading sections, making it incredibly difficult to comprehend. The inaccessibility to a general audience, combined with misleading content and ambiguity regarding the real impact of the work, leads me to conclude that I cannot recommend its publication in Nature Communications.

Response:

We greatly thank the reviewer's constructive comments and suggestions on the manuscript. It is very helpful for enhancing readability of our manuscript to a general audience. We will respond to these comments one by one below. The comments are marked in blue. The responses are noted in black. The revisions are highlighted in red.

Firstly, please allow us to briefly re-introduce our work, which will further promote the understanding of our work. Our work is given with the novelty as follows.

1) New technique: electric modulated photothermal force microscopy

In this work, we develop an electric modulated photothermal microscopy (ePTFM) by integrating polarized infrared excitation, electric bias modulation and photothermal force analysis into an AFM system. The principles are based on photothermal induced resonance (PTIR) effect to monitor the polarization-dependent infrared absorption through the amplitude of cantilever tip. The technique simultaneously enables nanoscale resolution and chemical specificity in polarization characterization, which is demonstrated by the observation of hysteretic rotation of $-CF_2$ during ferroelectric switching in edge-on P(VDF-TrFE) lamellae. Compared to previous techniques like piezoresponse force microscopy (PFM), the ePTFM shows better robustness and less interference under electric bias.

2) New mechanism: conformational transition of chain direction ferroelectricity in toroidal face-on P(VDF-TrFE)

Uncovering the different polarization switching mechanism of high-performance organic materials is one of the most concerning targets to researchers from multidisciplinary fields including physics, chemistry, biology, materials science. In this work, we demonstrated that ePTFM can provide detail information about the

correlation between molecular structure and polarization switching in nanoscale, and we have given the experimental evidence of the polarization switching mechanism in the emerged toroidal face-on P(VDF-TrFE). Using ePTFM, the mechanism of ferroelectric switching in toroidal face-on P(VDF-TrFE) was revealed. We found the existence of T₃GT₃G' chain conformation in toroidal face-on P(VDF-TrFE), which is corresponding to γ phase. Further analysis of TTTG molecular structure infrared absorption during polarization reversal shows that the configurational change of trans and gauche under modulated bias are contributed to chain direction ferroelectric. Our work not only provides a method for revealing polar mechanisms in organics, but also paves the way for further design of high-performance polar organics.

General points:

Comment 1 of Reviewer #3: “The manuscript is challenging to read, primarily due to the frequent use of specialized terminology without proper introduction, such as the term “gauche defects”. For a broad readership, it is essential to introduce and explain such terms. Additionally, the poor structure of the paragraphs contributes to the difficulty in reading. This lack of clear organization makes it hard to distinguish between new findings and established knowledge.”

Response:

Thanks for the comments and suggestions. We have carefully revised the manuscript to improve the language and have introduced the detail of specialized terminology in the main text. We also reconstructed the paragraphs to concentrate and present technical details in main text, including moving the technique detail from the supplementary materials to the main text. Trans and gauche means torsional bond angle arrangement having substituents at 180° (trans) to each other or at $\pm 60^\circ$ (gauche) (ref. 1).

Reference:

1. Lovinger, A. J. Ferroelectric Polymers. Science 220, (1983).

Revisions:

1) According to this comment, revision of specialized terminology is made as below.

Please refer to line 199-202 in the manuscript highlighted in red:

“The ePTFM signal can be obtained during the DC pulse by superimposing the measuring pulsed infrared radiation, or, alternately, immediately after each DC pulse. These two situations are named “on-field” and “off field” respectively (Applied Physics Letters 88, 062908 (2006)).”

Please refer to line 276-279 in the manuscript highlighted in red:

“Here, Trans(T) and gauche(G) feature torsional bond angle (Ferroelectric Polymers. Science 220 (1983)). Trans represents arrangement with substituents at 180° to each other, and gauche represents for that at $\pm 60^\circ$.”

2) The rearrangements of the structure in manuscript have been made including rearrange the description of technology detail from Supplementary Materials to main text.

Please refer to line 108-117 in the manuscript highlighted in red:

“Moreover, a tuning circuit was established to handle the probe’s response and control pulse frequency through OSC (oscillation), as shown in the top of Fig. 1a. It first divides the probe's deflection into two equivalent parts and multiplies them with their reference signals (produced by OSC and its 90-degree phase-shifted) respectively in Mixer A and Mixer B. After passing through low-pass filters, the results yield the in-phase component X and the quadrature component Y, which are then used to calculate the amplitude $R = \sqrt{X^2 + Y^2}$ and phase $\theta = \tan^{-1}(Y/X)$. Phase θ is fed to a phase lock loop (PLL) to ensure IR pulse frequency tracking to the AFM probe-sample contact resonance frequency. The amplitude was synchronously recorded with modulated bias using a multichannel data acquisition (DAQ).”

Please refer to line 142-153 in the manuscript highlighted in red:

“The photothermal expansion μ_0 of sample under pulsed IR can be given as (Journal of Applied Physics 2022, 131 (1), 010901)

$$\mu_0(\sigma, \theta) = G\alpha \frac{I_{\text{inc}}}{\rho C_p} t_p I_{\text{abs}}(\sigma, \theta), \quad (1)$$

where G contains all optical and geometrical parameters of the sample, α is the thermal expansion coefficient, I_{inc} is the incident laser energy, ρ is the sample density, C_p is the sample heat capacity, t_p is the duration of the laser pulse, $I_{\text{abs}}(\sigma, \theta)$ means the IR absorption coefficient, σ is the wavenumber and θ is the relative angle between the direction of IR electric field and transition dipole moment. Since the cantilever amplitude is directly proportional to the magnitude of thermal expansion, the cantilever amplitude is also related to the angle θ . Especially, in case no chemical reaction occurs, cantilever amplitude indicates the orientation of specific polar structure.”

Comment 2 of Reviewer #3: “‘However, the lack of electric modulation makes it still difficult to correlate the polarization behavior with the evolution of specific molecular structure in nanoscale.’

The authors emphasize the importance of chemical specificity and polarization, but its significance may not be immediately clear to a general audience. One might question, for instance, whether the chemical composition could be readily mapped using techniques such as EDX (Energy Dispersive X-ray Spectroscopy). Additionally, the manuscript should address whether the authors believe that the chemistry changes with ferroelectric switching. If so, it is pertinent to question whether the material still conforms to the definition of a ferroelectric, traditionally characterized as a material

exhibiting switchable net remanent polarization arising from a symmetry-lowering phase transition.”

Response:

We really appreciate the comment and suggestion. Our approach is designed to monitor the movement or deformation of specific chemical structure during polarization reversal, such as torsion and rotation of certain monomer or chemical group. By integrating chemistry with polarization, it is able to observe the specific evolution of chemical structures during polarization switching, thereby revealing the underlying chemical origins of this phenomenon. In fact, our work reveals via ePTFM that ferroelectric switching in two lamella involves different movement of chemical group. The chemical changes are neither expected nor observed in our experiments. As far as we know, EDX is a technique for elementary analysis, which should be irrelevant to our issue. To make our point clearer, we have revised our manuscript to further emphasize the significance of chemical specificity and polarization and make it more clear to general audience.

Revisions:

To the importance of chemical specificity and polarization, the relative discussions have been written in the revised manuscript.

Please refer to line 57-60 in the manuscript highlighted in red:

“By integrating chemistry with polarization, it is able to observe the specific evolution of chemical structures during polarization switching, thereby revealing the underlying chemical origins of this phenomenon. This integration is crucial for understanding the fundamental mechanisms that drive polarization switching.”

Please refer to line 70 in the manuscript highlighted in red:

“Nevertheless, the chemical specificity and its correlations with polarization, i.e., the movement or deformation of specific chemical structure during polarization reversal, are still missing.”

Comment 3 of Reviewer #3: “For other approaches, transmission electron microscopy (TEM) may be able to achieve nanoscale chemical and polar analyses, whereas it may cause radiation damage for organics.’

I believe this is readily solved by using cryo TEM. This would also give much higher resolution imaging.”

Response:

Thanks for the comments. Cryo-TEM (transmission electron microscopy) freezes the sample in low temperature, which can protect the sample from radiation damage. However, for thin film samples preparation procedure for TEM measurement involves slicing and ion thinning (ref. 1), which still cause damage to the original structure of

the organic film. In addition, it is more complicated for Cryo-TEM to applied external bias in compare with scanning probe technique. Both techniques have its own advantages. We have adopted a more accurate description to explain our opinion in revised manuscript.

References:

1. Yang, Z. et al. Electrospray-assisted cryo-EM sample preparation to mitigate interfacial effects. Nat Methods (2024) doi:10.1038/s41592-024-02247-0.

Revisions:

Please refer to line 76-78 in the manuscript highlighted in red:

“Even with Cryo-TEM, the preparation procedure for thin film samples involves slicing and ion thinning, which would cause damage to the original structure of the organic film.”

Comment 4 of Reviewer #3: “The manuscript does not clearly articulate the extent of information this approach can provide without preexisting knowledge. For instance, in Figure 1(c), while explaining the technique, the authors mention a significant response when the dipole is aligned with the AFM (Atomic Force Microscopy) tip. Although this is similar to vertical PFM (Piezoresponse Force Microscopy), as discussed in specific points below, it remains unclear whether this technique would yield new insights on an unknown sample.”

Response:

Thanks a lot for reviewer’s comments. PFM utilize local electromechanical response characterize nanoscale polarization. However, the structural origin and mechanism of polarization evolution cannot be obtained directly via PFM. For example, ferroelectric-like hysteresis and piezoelectric domain have been observed in many biological tissues such as elastin (ref. 1), butterfly wings (ref. 2) and abalone shell (ref. 3). The authenticity and structural origin of these ferroelectric-like behaviors are still obscure (ref. 4,5). In order to conduct in-depth research, the information of movements or content variation of certain chemical structure during switching is indispensable.

And this is what ePTFM can achieve and provide without preexisting knowledge. As we have shown in Fig. 2e and 2f, normal and upside-down butterfly pattern are observed with orthogonal transition dipole moment respectively, which demonstrate the hysteretic rotation of $-CF_2$. Note that such results cannot be obtained via PFM. Moreover, diverse patterns of chemical structure movements have been identified between edge-on and toroidal face-on P(VDF-TrFE) via ePTFM. Thus, it is largely different to PFM. While PFM observes total polarization of the sample, ePTFM observes the chemical group’s dipole moment that constitutes the total polarization. And the observed chemical group can be selected by the IR wavenumber. PFM

hysteresis loops of edge-on and face-on P(VDF-TrFE) are roughly similar. However, their ePTFM IR-E loops show huge differences (please refer to Fig. 2 and Fig. 3). This is direct evidence that additional information is included in ePTFM, which also help to reveal the origin of ferroelectric switching in face-on P(VDF-TrFE).

The Fig. 1c may not be clear enough to present our idea. Thus, we have added an IR electric field direction in it. This modification emphasizes that the response of our technique depends on the alignment between selective molecular dipole and IR electric field direction.

References:

1. Liu, Y., Zhang, Y., Chow, M.-J., Chen, Q. N. & Li, J. Biological Ferroelectricity Uncovered in Aortic Walls by Piezoresponse Force Microscopy. *Phys. Rev. Lett.* 108, 078103 (2012).
2. Kalinin, S. V. et al. Bioelectromechanical imaging by scanning probe microscopy: Galvani's experiment at the nanoscale. *Ultramicroscopy* 106, 334–340 (2006).
3. Li, T. & Zeng, K. Nanoscale piezoelectric and ferroelectric behaviors of seashell by piezoresponse force microscopy. *Journal of Applied Physics* 113, 187202 (2013).
4. Vasudevan, R. K., Balke, N., Maksymovych, P., Jesse, S. & Kalinin, S. V. Ferroelectric or non-ferroelectric: Why so many materials exhibit “ferroelectricity” on the nanoscale. *Applied Physics Reviews* 4, 021302 (2017).
5. Sun, Y., Zeng, K. & Li, T. Piezo-/ferroelectric phenomena in biomaterials: A brief review of recent progress and perspectives. *Sci. China Phys. Mech. Astron.* 63, 278701 (2020).

Revisions:

Please refer to Fig. 1c in the manuscript highlight in red.

“Fig. 1c. Schematic showing the correlation of IR absorption(amplitude) and rotation of molecular structure under vertical polarized IR beam. The red and blue region shown the polarization of entire specie. The yellow marker indicated the molecular which is observed. It tracks the specific molecular dipolar moment by selecting IR wavenumber. If the selective transition dipolar moment parallel to IR electric field, the amplitude reaches the maximum

value. Oppositely, if it is perpendicular to IR electric field, the amplitude reaches the minimum value.”

Comment 5 of Reviewer #3: “I have a specific question regarding the response time of materials to thermal effects. How does this response time correlate with the scan speed and other related phenomena in ferroelectrics?”

Response:

Thanks for the reviewer’s comments. As shown in the reference (ref. 1, please refer to its Fig. 3c), the response time of thermal expansion ranging from several nanoseconds to hundreds of nanoseconds. The scan speed was 1Hz, which is 1 second per line. It is much longer than the thermal effect response time. And the time for polarization establishment in P(VDF-TrFE) is much shorter than thermal response time. Thus, there has enough time for the sample to stabilize its polarization state under bias and be detected by infrared radiation. It has been proved in other literature (ref. 2) that the thermal effects will not affect the polarization state as long as the laser power is appropriately set.

Reference:

1. Chae, J. et al. Nanophotonic Atomic Force Microscope Transducers Enable Chemical Composition and Thermal Conductivity Measurements at the Nanoscale. *Nano Lett.* 17, 5587–5594 (2017).
2. Guo, M. et al. Toroidal polar topology in strained ferroelectric polymer. *Science* 371, 1050–1056 (2021).

Specific points:

Comment 6 of Reviewer #3: ““In brief, our technique integrates optical radiation, modulated electric bias and photothermal effect into an AFM system to achieve chemically specific characterization of polarization evolution in nanoscale.”

The claim about the effectiveness of your approach is reiterated often, yet it remains unclear to me as a reviewer whether you have conclusively established this. To substantiate your claim, I suggest demonstrating PFM (Piezoresponse Force Microscopy) independently, alongside a separate chemically specific characterization. Following this, it would be necessary to establish that these two methods measure different aspects (as referenced in another question). Only then, by combining both approaches, could you convincingly demonstrate the generation of new information.”

Response:

Thanks for the reviewer’s comment and suggestion. Following the reviewer’s suggestion, we have carefully revised our manuscript and demonstrate PFM independently. And additional discussion has been added. The generation of new

information has been detailly discussed above. Please refer to Comment 4. The ePTFM has the ability to analyze a local chemical group with specificity, which cannot be achieve by the PFM.

Revisions:

To clarify our points and to improve readability, the relative discussion has been added including a demonstration of PFM independently in manuscript and supplementary notes.

Please refer to line 163-167 in the manuscript highlight in red.

“In comparison, PFM utilizes local electromechanical response to characterize nanoscale polarization. The origin and mechanism of polarization evolution cannot be obtained directly. It is able to track the specific chemical structure by selecting IR wavenumber during polarization switching, thereby revealing the underlying chemical origins.”

Comment 7 of Reviewer #3: “However, the lack of electric modulation makes it still difficult to corelate the polarization behavior with the evolution of specific molecular structure in nanoscale.’

Do the authors believe the structure changes during switching? Perhaps they mean reorientation of a molecule with the same structure?”

Response:

Thanks for the reviewer’s comments. Here, the reorientation of a molecular is included in the evolution of specific molecular structure. For example, in PVDF, -CF₂ will undergoes rotation about the chain axis during switching (ref. 1). Also, the evolution includes the generation or vanish of specific chemical structure. For example, there exists gauche to trans structure transformation in dipole switching of P(VDF-TrFE-CFE) (ref.2).

References:

1. Uneda, K., Horike, S., Koshiha, Y. & Ishida, K. Dipole switching dynamics in P(VDF-TrFE) film revealed by in-situ polarization switching and infrared spectroscopy measurements with high-time resolution. *Polymer* 249, 124822 (2022).
2. Zhu, Y. et al. Operando Investigation of the Molecular Origins of Dipole Switching in P(VDF-TrFE-CFE) Terpolymer for Large Adiabatic Temperature Change. *Adv Funct Materials* 2314705 (2024) doi:10.1002/adfm.202314705.

Comment 8 of Reviewer #3: “Figure 1(c), the smaller amplitude of tip oscillation when the polar moment is in-plane rather than out of plane: is the same information not included in the PFM? Dipole moments out of plane will show up in the vertical PFM

while dipole moments in plane will show in in plane PFM. where a parallel”

Response:

Thanks for the reviewer’s comments. The new generated information in ePTFM and the different between ePTFM and PFM has been discussed above. Please refer to comment 4 and 6. Compared to PFM, the dipole we observe in ePTFM was the chemical group’s transition dipole moment that selected by IR wavenumber. It is different to the PFM. And the response depends on the relative angles between selective molecular dipole and IR electric field direction.

Revisions:

As in Comment 4, please refer to Fig. 1c and its caption in the manuscript highlighted in red.

Comment 9 of Reviewer #3: “In Figure 1(c), you note a smaller amplitude of tip oscillation when the polar moment is in-plane, as opposed to out-of-plane. However, isn't this information already captured in Piezoresponse Force Microscopy (PFM)? Specifically, dipole moments oriented out-of-plane would be evident in vertical PFM, while in-plane dipole moments would be observable in lateral PFM.”

Response:

Thanks for the comments. The different between ePTFM and PFM has been discussed above. Please refer to comment 4, 6 and 8. To clarify our points, Fig. 1(c) have been revised as described in comment 4. The ePTFM observes the chemical group’s transition dipole moment, which is selected by IR wavenumber. And it is different to the total dipole moment that PFM observed. In such way, this technique can provide a new insight of how a specific chemical group or other chemical structure evaluated during the polarization reversal which is absent in PFM.

Revisions:

As in Comment 4, please refer to Fig. 1c and its caption in the manuscript highlighted in red.

Comment 10 of Reviewer #3: “The introduction is written for general cases, but the references focus on very specific studies. The authors would be encouraged to include some reviews or more general articles.”

Response:

Thanks for the reviewer’s constructive suggestion. We have reconsidered the reference and include some more general articles in introductions.

Revisions:

The following section have revised and included some reviews or more general articles:

Please refer to the reference of line 47-48 in the manuscript.

“Organic ferroelectrics have diverse chemical constituents and microstructures allowing a wide range controllability of polarization behavior and thus diverse functionalities¹⁻².” (Chem. Rev. 116, 4260–4317 (2016).) (Science 380, eadg0902 (2023).)

Please refer to the reference of line 50-53 in the manuscript.

“However, due to the complexity in structure and chemical constituents, it is still challenging to have sufficient understanding in the correlations between the polarization behavior and molecular structure, impeding further explorations in evaluating and tailor-making of desired physical functionalities¹¹⁻¹³” (ACS Nano 16, 17708–17728, 2022.) (Chem. Rev. 122, 14594–14678, 2022.)

Please refer to the reference of line 58-61 in the manuscript.

“Only when nanoscale polarization and chemical component analysis is integrated, the mechanism and controllability principles of organic ferroelectrics can be established with certainty, other than empiricism or speculation^{14,15}.” (Appl. Phys. Lett. 111, 133701, 2017.) (Sci. China Phys. Mech. Astron. 63, 278701, 2020.)

Please refer to the reference of line 62-64 in the manuscript.

“To understand the polarization evolution, lots of experimental techniques have been proposed¹⁶, in which the use of these techniques still has constraints in monitoring chemical components during polarization evolution in nanoscale.” (Physical Sciences Reviews 5, 20190067, 2020.)

Comment 11 of Reviewer #3: “Only when nanoscale polarization and chemical component analysis is integrated, the mechanism and controllability principles of organic ferroelectrics can be established with certainty, other than empiricism or speculation^{14,15}.”

Do the authors propose that organic ferroelectrics adhere to different physical principles compared to conventional ferroelectrics? If this is the case, I would appreciate a detailed justification, supported by relevant references. From my perspective as a referee, while the origin of the polar component in organic ferroelectrics might differ, the ferroelectric properties of that component are anticipated to be analogous to those in conventional materials. This similarity is one of the reasons Piezoresponse Force Microscopy (PFM) is applicable to organic ferroelectrics.”

Response:

Thanks for the reviewer’s comment and suggestion. This sentence is not going to proposed that organic ferroelectrics adhere to different physical principles compared to conventional ferroelectrics. The term “chemical component analysis” here refers to the

identification of movement or deformation of specific chemical structure during ferroelectric reversal, which reveals the chemical origin of the bistable dipole and is considered as a mechanism of organic ferroelectrics. The ferroelectric properties of organic materials are analogous to those in conventional materials. However, bistable dipole is more complicated in organics, which has significant impact on polarization evolution. Our work is aiming to reveal the origin of polarization evolution, just as the title of our manuscript presents.

Revisions:

Please refer to line 61-63 in the manuscript highlighted in red.

“Only when nanoscale polarization and chemical component analysis is integrated, the **origins** and controllability principles of organic ferroelectrics can be established with certainty, other than empiricism or speculation”

Comment 12 of Reviewer #3: “To understand the polarization evolution, lots of experimental techniques have been proposed, in which the use of these techniques still have constraints in monitoring chemical components during polarization evolution in nanoscale.”

Could the authors provide references supporting the proposed experimental techniques? Are these techniques comparable to those long-established for measuring ferroelectrics in inorganic materials, such as Piezoresponse Force Microscopy (PFM)? For conventional ferroelectrics, there are numerous recent publications that offer overviews of these techniques. An example can be found at <https://doi.org/10.1515/psr-2019-0067>, which could be a useful reference in this context.”

Response:

Thanks for the reviewer’s comment and suggestion. The references of the proposed experimental techniques have been mentioned in line 61-73, including Sawyer-Tower cuticle, FTIR et al. These techniques are all long-establish and comparable to Piezoresponse Force Microscopy (PFM). The reference of PFM provide by the reviewer was helpful and had been cited as ref.16.

Revisions:

Please refer to line 64 in manuscript highlighted in red.

“To understand the polarization evolution, lots of experimental techniques have been proposed (**Physical Sciences Reviews 5, 20190067, 2020**), in which the use of these techniques still have constraints in monitoring chemical components during polarization evolution in nanoscale.”

Comment 13 of Reviewer #3: “‘Its signal often includes the unwanted contributions of bias-induced polarization and electrostatic response caused by cantilever-sample capacitance, which is detailly described in Supplementary Note 3.’

This representation is highly misleading. While such effects may occur, they are not intrinsic to the technique. Proper operation by the user can effectively eliminate these issues. Please refer to the specific comments below for further details.”

Response:

Thanks for the reviewer’s comment. Piezoresponse Force Microscopy (PFM) is essentially an atomic microscopy-based technique that observe bias-induced response in the sample (ref. 1). As having been reported in many articles (please refer to ref. 2, part V. mechanisms and bias-induced phenomena in AFM), the response that contributed to PFM signal is not only just inverse piezoelectric response, but also additional contribution including electrostatic response, joule heating and Vegard strain. These additional contributions will plague target signal, giving a misleading picture of the ferroelectric of the sample. For example, polarity switching can be observed in unexpected samples like glass (ref. 3) or silicon (ref. 4). Espectically, when the sample has weak piezoelectric constant, the result will be more likely to be affected by artifacts.

Lots of literatures have discussed about the interference and artifacts in PFM (ref. 2,5,6,7). Most of the literatures focus on distinguishing or reducing these additional contributions. However, it is difficult to completely eliminate these issues, especially in the present of directed current bias (DC bias). Please refer to specific comment’s response below for further detail.

Reference:

1. Gruverman, A., Alexe, M. & Meier, D. Piezoresponse force microscopy and nanoferroic phenomena. *Nat Commun* 10, 1661 (2019).
2. Vasudevan, R. K., Balke, N., Maksymovych, P., Jesse, S. & Kalinin, S. V. Ferroelectric or non-ferroelectric: Why so many materials exhibit “ferroelectricity” on the nanoscale. *Applied Physics Reviews* 4, 021302 (2017).
3. Chen, Q. N., Ou, Y., Ma, F. & Li, J. Mechanisms of electromechanical coupling in strain based scanning probe microscopy. *Appl. Phys. Lett.* 104, 242907 (2014).
4. Sekhon, J. S., Aggarwal, L. & Sheet, G. Voltage induced local hysteretic phase switching in silicon. *Applied Physics Letters* 104, 162908 (2014).
5. Ming, W., Huang, B. & Li, J. Decoupling competing electromechanical mechanisms in dynamic atomic force microscopy. *Journal of the Mechanics and Physics of Solids* 159, 104758 (2022).

6. Balke, N. et al. Exploring Local Electrostatic Effects with Scanning Probe Microscopy: Implications for Piezoresponse Force Microscopy and Triboelectricity. ACS Nano 8, 10229–10236 (2014).
7. Kalinin, S. V., Jesse, S., Tselev, A., Baddorf, A. P. & Balke, N. The Role of Electrochemical Phenomena in Scanning Probe Microscopy of Ferroelectric Thin Films. ACS Nano 5, 5683–5691 (2011).

Revisions:

According to this comment we have added a discussed about the bias-induced phenomena in PFM.

Please refer to Supplementary Note 1, line 27-31 in the supplementary materials.

“PFM is essentially an atomic microscopy-based technique that observe bias-induced response in the sample. As have been reported, the response that contributed to PFM signal is not only just inverse piezoelectric response, but also additional contribution including electrostatic response, induced chemical dipole, charge injection. (Applied Physics Reviews 4, 021302,2017; Nanoscale Adv. 4, 2036-2045, 2022). These additional contributions will plague target signal, giving a misleading picture of the ferroelectric of the sample.”

Comment 14 of Reviewer #3: “SI, ‘As a result, in addition to remanent polarization, the on-field PFM amplitude also includes the contributions of polarization induced by DC bias ($P_{DC}=\chi\epsilon_0V_{DC}$) and electrostatic response caused by cantilever-sample capacitance. And because DC modulated bias is coupling 60 with AC probing bias, it is also unable to separate the contribution in frequency domain.’

I would like to request clarification on a specific point from the authors:

- 1) The statement regarding the use of PFM (Piezoresponse Force Microscopy) is somewhat ambiguous. Typically, PFM operates without a DC bias; for switching experiments, poling can be conducted (a DC voltage applied and removed) followed by scanning with an AC voltage and zero DC voltage. Could the authors elaborate on what they mean in this context?
- 2) Regarding the inability to separate contributions in the frequency domain: could the authors clarify this point? Variations in resonant frequency during scanning are precisely the basis of Dual AC Resonance Tracking (DART) mode PFM.”

Response:

Thanks for the reviewer’s comment.

- 1) As discussed above, PFM (Piezoresponse Force Microscopy) is essentially an atomic microscopy-based technique that observe bias-induced response in the sample.

Takes SS-PFM (Switching Spectroscopy-PFM) experiment as example (which is used to measure PFM hysteresis loops). The piezoresponse can be obtained during the DC pulse by superimposing the measuring AC sinusoid, or, alternately, immediately after each DC pulse. The DC voltage is varied in a series of voltage pulses with varying magnitude. These two situations are named “on-field” and “off field” (ref. 1). The purposed of off-field observables was to minimize electrostatic contributions bring by DC bias. And selecting off-field results for analysis is a widely used routine. That was because the effect brings by DC bias generate a bending response that cannot distinguish from AC bias induced response.

Although through the testing procedure designing, the current PFM tries to avoid relevant measurement in the present of DC voltage. However, right after removing the DC voltage, it may still result influence including injected charge and redistribution of ionic and electronic species, which will also construct a bias electric field to play a similar interference effect. Moreover, it is also of high interest to study the voltage-dependence of the bias-induced polar state, especially for volatility polar organics whose polar state cannot maintain after removing DC bias. However, with the present of DC bias, the intrinsic polar state will be plagued.

2) For a more detailed description of the DC bias-induced additional contributions, we consider that deflection of PFM contains surface displacement U and electrostatic force contribution U_{ES} (ref. 2):

$$D = U + U_{ES}, \quad (1)$$

where $U = QP^2$, Q represent the electrostrictive coefficient and P is the polarization. For a polar organics P may including various contribution, here considering remanent polarization $\pm P_r$ and induced one $\chi\epsilon_0 V$, where χ is polarizability, ϵ_0 is vacuum permittivity and $V = V_{DC} + V_{AC}\sin(\omega t)$ is electric bias, in which V_{DC} is the DC modulated bias applied to the tip, and V_{AC} is the amplitude of tip AC probing bias. Thus, polarization can be expressed by $P = \pm P_r + \chi\epsilon_0(V_{DC} + V_{AC}\sin(\omega t))$ (ref. 3). And the surface displacement U can be written as:

$$\begin{aligned} U &= Q(\pm P_r + \chi\epsilon_0 V)^2 \\ &= Q(P_r^2 \pm 2\chi\epsilon_0 P_r V + \chi^2 \epsilon_0^2 V^2) \\ &= Q(P_r^2 \pm 2\chi\epsilon_0 P_r V_{DC} \pm \chi\epsilon_0 P_r V_{AC} \sin(\omega t) \\ &\quad + \chi^2 \epsilon_0^2 V_{DC}^2 + 2\chi^2 \epsilon_0^2 V_{DC} V_{AC} \sin(\omega t) + \chi^2 \epsilon_0^2 V_{AC}^2 \sin^2(\omega t)), \quad (2) \end{aligned}$$

On the other hands, electrostatic force between tip/cantilever and sample is $F = \frac{C}{2}(V_{DC} - V_{SP} + V_{AC}\sin(\omega t))^2$, where C is capacitance gradient between tip/cantilever

and sample, V_{SP} is the surface potential (ref. 4, please refer to SI). The corresponding displacement contribution U_{ES} can be expressed by:

$$U_{ES} = k^{-1} \frac{C}{2} \left(\left[(V_{DC} - V_{SP})^2 + \frac{1}{2} V_{AC}^2 \right] + 2[V_{DC} - V_{SP}]V_{AC} \sin(\omega t) - \left[\frac{1}{2} V_{AC}^2 \cos(2\omega t) \right] \right), \quad (3)$$

where k is effective contact stiffness. After the processing of lock-in amplifier, all the constants and signal with frequency other than ω will be eliminated. Thus, the DC-modulated PFM amplitude reads:

$$D_{amp} = \sqrt{[\pm 2Q\chi\varepsilon_0 P_r + 2Q\chi^2\varepsilon_0^2 V_{DC} + k^{-1}C(V_{DC} - V_{SP})]^2 V_{AC}^2} \quad (4)$$

Because only high-order terms and constant terms can be filtered out, the first order term of V_{AC} including contribution of DC bias-induced polarization remains, as shown in the second term in the radical sign $2\chi^2\varepsilon_0^2 V_{DC}$. In addition, the electrostatic force contribution term $k^{-1}C(V_{DC} - V_{SP})$ also remains. Thus, the DC bias additional contributions cannot be separated by lock-in amplifier. Because these contributions were modulated by V_{AC} and share the same frequency of lock-in reference's signal.

Dual AC Resonance Tracking (DART) mode PFM is used for tracking the contact resonance frequency. It is not related to the point we demonstrated here.

Reference:

1. Jesse, S., Baddorf, A. P. & Kalinin, S. V. Switching spectroscopy piezoresponse force microscopy of ferroelectric materials. *Applied Physics Letters* 88, 062908 (2006).
2. Vasudevan, R. K., Balke, N., Maksymovych, P., Jesse, S. & Kalinin, S. V. Ferroelectric or non-ferroelectric: Why so many materials exhibit “ferroelectricity” on the nanoscale. *Applied Physics Reviews* 4, 021302 (2017).
3. Chen, Q. N., Ou, Y., Ma, F. & Li, J. Mechanisms of electromechanical coupling in strain based scanning probe microscopy. *Appl. Phys. Lett.* 104, 242907 (2014).
4. Kim, Y. et al. Nonlinear Phenomena in Multiferroic Nanocapacitors: Joule Heating and Electromechanical Effects. *ACS Nano* 5, 9104–9112 (2011).

Revisions:

According to this comment we have added a detail derivation of the response of PFM in the Supplementary Note 1, quasi the derivation of equation (1)-(4) shown above.

Comment 15 of Reviewer #3: “SI “Therefore, electrostatic response appears as artifact, and the influence of P_{DC} blurs the intrinsic information in PFM when analyzing polar evolution.’

Could you specify which electrostatic response is being referred to? Piezoresponse Force Microscopy (PFM) is widely used in inorganic materials, often characterized by

substantial polarization (e.g., PZT, BaTiO₃, LiNbO₃), without encountering significant issues. Please detail the specific problem you are addressing. Additionally, consider the recent pivotal work by Killgore et al., which effectively demonstrates a simple way to mitigate such effects when they are present [DOI: <https://doi.org/10.1039/D2NA00046F>].”

Response:

Thanks for the reviewer’s comment. The electrostatic response here is referred to the PFM (Piezoresponce force microscopy) signal that induced by electrostatic potential difference and the capacitive gradient between the cantilever and sample surfaces (ref. 1), which has been discussed in Comment 14 by Eq. (3)-(4). The influence of electrostatic response depends on the sample. Even in traditional inorganic ferroelectrics like PZT, BaTiO₃, if the film is thinner than 10nm, the influence of electrostatic response should be carefully handled (ref. 2,3,4) since the deformation caused by inverse piezoelectric effect is small. For other sample, such as bulk periodic polarization LiNbO₃ (PPLN), the electrostatic response can be strong so that one must apply tip bias to neutralize the effect. For organic sample, electrostatic response must be carefully identified and eliminated because charge injection and Vegard strain can be significant compared to its low piezoelectric constant. There are several ways to reduce the electrostatic response, including tip bias application, using a stiffer probe, interferometer displacement sensor (IDS) and so on (ref. 5). The effort on this issue proves the importance of handling electrostatic response. As for recent pivotal work by Killgore et al, it is helpful to reduced electrostatic effect in normal condition. And it has been cited as reference in the main text.

Reference:

1. Killgore, J. P., Robins, L. & Collins, L. Electrostatically-blind quantitative piezoresponce force microscopy free of distributed-force artifacts. *Nanoscale Adv.* 4, 2036-2045 (2022).
2. Cheema, S. S. et al. Enhanced ferroelectricity in ultrathin films grown directly on silicon. *Nature* 580, 478–482 (2020).
3. Kelley, K. P. et al. Thickness and strain dependence of piezoelectric coefficient in BaTiO₃ thin films. *Phys. Rev. Materials* 4, 024407 (2020).
4. Kang, S. et al. Highly enhanced ferroelectricity in HfO₂-based ferroelectric thin film by light ion bombardment. *Science* 376, 731–738 (2022).
5. Vasudevan, R. K., Balke, N., Maksymovych, P., Jesse, S. & Kalinin, S. V. Ferroelectric or non-ferroelectric: Why so many materials exhibit “ferroelectricity” on the nanoscale. *Applied Physics Reviews* 4, 021302 (2017).

Revisions:

According to this comment, the discussion about electrostatic response and the influence of DC had been added in Supplementary Note 1.

Comment 16 of Reviewer #3: “SI, ‘Specifically, infrared absorption is directly correlated with the absorptive chemical group whose characteristic frequency reaching about terahertz in the experiment which is far away the DC-induced dipolar movement. Thus, DC-induced polarization is unable to contribute IR absorption signal and the influence is excluded.’

This section may not be convincing to a general audience. Could you please elaborate on how the infrared (IR) absorption changes as a function of an applied electric field? Specifically, if electric fields induce changes in the unit cell through electret effects, would these not also lead to alterations in IR absorption?”

Response:

When the DC bias is applied to the sample, it produces two major influences. First, DC bias will induce the reorientation of specific chemical structure and the corresponding transition dipole moment. Because the IR absorption is related to the direction of specific transition dipole moment as discuss above, the reorientation further leads to changes in IR absorption. Whether the IR absorption will change depends on whether the transition dipole moment is altered. Hypothetically, if electric fields induce change in the unit cell through electret effect and influence the selected transition dipole moment direction, it will lead to alterations in IR absorption. For the case of electrostatic electrets (ref. 1), in which the dipole is induced by charging on defects or interface, the characteristic frequency of electret effect-induced dipole moment is normally much lower than that of observed polar bonds and groups. Thus, the electrostatic electret effects would not affect the IR absorption within the observable wavenumbers. In fact, Fig. 2 has shown that no obvious difference can be observed between on-field and off-field IR-E loops.

Second, the DC bias will also induce effect that was not intrinsic to the sample including charge injection, ionic and charged species redistribution and accumulation resulting induced-polarization and electrostatic effect. (ref. 2). These will contribute to PFM signal, as discussed above in comments 13 and 14. However, these additional contributions do not exhibit the same infrared absorption characteristic frequency as inherent dipole moments. Moreover, in ePTFM, there is no AC voltage signal that couples with these spurious effects as in Comment 14, Eq. (4) (the AC electric filed

bring by light reaches terahertz, which do not share the same frequency of lock-in amplifier's reference signal). Therefore, it is possible to eliminate the DC bias-induced unwanted contribution by using pulsed IR.

References:

1. Sessler, G. M. Electrets. (Springer-Verlag Springer e-books, Berlin, Heidelberg, 2005).
2. Vasudevan, R. K., Balke, N., Maksymovych, P., Jesse, S. & Kalinin, S. V. Ferroelectric or non-ferroelectric: Why so many materials exhibit “ferroelectricity” on the nanoscale. Applied Physics Reviews 4, 021302 (2017).

Revisions:

According to this comment, the discussion about the influence of bias in ePTFM has been added.

Please refer to Supplementary Note 1, line 76-86 highlighted in red.

“When the DC bias is applied to the sample, it produces two major influences. First, DC bias will induce the reorientation of specific chemical structure and the corresponding transition dipole moment, which will lead to changes in IR absorption and detected by ePTFM. Secondly, DC bias will also induce additional contributions such as accumulations of charged species et.al. These additional contributions normally have a different IR absorption characteristic frequency to the polar bonds and groups we observed. Thus, they would not affect the results because of the selection of characteristic wavenumber. Moreover, in ePTFM, the AC electric field bring by IR light reaches terahertz, which do not share the same frequency of lock-in amplifier's reference signal. Since there is no AC voltage signal is coupled with the spurious effects as in Eq (4), it is possible to eliminate the DC bias-induced additional contribution by using pulsed IR.”

Comment 17 of Reviewer #3: “As an additional point, the term “DC-induced polarization” appears to describe a paraelectric effect, or possibly an electret effect, depending on the specifics. How do the authors perceive this contributing to Piezoresponse Force Microscopy (PFM)? To a general audience, it might seem more plausible that such effects would influence infrared (IR) absorption to a greater extent than the relative piezo response, considering that the former is an absolute measurement while the latter is relative.”

Response:

The reviewer's comment is appreciated. The DC-induced polarization mentioned here refers to the term $P = \chi\epsilon_0 V$, as has been discussed in comment 14. This term can

originate from a variety of processes, including charge injection, ion aggregation, and the electret effect, which can produce dipole, atomic, or electron polarization. The influence of DC-induced polarization on PFM signal has been shown by the second term in the radical sign $2Q\chi^2\varepsilon_0^2V_{DC}$ in Eq. (4) of comment 14.

The influence of DC bias and related electret effect to IR absorption has been discussed above. Please refer to comment 16.

Comment 18 of Reviewer #3: “Going further, in the main text you contradict this sentence ‘The ePTFM enables the identification of the two cases by observing variations of polarization-dependence infrared absorption at multiple wavenumbers.’ I.e. that the DC polarisation does change the absorption. This lack of consistency is very concerning.”

Response:

We appreciate the reviewer’s comment. It makes us realize that there has misleading in the description of DC bias influence in the main text and Supplementary Materials. The influence of DC bias to IR absorption has been discussed above in comment 16. It affects the infrared absorption by inducing a change in the structure-associated transition dipole moment. However, DC bias also induce additional contribution, which may plague the piezoelectric signal as discussed in comment 13 and 14. The description about excluding the influence of DC bias in ePTFM is refer to these additional contributions.

Revisions:

We have revised relative to descript the DC influence in IR absorption more clearly. Please refer to Supplementary Note 1, line 76-86 highlighted in red.

Comment 19 of Reviewer #3: “SI ‘Consequently, the on-field and off-field IR-E loops are consistent.’

Please show experimental evidence to back up this claim.”

Response:

Thanks for the reviewer’s comment and suggestion. The comparison of on-field and off-field results are shown in Fig. R12. They have been shown in Fig. 2 in the main text. The results show that IR-E loops share the similar tendency between on-field and off-field. In contrast, there are obvious differences between on-field and off-field PFM butterfly loops. Unsaturated PFM amplitude with lower coercive voltage is shown in the on-field PFM butterfly loops.

Fig. R12 Comparison of on-field and off-field IR-E loops (a) PFM hysteresis loops with on-field and off-field (bias on and bias off). **(b)** IR-E loops of 1289 cm^{-1} with on and off-field. The gray dot corresponds to 64 sampling data points of cantilever amplitude at each on/off-field stage. Red solid line corresponds to fitting curve according to average value of sampling points

Comment 20 of Reviewer #3: “Because the use of linear polarized laser source, the variation of IR absorptions can reflect both the changes in chemical concentration and molecular orientation during electric modulation’

This is a nice point, if you can show that you can use certain wavelengths to tack specific molecules it would add significantly to your story.”

Response:

Thanks for the reviewer’s appreciation. It is actually what we are going to demonstrate in our manuscript. The writing quality and structure arrangement may have misled the reviewer. Therefore, we carefully revised our manuscript to make it clearer and hope the reviewer can gain a more comprehensive understanding of our work.

Comment 21 of Reviewer #3: “Due to the coupling of AC probing electric bias and DC modulated electric bias, it is difficult to separate these contributions.’

No it isn’t. They are at different frequencies and lock-in amplifiers do this trivially.”

Response:

Thanks for the comment. The coupling of AC probing electric bias and DC modulated electric bias has been discussed above in Comment 14, equation (2) - (4). The DC-induced contribution terms are multiplied with the V_{AC} according to the formulate. So, these contributions are modulated by V_{AC} and share the same frequency of V_{AC} . In

PFM, the demodulation frequency of lock-in amplifiers is the same as V_{AC} . Through lock-in amplifier, only high-order terms of V_{AC}^2 and constant terms can be filtered out, remaining the first order term of V_{AC} including DC-bias induced additional contribution. Therefore, it is still unable to distinguish these contributions.

Revisions:

According to this comment, we have revised the manuscript and added a discussion about the effect of lock-in amplifiers in processing the DC-induced influences.

Please refer to Supplementary Note 1, line 55-59 highlighted in red.

“Because only high-order terms and constant terms can be filtered out, the first order term of V_{AC} including contribution of DC bias-induced polarization remains, as shown in the second term in the radical sign $2\chi^2\epsilon_0^2V_{DC}$. In addition, the electrostatic force contribution term $k^{-1}C(V_{DC} - V_{SP})$ also remains. Thus, the DC bias additional contributions cannot be separated by lock-in amplifier.”

Comment 22 of Reviewer #3: “However, ePTFM signal is directly determined by the IR absorption of chemical group, which is decoupling with DC modulated electric bias.’ ‘The ePTFM enables the identification of the two cases by observing variations of polarization-dependence infrared absorption at multiple wavenumbers.’

The manuscript contains two statements that seem to directly contradict each other: Does the polarization change the infrared (IR) absorption or not? Intuitively, I am inclined to agree with the statement suggesting that the absorption is likely polarization-dependent, and that DC modulation is intended to specifically alter the polarization.”

Response:

The reviewer is correct that IR absorption is polarization-dependent and DC modulation is used to alter the polarization and corresponding specific structure. It is one of the principles of our methods. The contradiction may be due to the misunderstanding of the decoupling of DC bias, which is similar to that in comment 16-18. According this comment, we have revised that statement of the decoupling in the main text to make it more comprehensive.

Revisions:

Please refer to Supplementary Note1, line 76-86 highlighted in red.

“...When the DC bias is applied to the sample, it produces two major influences. First, DC bias will induce the reorientation of specific chemical structure and the corresponding transition dipole moment, which will lead to changes in IR absorption and detected by ePTFM. Secondly, DC bias will also induce additional contributions such as accumulations of charged species et.al. These additional contributions normally

have a different IR absorption characteristic frequency to the polar bonds and groups we observed. Thus, they would not affect the results because of the selection of characteristic wavenumber.”

Comment 23 of Reviewer #3: “The utilization of pulsed infrared radiation is also able to get rid of electrostatic interference or ionic artifact induced by direct current (DC) electric bias, which is usually encountered in PFM^{33,34}.”

These issues are not typically encountered in standard operation. They may arise if the Piezoresponse Force Microscopy (PFM) is not operated correctly or if the sample is leaky. However, such issues are not inherent to the proper operation of PFM.”

Response:

Thanks for the reviewer’s comment. Related question about DC bias has been detailly discussed in Comment 13, 14 and 15.

Comment 24 of Reviewer #3: “...but also simultaneously contain chemical correlations which is absent in PFM loops.’

Could the authors clarify their statement? Are they suggesting that during ferroelectric switching, they can specifically monitor a molecule (such as CF₂) to observe its switching? Since this mechanism appears to be known, its significance in this context is not immediately apparent. Could the authors please elaborate on the benefits or novel insights this approach provides? While there may be interesting observations in this research, for publication in a general journal, the results and their implications need to be clearly understandable to non-specialists, which, at present, they are not.”

Response:

The reviewer’s comment and suggestion are appreciated. We can specifically monitor a molecule during ferroelectric switching by observing the alteration of its IR absorption. Compared to PFM, this technique can provide a new insight of how a specific chemical group evaluated during the polarization reversal. And this approach is able to reveal the chemical origins of polarization switching in polar organics by tracking it. Moreover, the ePTFM is an AFM based technique, which breakthroughs the diffraction limit of IR and enables the nanoscale characterization.

Revisions:

According to reviewer’s suggestion, we have rearranged the structure and clarified this statement in the manuscript to make it more clearly understandable to non-specialists.

Please refer to line 209-210 in the manuscript highlighted in red.

“It is worth noting that the IR-E loops not only reflect ferroelectric switching, but also simultaneously contain chemical correlations which is absent in PFM loops. In other words, it is able to specifically monitor a chemical group during ferroelectric switching by observing the alteration of its IR absorption.”

Comment 25 of Reviewer #3: “In addition, the repeatability and the exclusion of artifacts rising from shifting of absorption peak under electric bias was conducted and shown in Supplementary Figure S4 and Supplementary Figure S5.’

The authors mention repeating the loops once, but it’s not clear if attempts were made to repeat them more than once. The lack of clarity is compounded by Figures S4 and S5, which seem to show no shift in absorption peak under bias. This casts further doubt on the statement, especially considering that other data (Figure 3 f and g) appear to indicate an absorption shift. To conclusively demonstrate this, one diagnostic approach could be to measure the absorption spectra at different field strengths, for example, at 0 V, 10 V, 20 V, and 30 V.”

Response:

Thanks for the reviewer’s comment and suggestion. Repeating the measurements is attempts to show the reliability of the results. And we have conducted the IR absorption spectra at 0 V, 10 V, 20 V and 30V on both edge-on and face-on P(VDF-TrFE) lamellae according to reviewer’s suggestion. As shown in Fig. R13, there are no obvious peak shift as the bias change and shown the same trend in the Figure 2 and Figure 3 f and g.

Fig. R13. On-field AFM-IR spectrums at edge-on and face-on P(VDF-TrFE). The evolution of the IR spectra shows that there has nearly no absorption peak shifting.

Revisions:

According to this comment, relative section has been revised and Fig. R13 has added as Supplementary Fig. 8.

Please refer to line 236-239 in the manuscript highlighted in red.

“In addition, additional IR-E loops have been obtained in Supplementary Fig. 6, showing the reliability of the results. And the exclusion of artifact rising from shifting of absorption peak under electric bias was conducted and shown in Supplementary Fig. 7 and Supplementary Fig. 8.”

Comment 26 of Reviewer #3: “The ePTFM not only enables chemical correlation, but also produces a signal with more certainty.”

Could you specify what the "more" in "more certainty" is relative to? The term "more" implies a comparison, so it would be helpful to know what the baseline or reference point is that you are comparing against.”

Response:

Thanks for the reviewer’s comment and suggestion. The more certainty is relative to Piezoresponse force microscopy (PFM). As discussed in comment 13 and 14, PFM signal not only contributed from piezoelectric response, but also contain electrostatic force, joule heater, bias-induced dipole moment, et.al. Thus, the interpretation of PFM signal must be carefully, especially while DC bias was applied. On the other hand, DC bias-induced contributions that non intrinsic to the sample dipole can be filter out in ePTFM as discussed in comment 16. In this case, we think it is more certainty.

Revisions:

According to this comment, we have tone down the description about this sentence.

Comment 27 of Reviewer #3: “...two set of data are sorted and processed 167 according to whether the DC electric bias is applied, yielding the on-field/off-field PFM loops³⁸.”

Could the authors clarify what they mean by "on and off field PFM loops"? The referenced citation pertains to switching spectroscopy PFM, which involves observing hysteresis loops with a small AC voltage superimposed on a DC voltage. However, a search of the manuscript reveals no mention of “on-field” or “on-bias.” Understanding these terms is crucial for comprehending the manuscript.”

Response:

Thanks for the reviewer’s comment and suggestion. During the switching spectroscopy PFM measurement, the piezoresponse can be obtained during the DC pulse by superimposing the measuring AC sinusoid, or, alternately, immediately after each DC pulse. These two situations are named “on-field” and “off field”. The DC voltage is

varied in a series of voltage pulses with varying magnitude. Another example of on and off filed PFM loops can be shown in reference (ref. 1, please refer to its fig. 3a).

References:

1. Liu, Y., Zhang, Y., Chow, M.-J., Chen, Q. N. & Li, J. Biological Ferroelectricity Uncovered in Aortic Walls by Piezoresponse Force Microscopy. Phys. Rev. Lett. 108, 078103 (2012).

Revisions:

According to reviewer's comment, we have supplemented the description of on-field and off-filed.

Please refer to line 199-202 in the manuscript highlighted in red.

“The ePTFM signal can be obtained during the DC pulse by superimposing the measuring pulsed infrared radiation, or, alternately, immediately after each DC pulse. These two situations are named “on-field” and “off field”. respectively”

Comment 28 of Reviewer #3: “...in Fig 2d, the on-field PFM amplitude loop exhibits obviously lower coercive voltage and unsaturable amplitude than off-field.”

The interpretation of these loops is challenging, as the manuscript does not clearly define what is meant by “on and off field.” However, if the authors are suggesting that loops obtained using AC+DC voltages differ from those obtained with a large AC voltage alone, this observation is indeed correct. They also note that the ‘coercive field’ varies with the frequency of the AC voltage. This phenomenon is well-known and observable by taking bulk P(E) loops at different frequencies. Consequently, quantifying the significance of their findings in relation to these loops is difficult.”

Response:

We thank for the reviewer's comments. The definition of “on and off field” are shown above, please refer to comment 27. The PFM loops of Fig. 2d was obtain under the same AC voltage amplitude and frequency. Thus, the different and varies of coercive field is not owing to frequency or amplitude of the AC voltages. The only difference of measurement parameters between these two PFM loops in Fig. 2d is that one was obtained during the poling DC bias (on-field) and the other was obtained right after reach DC pulse (off-field). The comparation is attempted to show the influence bring by DC bias, which dramatic alter the PFM signal and blurs the intrinsic polarization behavior, as has discussed in comment 14.

Comment 29 of Reviewer #3: “In fact, the differences between on-field and off-field

PFM amplitude loops are the results of the coupling between DC modulation and AC excitation electric bias, and the replacement of AC electric bias with pulsed infrared laser in ePTFM successfully alleviated this influence as discussing above and in Supplementary Note 3.'

This sentence makes no sense, and the information in note 3 is subject to critique (see above)."

Response:

We appreciate the reviewer's comment. The influence of DC bias in ePTFM and PFM has been discussed above (please refer to comment 14, equation (2) - (4) and comment 16, 21). The contributions of DC bias are multiplied with AC voltage. Therefore, it is at the same frequency with AC voltage and coupling with each other. However, the replacement of AC voltage in ePTFM can be alleviated with this influence (please refer to comments 16, para.2). The reason is that the characteristic frequency of dipole moment induced by unwanted contributions such as charge injection or ion aggregation is far from our selective IR frequency (please refer to comments 17).

Revisions:

Relative reversions about the influence of DC bias in PFM have been added. Please refer to the reversion of comment 14 and 15

Comment 30 of Reviewer #3: "These results demonstrated that ePTFM will be a powerful tool for in-situ studies of nanoscale polarization evolution and will become more scientifically valuable in ferroelectric organics.'

As evident from the detailed comments provided in the preceding seven pages, I, as the referee, do not concur with this position."

Response:

The reviewer's comments and suggestions above is very much appreciated. A point-to-point response has been prepared carefully and detailly. We believe that respond letter and revised manuscript can address the reviewer's concerns.

Comment 31 of Reviewer #3: "(d) PFM phase distribution of the same region. The square pattern in phase and amplitude was still exist after several day confirming the out of plane ferroelectricity.'

The duration of stability does not confirm ferroelectricity. While some ferroelectric materials may back switch within hours, certain surface charges can remain stable for days. It is important to note that your Figures e and f do provide strong evidence of

ferroelectricity. However, using time as a defining criterion for ferroelectricity is not particularly useful or definitive.”

Response:

Thanks for the reviewer’s constructive comment and suggestion. We have rewritten the description here to emphasis the result of Fig. e and f.

Revisions:

Please refer to the caption of Supplementary Fig. 9 in the Supplementary Materials highlighted in red.

“(f) The phase variation in PFM-switching spectrum of toroidal face-on P(VDF-TrFE). The hysteresis loops in PFM confirming the out of plane ferroelectricity.”

Comment 32 of Reviewer #3: “Regarding Fig S6, I'm curious as to why a 'box-in-a-box' poling approach was not employed? This method typically involves poling a large area positively, a smaller area negatively, and an even smaller area positively again.”

Response:

Thanks for the reviewer’s comment and suggestion. There has no special purpose in choosing the poling pattern. According to this comment, we also reconducted another poling experiment as shown in Fig. R14. A ‘box-in-a-box’ domain was successfully written.

Fig. R14 Domain writing at toroidal face-on P(VDF-TrFE). (a) topography of face-on P(VDF-TrFE). (b) PFM amplitude of the same region. (c) PFM phase distribution of the same region.

Revisions:

According to this comment, Fig. R14 has been added as Supplementary Fig. 9 in Supplementary Materials.

Comment 33 of Reviewer #3: “For a general journal please explain what a gauche defect is.”

Response:

We thank the reviewer for the constructive suggestion. Gauche structure means torsional bond angle of two adjacent C-C have substituents at $\pm 60^\circ$ to each other (ref. 1, please refer to its Fig.1). Correspondingly, trans means 180° torsional bond angle to each other. Comparing to TTTT arrangement, the introduction of gauche will cause chain distortion. Thus, the appearance of gauche is also called gauche defects.

References:

1. Lovinger, A. J. Ferroelectric Polymers. Science 220, (1983).

Revisions:

The revised manuscript has added the explanation about the gauche defect.

Please refer to line 276-279 in the manuscript highlighted in red.

“Here, Trans(T) and gauche(G) feature torsional bond angle arrangements (Ferroelectric Polymers. Science 220, 1983). Trans represents arrangement with substituents at 180° to each other, and gauche represents for that at $\pm 60^\circ$.”

Comment 34 of Reviewer #3: “Analogous to this, we believe such strain also facilitates the transformation of chain conformation under electric bias. And it is a nature deduction that the OOP polarization reversal is essentially the result of electric bias inducing conformational change of polymer chains.’

This section is quite difficult to read and seems to contain speculative content. Could you please clarify what is meant by “strain facilitate” The term is ambiguous and needs further explanation.”

Response:

Thanks for the reviewer’s comment. We suggest that the formation of TTTG conformation is induced by the biaxial tensile stress and electric field. The biaxial tensile strain was perpendicular to the chain axis. It tends to distort the polymer chain. Thus, it’s not a stable energy state to maintain TTTT structure in this condition. The introduction of gauche can cause a distortion in polymer chain and reduced the energy (ref. 1). Thus, it facilitates or promotes trans to gauche transformation in polymer chain. A similar mechanism of TG transition under electric field was proposed by Zhu et al. recently (ref. 2), which support our model.

Reference:

1. Su, H., Strachan, A. & Goddard, W. A. Density functional theory and molecular dynamics studies of the energetics and kinetics of electroactive polymers: PVDF and P(VDF-TrFE). Phys. Rev. B 70, 064101 (2004).
2. Zhu, Y. et al. Operando Investigation of the Molecular Origins of Dipole Switching

in P(VDF-TrFE-CFE) Terpolymer for Large Adiabatic Temperature Change. Adv Funct Materials 2314705 (2024) doi:10.1002/adfm.202314705.

Revisions:

This section has been revised to be more comprehensive.

Please refer to line 314-319 in the manuscript highlighted in red.

“Biaxial strain in face-on P(VDF-TrFE) flattened the energy landscape of polarization states and it is curial for the development of toroidal polar topology⁴⁰. Analogous to this, we believe such strain also reduced the energy barrier between T and G, promoting a T-to-G transformation under electric bias. And above results promoted us to proposed that electric bias inducing T-to-G conformational change was contributed to the OOP polarization reversal.”

Comment 35 of Reviewer #3: ““Meanwhile, the pristine p-T3GT3G’ are unstable under the opposite electric bias and begin to degrade.”

If you cannot switch it, can you really call it a ferroelectric?”

Response:

Thanks for the reviewer’s comment. The degradation of positive TTTGTTT’ was actually to generate oppose polarization and realize polarization switch. It is analogy to the realignment of positive direction dipole moment to the negative direction. Thus, it is not meaning that it cannot switch.

Comment 36 of Reviewer #3: ““The mottled pattern denotes the co-existence of T3GT3G’ and other configurations, which affirms the aforementioned hypothesis of configurations transition.’

Could the authors explain how they have ruled out the possibility that the observed pattern is merely a result of strain gradients arising from topographical interactions? The detection of ferroelastic domains using PT IR has already been established in the literature [DOI: 10.1126/sciadv.1602165].”

Response:

Thanks for the reviewer’s comment and suggestion. The topographical interactions influence the signal by causing shift of the contact resonance frequency. Thus, during the measurement, we adopt phase-lock-loop to track the contact resonance frequency. A PLL tracking result has been record during the measurement as shown in Fig. R15, which shown that PLL was working well. That is a widely used method for ruled out the influence of topographical interactions in AFM-IR (ref. 1). As for the ferroelastic domain, ferroelastic rarely exists in P(VDF-TrFE). Thus, it will not influence the signal

as well.

Fig. R15. IR absorption mapping in face-on P(VDF-TrFE). (a) The ePTFM mapping of 1183 cm⁻¹ absorption. (b) Corresponding PLL tracking condition during the measurement in (a), which shows that PLL keep tracking the resonance frequency.

References

1. Schwartz, J. J., Jakob, D. S. & Centrone, A. A guide to nanoscale IR spectroscopy: resonance enhanced transduction in contact and tapping mode AFM-IR. *Chem. Soc. Rev.* 51, 5248–5267 (2022).

Responses to Reviewers' comments and corresponding changes

Reviewer #1 (Remarks to the Author):

with all my concerns addressed, this paper can be accepted for publication in Nature Communications.

Responses:

We sincerely appreciate the reviewers' positive feedback and his/her recommendation regarding our revised manuscript and responses.

Reviewer #2 (Remarks to the Author):

Upon revision the manuscript has improved somewhat, at least from the explanations and logic of exposition point of view. However, no efforts have been made for improving the quality of the writing, which remains a poor. The newly added text is of even poorer quality from the English perspective. In particular, the 53 pages long rebuttal letter which will be published if the paper is accepted is also poorly written. Correction of all inaccuracies in the rebuttal letter is beyond any reasonable effort that should be expected from the referees.

In summary, I find content of the paper to be interesting but the quality of the written discussion underwhelming.

Responses:

We sincerely thank the reviewer for pointing out the deficiencies in our responses and revised manuscript. We also thank the reviewer for his/her time and effort in carefully reviewing our manuscript. According to your advice, we have focused on improving the quality of writing in the second revised manuscript. The grammar, phrasing, and punctuation have been revised. The corrections are highlighted in red in the manuscript. Meanwhile, the comments and suggestions are very useful. We will respond to these comments one by one below.

Below, we follow the reviewer's marker. We report in black text from our original response, in light blue the reviewer's additional comments, and in dark red our response and corresponding changes.

Additional Comment 1 of Reviewer #2:

From response to Comment 1 of Reviewer #2: "...The AFM tip is platinum coated and made of silicon. The repetition rates of IR laser match the second resonance frequency

of the cantilever”

This information is pretty much useless, what is the frequency of the cantilever contact resonant frequency?

Response and related changes:

Thank you for the suggestion. The AFM probe we used has a force constant of about 0.2 N/m and a resonance frequency of about 13kHz. Thus, the contact frequency of the cantilever is about 200 kHz.

Related contents have been revised.

Please refer to lines 401-402 in the manuscript highlighted in red

“During IR-E loops measurement, repetition rates of IR laser were about 200 kHz matching the second resonance frequency of the cantilever.”

Please refer to lines 428-429 in the manuscript highlighted in red

“Repetition rates of IR laser were about 200 kHz, matching the second resonance frequency of the cantilever.”

Additional Comment 2 of Reviewer #2:

From response to Comment 2 of Reviewer #2: The IR spectrum under 0 V, 10 V, 20 V, and 30 V bias have been conducted and shown in Fig. R9.

Thank you. From the method perspective I find this the most useful information provided by the authors in the whole the manuscript. I would have much preferred to see the spectra corresponding to a full cycle, as 200 AFM-IR spectra do not lead to degradation of the sample, but I guess it is ok. Please note that the caption in Fig R9 (Fig 8 of the SI) is poorly written. What do you mean with “without notarization”? it seems that “IR abortion” should be changed to “IR absorption”

Response and related changes:

Thank you for your appreciation of the additional data. We apologize for the mistake in the caption and thank you for pointing it out. The spelling of “notarization” should be changed to “normalization”. The corresponding mistake has been corrected in the Supplementary Materials.

A full cycle of the corresponding spectra can provide more information. However, such a test also causes greater damage to the sample. As shown in Fig. R1, the combined effect of voltage and infrared irradiation over 10 seconds can cause irreversible damage to the sample. In our approach, the electric bias is applied in the form of pulses, which can significantly reduce this impact. We believe that prolonged application of high voltage along with IR thermal effects will result in the accumulation of heat, and cause

damage to the probe or sample. This will further result in signal instability, which hampers further measurements. By properly controlling the sample quality, testing bias, laser parameters, and contact force between the tip and the sample, a full cycle spectrum might be acquired. However, additional exploration is needed and it is beyond the scope of our work. We suggest studying it in the future.

Fig. R1. The influence of electric bias in AFM-IR measurements. AFM topography was shown after long-term voltage loading. The bias was applied during the IR spectra measurements lasting more than 10 s. The voltage ranges from 30 V to 40 V. The scale of the whole image was 1.5 μm .

Additional Comment 3 of Reviewer #2:

From response to Comment 3 of Reviewer #2: We have conducted a larger scale of ePTFM mapping, as shown in Fig. R10. The two maps of orthogonal dipole moment confirm that there exists hierarchical structure related to the toroidal domain. The local IR spectra result was also shown in Fig. R10, which shown that the variation was not dominated by noise or mechanical effects.

Fig. R10. The ePTFM maps of toroidal domain face-on P(VDF-TrFE). **a** The topography of face-on P(VDF-TrFE). **b** The ePTFM mapping of 1289 cm^{-1} absorption. **c** The ePTFM mapping of 1183 cm^{-1} absorption. **d** Comparison of local IR spectrum of different point marked in **e**, which show that the distribution of IR mapping was not owing to the noise or mechanical effect. **e** Zoom in 1289 cm^{-1} IR mapping of **b**. **f** Zoom in 1183 cm^{-1} IR mapping of **c**

Thank you for the additional data. If ePTFM is sensitive to the molecular orientation/conformation, and the sample is structured (i.e. its molecules are oriented in different directions and groups oriented in different direction at different locations as it should be expected for the toroidal domains) maps of modes with orthogonal moment should give raise spatially different maps. Instead, the maps presented above depicts essentially the same spatial distribution of intensities. This seems a strong contradiction and may indicate some fundamental problem with the methodology.

Response and related changes:

Thank you for the comments. The sensitivity of ePTFM to the molecular orientation/conformation is based on the polarization dependence of AFM-IR, which has been verified in various literature (ref. 1). We believe there is no need to worry

about the fundamental issue of the method.

There may have been a misunderstanding due to the lack of description of Fig. R10, resulting in this confusion. We apologize for that. Fig. R10 aimed to verify that the variation of IR mapping was not dominated by noise, addressing previous comments. Therefore, to maintain the consistency of the measuring conditions, it was measured with a p-polarization IR beam. Under p-polarization, the IR mapping can barely reflect the different directions or orientations of the in-plane component. Considering that the toroidal domains are mainly oriented along in-plane directions, it is reasonable that maps of modes with orthogonal moments do not reflect the toroidal molecular orientations. Therefore, their IR mapping under p-polarization will give rise to a uniform absorption level for the area in face-on P(VDF-TrFE). This is consistent with ref. 2 (please refer to its Supplementary Materials, Fig. S31).

The sensitivity of molecular orientation can be shown by characterizing face-on P(VDF-TrFE) under s-polarization, as shown in Fig. R1a and b. The results show that the IR mapping gives rise to different evolution tendencies, which is consistent with ref. 2 (please refer to Fig. 4e). Moreover, for verifying the orientation sensitivity in p-polarization, an IR mapping of edge-on P(VDF-TrFE) with a designed out-of-plane domain was characterized under p-polarization. As shown in Fig. R2, the IR absorption of 1289 cm^{-1} and 1183 cm^{-1} exhibit different intensity contrasts with their surroundings owing to the molecular orientation sensitivity.

Fig.R2. The sensitivity of ePTFM to molecular orientation. **a** IR mappings of toroidal P(VDF-TrFE) in s-polarization at the wavenumbers of 1289 cm^{-1} (**a**) and 1183 cm^{-1} (**b**). The intensity varies with the orientation. And different distribution of intensity is observed. **c** Out-of-plane PFM phase of edge-on P(VDF-TrFE). **d** PFM amplitude of edge-on P(VDF-TrFE). It shows that domains were written in it. Under p-polarizations, IR absorption is also sensitive to molecular orientation. The corresponding IR mappings

at 1183 cm^{-1} (d) and 1289 cm^{-1} (e) in p-polarizations show different contrasts compared to their surroundings. This is consistent to the results in Fig.2.

References:

1. Hinrichs, K. & Shaykhutdinov, T. Polarization-Dependent Atomic Force Microscopy–Infrared Spectroscopy (AFM-IR): Infrared Nanopolarimetric Analysis of Structure and Anisotropy of Thin Films and Surfaces. *Appl Spectrosc* 72, 817–832 (2018).
2. Guo, M. et al. Toroidal polar topology in strained ferroelectric polymer. *Science* 371, 1050–1056 (2021).

Additional Comment 4 of Reviewer #2:

From response to Comment 2.5 of Reviewer #1: “The probing depth of AFM-IR is determinate by experiment setting. By setting different laser power, the depth sensitivity can be altered.”

This is incorrect. The probed depth in AFM-IR is still not yet fully understood and subject of research, but it does not depend on the laser power.

Response and related changes:

Thank you for the comment. We notice that the depth sensitivity may depend on various factors. Some researchers have showed that depth sensitivity can be influenced by the laser power, laser pulse frequency, and laser pulse width (ref. 1). The increase of laser power may enhance the thermal oscillation generated in the buried layer. Thus, it can increase the sensitivity of the deeper layer. And we agree with the statement that probed depth is still under investigation and it is still not yet fully understood. Thus, we did not discuss it in the main text. Relevant issues have been discussed recently. For example, different operation modes also influence probing depth (ref. 2). And surface sensitivity mode has been proposed by controlling the repetition rate of the laser pulse and force modulation frequency (ref. 3 and ref. 4).

References:

1. Prine, N., Cardinal, C. & Gu, X. Understanding and controlling the depth sensitivity of scanning probe based infrared imaging and nanospectroscopy for buried polymeric structures. *Nanoscale* 15, 10244–10253 (2023).
2. Xie, Q. & Xu, X. G. What Do Different Modes of AFM-IR Mean for Measuring Soft Matter Surfaces? *Langmuir* 39, 17593–17599 (2023).
3. Mathurin, J. et al. Photothermal AFM-IR spectroscopy and imaging: Status, challenges, and trends. *Journal of Applied Physics* 131, 010901 (2022).

4. Schwartz, J. J., Jakob, D. S. & Centrone, A. A guide to nanoscale IR spectroscopy: resonance enhanced transduction in contact and tapping mode AFM-IR. *Chem. Soc. Rev.* 51, 5248–5267 (2022).

Additional Comment 5 of Reviewer #2:

From response to Comment 7 of Reviewer #3: “Also, the evolution includes the generation or vanish of specific chemical structure. For example, there exists gauche to trans structure transformation in dipole switching of P(VDF-TrFE-CFE) (ref.2).”

I disagree with the wording “chemical structure” used in the rebuttal and throughout the manuscript when referring to trans and gauche conformations. Such wording is confusing as one can easily deduce from the numerous referees’ questions. The polymer molecules remain the same throughout the experiments, just their conformation changes. The author shall refer to changes in the trans and gauche population as to changes in their conformations.

Response and related changes:

Thank you for the comments and suggestions. It is very helpful. The “chemical structure” is a bit inaccuracy in the description of trans and gauche conformations. And we have also reconsidered the description referring to changes in trans and gauche conformations. Also, the description of changes in gauche and trans have been revised. Corresponding changes are shown below.

Please refer to lines 250-251 in the manuscript highlighted in red

“The appearance of more gauche **conformations** is understandable.”

Please refer to lines 253-255 in the manuscript highlighted in red

“Compared **with** TTTT, both TTTG, and TGTG can be considered to be disordered **conformations** since the introduction of gauche will cause chain distortion⁵⁰”

Please refer to lines 265-267 in the manuscript highlighted in red

“**On the basis of these** observations, we postulated that OOP polarization reversal is caused by electric bias-induced **changes in trans and gauche populations.**”

Reviewer #3 (Remarks to the Author):

The authors have conducted an extensive amount of work and have provided detailed responses to all questions raised. This additional effort, particularly the high-quality new data sets, has significantly improved the quality of the manuscript. While I still disagree with the authors on several points in the review process, this disagreement is on the level of scholarly debate, and the authors have justified their position well.

Therefore, I see no reason why this work cannot be published as it stands.

Response:

We sincerely thank the reviewer's recognition and recommendation of our work. We also thank the reviewer for taking the time and effort to review our work.

Responses to Reviewers' comments and corresponding changes

Reviewer #3 (Remarks to the Author):

I have been asked to assess the authors' responses to Reviewer 2's comments and determine whether they have been satisfactorily addressed. I will address each point in detail below; however, in my opinion, none of the responses adequately address Reviewer 2's concerns.

Response:

We sincerely appreciate the reviewer's dedication to reviewing our manuscript. The authors also thank the reviewer for the useful perspective on the previous comments and suggestions for improving our work. However, as the reviewer joined the discussion midway, there may have been some misunderstandings of previous matters. To address the reviewers' concerns, we have conducted additional experiments, which further affirm the validity of our findings and methodologies. The comments are responded to one by one below. For ease of reading, the reviewer's original comments are highlighted in light blue, while our responses and subsequent revisions are clearly noted in black and red, respectively.

Comment 1:

Reviewer 2 has asked for improvements in the writing quality, making this the second request for the same thing. To offer an objective perspective on the updated writing quality, I have utilized a large language model for feedback. The following are the five key issues identified by the model (all of which I agree with):

- 1) Clarity and Precision: Some sentences are overly complex, with nested clauses or passive constructions that obscure the meaning. Simplifying sentence structure and using more active voice could improve the clarity.
- 2) Consistency in Terminology: The manuscript uses some terms interchangeably (e.g., polarization switching, ferroelectric reversal) without clear definitions for each. It would be helpful to define these terms early on and use them consistently.
- 3) Redundancy and Repetition: There are instances of repetition, particularly in the Results and Discussion sections, where similar points are reiterated. Eliminating redundancy could make the paper more concise and easier to follow.
- 4) Transitional Phrases: The flow between paragraphs and sections can be improved by using more effective transitional phrases. Some paragraphs feel disjointed because they jump from one concept to another without a clear link.
- 5) Jargon and Technical Language: While the manuscript is meant for a technical audience, it sometimes relies heavily on jargon without sufficient explanation. Providing brief explanations or definitions for highly specialized terms (e.g., "toroidal face-on P(VDF-TrFE)") could help make the manuscript more accessible.

Response:

We are grateful for the reviewer's comments and suggestions, which have been useful in enhancing our writing quality. In accordance with the feedback provided, we have

carefully implemented corresponding revisions to the manuscript. Additionally, we also utilized the English language editing services offered by the Springer Nature author service, which further refined the phrasing and enhanced the clarity and flow of our sentences. We believe that the revised manuscript will meet the expectations of the reviewers.

Revisions:

Revisions have been made throughout the manuscript regarding the phrasing, clarity, and fluency. The following lists the primary revisions.

1) Sentence structure has been simplified and a more active voice has been used to improve the clarity of the manuscript.

Please refer to lines 45-51 in the manuscript highlighted in red

“However, the inherent complexity of the structure and chemical constituents **poses challenges in fully comprehending the intricate relationship** between the polarization behavior and molecular structure. **This gap in understanding has hindered** further exploration in evaluating and tailoring **these materials for** desired physical functionalities¹¹⁻¹³. To **bridge this gap**, an **integrating** analysis of nanoscale chemical **properties** and polar evolution is needed.”

Please refer to lines 97-98 in the manuscript highlighted in red

“**The amplitude of cantilever oscillation allows for the determination of** Infrared absorption³⁵”

Please refer to lines 110-112 in the manuscript highlighted in red

“Finally, **ePTFM captures** a switching spectrum **that shows case selective** IR absorption evolving with a modulated electric bias (**Fig. 1b, Supplementary Fig. 1**)”

Please refer to lines 137-139 in the manuscript highlighted in red

“In comparison, PFM **characterizes** nanoscale polarization **through the local electromechanical response**. However, **it cannot directly reveal** the origin and mechanism of polarization evolution.”

Please refer to lines 165-167 in the manuscript highlighted in red

“Then, **we monitored** the IR absorption at 1289 cm^{-1} **to explore the configuration changes** during **ferroelectric switching**.”

Please refer to lines 189-190 in the manuscript highlighted in red

“Moreover, **we can observe correlations among** different chemical components **by selecting various** IR absorption peaks in Supplementary Fig. 3”

Please refer to lines 223-224 in the manuscript highlighted in red

“Toroidal polar topology has never been observed in ferroelectric polymers until its discovery in the **biaxial strained** P(VDF-TrFE) lamellae **with face-on structure** (Fig. 3a). **These developments have led to** emerging functionalities^{44,45}.”

Please refer to lines 271-272 in the manuscript highlighted in red

“A ferroelectricity-related inverse butterfly pattern is observed in IR-E loops at 1120 cm^{-1} . **This pattern interconnects the** evolution of the conformation and polarization, confirming our deduction.”

Please refer to lines 292-294 in the manuscript highlighted in red

“The **subsequent** increase in downward polarization is **mainly** caused by the degradation of p-T₃GT₃G'. **This result** decreases **in** the IR absorption intensity at 1120

cm⁻¹ but increases in the PFM amplitude”

2) Interchangeable terms have been defined early and use them consistently

Please refer to lines 148-149 in the manuscript highlighted in red

“The polarization reversal of edge-on P(VDF-TrFE) exhibits hysteresis characteristics of ferroelectric switching.”

Please refer to lines 177-178 in the manuscript highlighted in red

“Therefore, Fig. 2e indicates the direct correspondence between the -CF₂ rotation and ferroelectric switching.”

Please refer to lines 198-199 in the manuscript highlighted in red

“...which exhibits minimal variation during OOP ferroelectric switching.”

3) Redundant sentences have been eliminated to make the paper more concise and easier to follow.

Please refer to lines 50-51 in the manuscript highlighted in red

~~“By integrating chemistry with polarization, the specific evolution of chemical structures during polarization switching can be observed, thereby revealing the underlying chemical origins of this phenomenon.”~~

Please refer to lines 161-163 in the manuscript highlighted in red

“The ePTFM signal can be obtained during the direct current (DC) pulse acquisition by superimposing the measured pulsed infrared radiation or immediately after each DC pulse.”

Please refer to lines 219 in the manuscript highlighted in red

~~“These results demonstrate ePTFM as a powerful tool for in situ studies of nanoscale polarization evolution, particularly in ferroelectric organics.”~~

4) More transitional phrases have been used to improve the flow between paragraphs and sections.

Please refer to lines 57-58 in the manuscript highlighted in red

“However, there are still constraints in employing these methods to monitor chemical components during nanoscale polarization evolution.”

Please refer to lines 74-75 in the manuscript highlighted in red

“Among these, linearly polarized AFM-IR stands out as a method used to achieve nanoscale analysis of molecular orientation³⁴.”

Please refer to lines 105-106 in the manuscript highlighted in red

“Following low-pass filters, the results yield the in-phase component *X* and the quadrature component *Y*.”

Please refer to lines 139-141 in the manuscript highlighted in red

“By selecting a specific IR wavenumber during polarization switching, the technique enables the tracking of particular molecular structure, thereby revealing the underlying chemical origins.”

Please refer to lines 171-172 in the manuscript highlighted in red

“This means that by observing the alteration in IR absorption, we can monitor the behavior of specific molecular configurations during ferroelectric switching.”

Please refer to lines 178-179 in the manuscript highlighted in red

“To elucidate the correlation, three critical stages of ferroelectric switching are marked in the IR-E loop”

Please refer to lines 230-232 in the manuscript highlighted in red

“To understand the electrodriven mechanism of ferroelectric switching, toroidal P(VDF-TrFE) lamellae were fabricated, and IR-E analysis was conducted via ePTFM.”

Please refer to lines 253-255 in the manuscript highlighted in red

“As the introduction of gauche conformation will cause chain distortion⁵⁰, both TTTG and TGTG can be considered to be disordered conformations compared to TTTT. ~~Thus, it induces relaxor behavior in toroidal face-on P(VDF-TrFE).~~”

Please refer to lines 306-308 in the manuscript highlighted in red

“To provide a comprehensive understanding of the conformational evolution during ferroelectric reversal, on-field ePTFM mapping of the IR absorption at 1120 cm⁻¹ was conducted on toroidal P(VDF-TrFE).”

5) A brief explanation or definitions have been provided for specialized terms “toroidal face-on P(VDF-TrFE)”.

Please refer to lines 228 in the manuscript highlighted in red

“For brevity, it is referred to as toroidal P(VDF-TrFE).”

Please refer to lines 231-232 in the manuscript highlighted in red

“...toroidal P(VDF-TrFE) lamellae were fabricated...”

Please refer to lines 239-240 in the manuscript highlighted in red

“...indicating an increased amount of TTTG in face-on P(VDF-TrFE) with toroidal domains.”

In addition, numerous sentences have been polished. Please refer to the manuscript for more detailed modifications.

Comment 2:

Information requested: “...What is the frequency of the cantilever contact resonant frequency?”

Information given: “about 200 kHz”.

The term “about” is not a precise number. This is a fundamental and important question. If the authors are unable to provide an exact frequency (which should be recorded in both their lab book and AFM data files), it raises concerns about their experimental proficiency and casts doubt on the reliability of the data as a whole.

Response:

We thank the reviewer for reminding us. The precise contact resonance frequency used in our experiment is 187.815 ± 14.1 kHz, and detailed information is presented below in Table 1. Specifically, we chose to utilize the second-order contact resonance mode in our experiment, which is commonly adopted in atomic force microscopy-based infrared spectroscopy (AFM-IR) measurement (please refer to Reference 1, Figure 2b and Section 4: Resonance-enhanced). It is important to note that the precise contact resonant frequency differs among each individual probe. Consequently, the contact

resonance frequency adopted in our study also varies accordingly. In addition, during the experiment, a phase-locked-loop (PLL) circuit was adopted to track the contact resonance frequency. Thus, the repetition rates of the IR laser are maintained at the contact resonance frequency.

Table 1. The settings for the IR pulse repetition rate in the experiment.

Corresponding Figures	Setting of IR pulse frequency
Figure 1c	201.69 kHz
Figure 2e	178.73 kHz
Figure 2f	198.32 kHz
Figure 3c	201.91 kHz
Figure 3f	173.72 kHz and 201.91 kHz
Figure 3f and 3g	196.48 kHz
Figure 4b~h	196.81 KHz

Reference:

1. Schwartz, J. J., Jakob, D. S. & Centrone, A. A guide to nanoscale IR spectroscopy: resonance enhanced transduction in contact and tapping mode AFM-IR. *Chem. Soc. Rev.* **51**, 5248–5267 (2022).

Revisions:

To be more detailed, the methods have been revised in the manuscript as shown below and Table 1 has been added to the Supplementary Materials as Supplementary Table 1.

Please refer to lines 405-407 in the manuscript highlighted in red

“During the IR-E loop measurements, the repetition rate of the IR laser was in the range of 187.815 ± 14.1 kHz, matching the second-order contact resonance mode of the cantilever. The precise frequency is shown in Supplementary Materials Table 1.”

Please refer to lines 433-435 in the manuscript highlighted in red

“The repetition rate of the IR laser was in the range of 187.815 ± 14.1 kHz, matching the second second-order contact resonance mode of the cantilever. The precise frequency is shown in Supplementary Materials Table 1.”

Comment 3:

The reviewer requested the full cycle data sets, which the authors have declined to provide, citing sample damage. However, I expect this is precisely why the reviewer asked for both spectra—to assess the extent of the damage.

If the authors believe the sample degrades at 10V, it seems counterintuitive to then apply voltages of 20V and 30V. Wouldn't it be more prudent to experiment with voltages below 10V in order to avoid damaging (changing) the sample?

Additionally, in the next section, the authors claim that ± 20 V switches their sample, raising the question: does this switching voltage cause damage to the sample? Could this suggest that the PFM signal originates from an electret effect?

In my opinion, if the measurement process itself alters the sample, the resulting data becomes problematic and does not provide meaningful insight into the true properties of the sample. For instance, does the data reflect the initial state, a transient state, or a

final, degraded state?

Furthermore, when introducing a new technique, it is essential to demonstrate its reliability. At a minimum, this requires strict control over variables and so the authors statement that “.... properly controlling the sample quality, testing bias, laser parameters, and contact force between the tip... is beyond the scope of our work”, is tough to accept.

Response:

We thank the reviewer for the comments and suggestions. We noticed that there are mainly two concerns. The first issue is about the sample damage. Second, a full cycle spectrum is requested to demonstrate the reliability of our IR-E loops measurement.

(1) Sample damage

It's worth noting that there are two distinct experimental schemes involved. One is proposed in our original manuscript and is called the “IR-E loop” method, which measures the IR response at a fixed wavenumber while applying pulse bias sequences of a full cycle. The other experimental scheme is suggested by reviewers, which acquires IR spectrum under each DC voltage level. It is referred to as “multi-spectrum (MS) method”. We would like to note that the sample damage observed when voltages exceeded 30V, as mentioned in our previous response letters, was specifically related to the MS method proposed by Reviewer #2. Thus, the sample damage that occurred in the MS method does not imply that our proposed IR-E loop method is problematic. No obvious sample damage is observed in the IR-E loop method. To further validate our IR-E loop method, we conducted additional experiments within the same sample, in which the IR spectra before and after the IR-E measurement are compared. The result confirmed that the measurement of IR-E loops does not cause damage to the sample, as shown in Fig. R1. Moreover, the IR-E loop is repeatable at the same point.

Fig. R1. Comparison of IR spectra before and after IR-E measurement. Initially, the sample was acquiring a spectrum before conducting the IR-E loop measurement. Subsequently, a triangle-

squared bias sequence with voltage up to 35V was applied to the sample along with IR radiation for IR-E loop measurement. The IR power was set at 11.83% (~0.3mW). The IR radiation was set at 1289 cm^{-1} . After the IR-E loop measurement, the spectrum was collected at the sample point, as shown in the red line.

(2) Full cycle IR spectrum

The reviewer requested a full cycle spectrum, i.e., the MS method, to demonstrate the reliability of our IR-E loop measurement. In our previous response letter, we showed that under the combined effect of an electric field and IR exposure, sample damage occurs when the voltage exceeds 30 V for 10 s. Thus, it is difficult to obtain a full cycle IR spectrum, in which the required time is much greater. However, we have managed to obtain the full cycle spectrum by means of a series of optimizations. As shown in Fig. R2, a full cycle spectrum data, involving all the key points during polarization reversal, are presented. We have carefully controlled the various parameters to ensure the integrity of our samples. These include fabricating a sample with a switching voltage below 10 V, minimizing the duration of IR exposure and voltage loading, using a laser power less than 27.12% (~0.7mW), and controlling the change of deflection value below 0.05 V between engage and withdraw state. The results exhibit a similar trend to the results of Fig.2e and Fig. 2f in the manuscript, which provides additional evidence for the robustness and reliability of our results.

Fig. R2. A full cycle spectrum experiment conducted on P(VDF-TrFE). The PFM hysteresis loops (a) exhibit distinct ferroelectric behavior, while OOP-plane PFM phase mapping results (b),

and the corresponding amplitude results (c) further confirm the ferroelectricity of the edge-on P(VDF-TrFE) sample. Notably, the switching voltages are observed at approximately 5V. The spectrum (d) was collected under a full cycle DC bias, ranging from -10 V to 10 V and then back to -10 V. The insets provide a closer examination of the spectrum at the peaks of 1289 cm^{-1} and 1183 cm^{-1} . Additionally, e and f show the evolution of the amplitude at 1289 cm^{-1} and 1183 cm^{-1} with changing bias, respectively. These results are consistent with the IR-E loops result in our manuscripts, offering a consistent understanding of the ferroelectric evolution characteristics. The P(VDF-TrFE) film was fabricated using the same methods described in the manuscript, but with a lower concentration solution of 1% w/v and the omission of self-assembly step.

3) Other concerns

“If the authors believe the sample degrades at 10V, it seems counterintuitive to then apply voltages of 20V and 30V. Wouldn't it be more prudent to experiment with voltages below 10V in order to avoid damaging (changing) the sample?”

The reviewers consider that it is counterintuitive to apply voltages of 20 V and 30 V. There may exist a misunderstanding. The sample does not degrade at 10 V. As discussed in the section titled “(2) Full cycle IR spectrum”, our previous response letter presented results indicating that sample damage occurs when the bias exceeds 30 V for 10 s. Consequently, the acquisition of the spectrum was performed with the voltage control maintained below 30 V in the previous measurement. Thus, it is not counterintuitive.

The reviewer suggested experimenting with voltages below 10 V to avoid damaging (changing) the sample. This is a useful suggestion as it minimizes potential damage to the samples. In fact, as shown in Fig R2, we also did it this way. However, there are instances where applying a bias greater than 10 V is unavoidable. For example, the toroidal polar structure of P(VDF-TrFE) typically exists at a thickness of 100 nm (please refer to Reference. 3 and its Fig. S17). Under these circumstances, reducing the switching bias is not feasible. Therefore, the IR-E loop method was adopted in our experiment, which is more capable of tolerating higher voltages than the MS method, reducing the risk of sample damage.

“Additionally, in the next section, the authors claim that $\pm 20\text{V}$ switches their sample, raising the question: does this switching voltage cause damage to the sample? Could this suggest that the PFM signal originates from an electret effect?”

The reviewer was concerned about the damage caused by the switching voltage. Our measurement in the manuscript is based on the IR-E loop method. As discussed above in the section titled “(1) Sample damage”, the switching voltages of ± 20 V are safe for our sample. Moreover, the IR-E loop results are repeatable under the same engagement point, which also shows that the sample has not been damaged. Since no damage occurred, it is unlikely that the PFM signal originated from an electret effect.

“In my opinion, if the measurement process itself alters the sample, the resulting data becomes problematic and does not provide meaningful insight into the true properties of the sample. For instance, does the data reflect the initial state, a transient state, or a final, degraded state?”

We share the same opinion with the reviewers that the measurement process should

alter the samples less. In our research, we have noticed the potential impact of sample damage and have implemented a strategy to prevent it. Specifically, the IR-E loop method we proposed is one of our strategies to avoid sample damage, which provides greater tolerance in the selection of voltage, laser power and sample.

“Furthermore, when introducing a new technique, it is essential to demonstrate its reliability. At a minimum, this requires strict control over variables and so the authors statement that “... properly controlling the sample quality, testing bias, laser parameters, and contact force between the tip... is beyond the scope of our work”, is tough to accept.”

Our techniques mainly focus on the IR-E loop methods, whose reliability has already been demonstrated. Since the MS method is not the technique that we proposed, we previously believe that further tuning its parameters for the full cycle testing is unnecessary. Nevertheless, as shown above in Fig.R2, we also explored the MS method of measuring the spectrum under each DC voltage level. The results align well with our original method (IR-E loops method). Both methods reflect the same polarization reversal properties.

Reference:

1. Guo, M. *et al.* Toroidal polar topology in strained ferroelectric polymer. *Science* **371**, 1050–1056 (2021).

Revisions:

To better clarify the reliability of our experiment, Fig. R2 has been added as Supplementary Fig.8 in the Supplementary Materials.

Please refer to lines 203-205 in the manuscript highlighted in red.

“Furthermore, we also conducted on-field spectrum characterizations (Supplementary Fig. 8), which demonstrate consistency of the IR-E loops results.”

Comment 4:

The reviewer requested additional ePTFM data and suggested that it showed a contradiction with the existing data. While I understand the authors’ argument that strong out-of-plane contrast may not be expected in this particular orientation, I find the response unconvincing. In my opinion, further investigation and additional data are necessary to adequately address this issue.

Response:

We thank the reviewer for understanding our argument. We have identified two primary concerns raised in this comment.

First, the reviewer expressed concern about why the maps presented in the previous response (Response Letter A, Comment 3 of Reviewer #2, Fig. R10) depict essentially the same spatial distribution of intensities. In fact, this is not contradictory to the existing data. Our data demonstrate that for P(VDF-TrFE) with toroidal polar structures, the transition dipole moment is predominantly distributed within the plane. In addition, the characterization of the out-of-plane PFM (Piezoresponse Force Microscopy) also shows that their polar orientations are basically the same in the out-of-plane direction (please refer to Figure 3 in the manuscript and the corresponding

discussion, as well as Reference 1 for further details). Thus, variations in orientation of the sample primarily occur within the plane. However, our experiments use polarized light oriented in out-of-plane direction, which primarily detects infrared absorption that reveals orientations or conformations change in out-of-plane dimension. Therefore, even though the in-plane polarization of toroidal P(VDF-TrFE) was varied at different locations, it's reasonable that our maps do not reflect corresponding contract and presented nearly the same spatial distribution.

Second, the reviewers expressed concern about the fundamental issues of the methodology, specifically its sensitivity to molecular orientation or conformation. To address these concerns, we provide additional ePTFM data that demonstrates the successful characterization of the molecular orientation/conformation. As shown in Fig. R3, the maps of modes with orthogonal moments, i.e. 1289 cm^{-1} and 1183 cm^{-1} , clearly show different distributions of intensity. The results confirm the method's sensitivity to molecular orientation or conformation. Details are provided in the caption.

Fig. R3. The sensitivity of molecular orientation or conformation in IR mapping. **a** Topography of the edge-on P(VDF-TrFE). **b** Out-of-plane PFM (Piezoresponse force microscopy) amplitude (**b**) and phase (**c**) mappings showing the domain structure written in it. **d** Corresponding IR mapping at 1289 cm^{-1} . **e** Cross-sectional profile corresponding to the dashed white line in **d**. **f** Corresponding IR mapping at 1183 cm^{-1} . **g** Cross-sectional profile corresponding to the dashed white line in **f**. A cross-sectional line was acquired with a width of 50 pixels for averaging. A domain was induced on the edge-on P(VDF-TrFE), leading to variations in the molecular orientation within the sample. Inside the domain writing area, the molecules were aligned by bias. In contrast, outside the domain writing box, molecules were randomly aligned. Therefore, for the transition dipole moment corresponding to 1289 cm^{-1} , there is a greater alignment in the out-of-plane direction within the domain writing area compared to outside. Consequently, under vertical polarized infrared radiation, this led to stronger IR absorption at 1289 cm^{-1} within the domain writing area. Owing to the sensitivity of molecular orientation/conformation in ePTFM, the orthogonal moment at 1183 cm^{-1} was distributed in the opposite direction as shown in **f** and **g**. The sample used was the same in Fig. R2.

Reference:

1. Guo, M. *et al.* Toroidal polar topology in strained ferroelectric polymer. *Science* **371**, 1050–1056 (2021).

Revisions:

To clarify the ePTFM mapping in toroidal P(VDF-TrFE), additional discussion has been added in the caption of Supplementary Fig. 12.

Please refer to lines 191-195 in the Supplementary Materials highlighted in red

“For P(VDF-TrFE) with toroidal polar structures, the transition dipole moment is predominantly distributed within the plane (Please refer to Figure 3 and the corresponding discussion). Thus, even though the in-plane polarization of toroidal P(VDF-TrFE) was varied at different locations, it’s reasonable that the maps do not reflect corresponding contrast and presented nearly the same spatial distribution..”

Comment 5:

The reviewer rightfully pointed out the incorrect statement: "The probing depth of AFM-IR is determined by experiment setting." Although you acknowledge in your response that "the probed depth is still under investigation and not yet fully understood," the suggestion that increased power simply results in greater depth is problematic. While higher power may indeed increase the volume of the sample being probed, this is not equivalent to having precise control over the probing depth. This distinction is crucial.

I recognize that this might stem from a linguistic issue; however, for a published paper, the language must be both technically accurate and precise. Misstatements like this can lead to confusion and cast doubt on the overall rigor of the manuscript. It also raises concerns about whether other such inaccuracies may have gone unnoticed.

Response:

We thank the reviewer for providing their perspective in this comment. The statement regarding the probe depth is not included in our manuscript. Rather, it is only discussed in response letters. We understand that increased power did not simply result in greater

depth. We have conducted in-depth investigations on this topic (Reference. 1-4). In our previous response, our intention was to express that the laser power may be one of the parameters influencing on probe depth. We believe that no linguistic issues related to this aspect have occurred in the manuscript. Furthermore, we exercised great care in revising the manuscript to avoid inaccuracies as much as possible.

Reference:

1. Prine, N., Cardinal, C. & Gu, X. Understanding and controlling the depth sensitivity of scanning probe based infrared imaging and nanospectroscopy for buried polymeric structures. *Nanoscale* 15, 10244–10253 (2023).
2. Xie, Q. & Xu, X. G. What Do Different Modes of AFM-IR Mean for Measuring Soft Matter Surfaces? *Langmuir* 39, 17593–17599 (2023).
3. Mathurin, J. et al. Photothermal AFM-IR spectroscopy and imaging: Status, challenges, and trends. *Journal of Applied Physics* 131, 010901 (2022).
4. Schwartz, J. J., Jakob, D. S. & Centrone, A. A guide to nanoscale IR spectroscopy: resonance enhanced transduction in contact and tapping mode AFM-IR. *Chem. Soc. Rev.* 51, 5248–5267 (2022).

Comment 6:

The reviewer expressed concerns over the use of the term "chemical structure" in the manuscript. While the authors have made changes to the wording in many places, which is commendable, the term "chemical origins" remains in the title. In my view, this creates a misleading title, as it suggests a deeper exploration of chemical mechanisms than is actually present in the study. Going forward, I suggest that "molecular configuration" might be a better term.

Response:

We thank the reviewer for the comment and suggestion. We revised the title to "Electrically Modulated Photothermal Force Microscopy for Revealing Molecular Configuration Changes During Polarization Switching at the Nanoscale".

Revisions:

Please refer to the title in the manuscript highlighted in red

"Electrically Modulated Photothermal Force Microscopy for Revealing **Molecular Configuration Changes During** Polarization Switching at the Nanoscale"

Responses to Reviewers' comments and corresponding changes

Reviewer #4:

As we only access a summary of the reviewer's concerns, for the detailed discussion, below we will respond point by point regarding this summary. The reviewer's concerns are highlighted in light blue, while our responses and subsequent revisions are noted in black and red, respectively.

Summary:

The reviewer #4 acknowledged your great efforts to address the previous concerns, but find that additional work is necessary regarding the AFM-IR part. Regarding the PLL to correct artefacts in the IR maps, the reviewer notes that you must show the IR phase and PLL maps to check the quality of the PLL maps, where a flat IR phase map is necessary to demonstrate the working of the PLL. The reviewer further notes that to demonstrate the working of the ePFTM, you should conduct a negative control experiment on a non-ferroelectric polymer and show the cycle for specific bands of this polymer. Regarding your evaluation of the damage, the reviewer notes differences at 1183 and 1120 cm⁻¹, while only the band at 1289 cm⁻¹ is not decreasing. To check the origin of the differences, the reviewer suggests performing 10 cycles and track the differences and fluctuations during the cycles. You should also evaluate the effect of the frequency of the bias voltage on the damage and the effect of the duty ratio. In particular, the reviewer also inquires what happens when the voltage is modulated at 50 ms for 30 V. Finally, the reviewer also asks how modulation parameters (frequency, duty cycle) of the bias voltage impact the IR signal, and whether there is an optimal setting to obtain the best sensitivity and what are the limits not to damage the sample

Response:

We thank the reviewer for their recognition of our efforts in the previous response. And we also thank the reviewer for their suggestions and comments. Below, we will address the reviewer's concerns one by one. And for ease of reading, the concerns were presented again before each discussed section.

Point 1:

Regarding the PLL to correct artefacts in the IR maps, the reviewer notes that you must show the IR phase and PLL maps to check the quality of the PLL maps, where a flat IR phase map is necessary to demonstrate the working of the PLL.

To show the PLL tracking quality and reproducibility of the IR maps results in our main text, we repeat the previous experiment with the addition of phase mapping. The corresponding PLL maps and IR phase maps at 15V are shown in Fig. R1d and Fig. R1e. Other PLL frequency maps and IR phase maps are shown in Fig. R2. The results show that the variation of phase is within 10 degrees in each IR map, proving that the PLL has been properly set and is working well.

Notably, the results shown in Fig. R1 are consistent with the previous conclusion.

First, IR maps still demonstrate nonuniform IR absorption with a mottled pattern under all biases, which means coexistence of T_3GT_3G' and other configurations. Second, the variation of amplitude also experienced the same trend as Fig. 4 in the main text. As the electric bias increased, the uniformity and intensity of the 1120 cm^{-1} absorption increased, indicating the formation of T_3GT_3G' configurations. A further increase in the electric bias results in a decrease in IR absorption, aligning well with the aforementioned IR-E analysis in the main text. Therefore, the above results confirm that both ePTFM mapping (Fig. 4 in the main text and Fig. R1) reflect evolution correctly.

Fig. R1. Nanoscale mapping of electrodriven evolution via ePTFM. **a** Corresponding nanoscale ePTFM maps of 1120 cm^{-1} absorption at 5 V, 10 V, 15 V, 20 V, 25 V, and 30 V. To demonstrate the intrinsic evolution, the mapping results use the original data without flattening or further processing. The upper color box applies to the height image, whereas the lower color box applies to the other images. **b** Histograms of the 10 V, 15 V, 20 V, and 30 V ePTFM mappings. The solid line represents the fitting curve of each histogram. **c** Corresponding peak value of the ePTFM signal under each

bias. **d** The corresponding PLL frequency maps in ePTFM maps at 15 V. **e** The corresponding IR phase maps at 15 V. **f** Cross-sectional profile corresponding to the dashed white line in **d**.

Fig. R2. The corresponding PLL frequency maps and IR phase maps of electrodriven ePTFM maps. **a~f** The corresponding PLL frequency maps at 5 V, 10 V, 15 V, 20 V, 25 V, and 30 V, respectively. **g~l** The corresponding IR phase maps at 5 V, 10 V, 15 V, 20 V, 25 V, and 30 V, respectively.

Revision:

Fig. R1 and the corresponding PLL frequency maps and IR phase maps have been added as Supplementary Fig. 15 and Supplementary Fig. 16, respectively. Also, a relative description has been added in the main text.

Please refer to lines 333-336 in the manuscript highlighted in red

“A repetitive experiment has been conducted to show the reproducibility of this evolution (Supplementary Fig. 15). The results exhibit the same variation. The PLL frequency maps and flat IR phase maps demonstrated a good quality of PLL tracking (Supplementary Fig. 16).”

Point 2:

The reviewer further notes that to demonstrate the working of the ePFTM, you should conduct a negative control experiment on a non-ferroelectric polymer and show the cycle for specific bands of this polymer.

We have conducted a negative control experiment using PMMA (poly(methyl methacrylate)), a non-ferroelectric polymer (Fig. R3). The results exhibit a flat evolutionary trend, without the characteristic butterfly shape. This result corroborates the function of ePFTM. Specifically, we leveraged a specific IR absorption band at 1719 cm^{-1} , which corresponds to the C=O stretching vibration, to generate IR-E loops.

Fig. R3. The IR-E loops at 1719 cm^{-1} of the PMMA film. 1719 cm^{-1} was assigned to the C=O stretching vibration. The PMMA film was spin-coated on a silicon substrate with a Pt coating. The grey dots correspond to the distribution of 64 sampling data points of cantilever amplitude at each off-field stage. The red solid line corresponds to the fitting curve according to the average value of the sampling points. Curve fitting was conducted via local regression methods.

Revision:

The Fig. R3 has been added as Supplementary Fig. 5, and a relative description has

been added in the main text.

Please refer to lines 173-176 in the manuscript highlighted in red

“We also conducted a negative control experiment using a non-ferroelectric polymer, specifically PMMA (poly(methyl methacrylate)). The IR-E loop results reveal a flat evolutionary trend without the characteristic butterfly shape (Supplementary Fig. 5).”

Point 3:

Regarding your evaluation of the damage, the reviewer notes differences at 1183 and 1120 cm^{-1} , while only the band at 1289 cm^{-1} is not decreasing. To check the origin of the differences, the reviewer suggests performing 10 cycles and track the differences and fluctuations during the cycles.

The reviewer might have misread the direction of the x-axis in Fig. R1 of the previous response. There are no results featuring “differences at 1183 and 1120 cm^{-1} , while only the band at 1289 cm^{-1} is not decreasing” in our last response letters. The main different difference of which occurred in 1289 and 1183 cm^{-1} , while 1120 is not decreasing. Perhaps the reviewer is concerned about the differences at 1183 and 1289 cm^{-1} , while 1120 cm^{-1} is not decreasing. And the reviewer suggested checking the origin of these differences.

We have performed 10 cycles of spectrum upon the same engagement point as the reviewer suggested (Fig. R4a). The differences and fluctuations of each spectrum are nearly negligible, which is smaller than the full cycle spectrum experiment in the last response letter (please refer to its Fig. R2). Further, we track the differences and fluctuations during the cycles at 1289, 1183, and 1120 cm^{-1} (Fig. R4b~d). The fluctuations of 1289 and 1183 cm^{-1} exhibit the same tendency, which means that the difference is caused by random error. Thus, the differences in previous results (please refer to Fig. R1 in the last response letter) do not originate from the sample damage, which is consistent with the conclusion in the last response letter.

Fig. R4. Evaluation of the fluctuation of IR Spectrum in P(VDF-TrFE) **a** Ten cycles of spectrum upon the same engagement point as the reviewer suggested. **b** The fluctuations at 1289 cm^{-1} . **c** The fluctuations at 1183 cm^{-1} . **d** The fluctuations at 1120 cm^{-1} .

Point 4:

You should also evaluate the effect of the frequency of the bias voltage on the damage and the effect of the duty ratio. In particular, the reviewer also inquires what happens when the voltage is modulated at 50 ms for 30 V.

We have conducted additional experiments for evaluation. According to our observation, the damage was primarily influenced by the value and the accumulated time of voltage loading. The frequency and duty ratio determine the accumulated time and should be set according to the voltage value. We applied a triangle-squared bias sequence to the sample. As shown in Fig. R5a, under the conditions of 35 V, it can be observed that when the frequency of the drive voltage ranges from 50 mHz to 800 mHz, no significant damage is inflicted on the sample (Fig. R5b). As a contrast, when the voltage exceeds 40 V, the sample undergoes severe deformation (Fig. R5c-d). In addition to the effect of duty ratio, we apply a bias for different durations to evaluate this effect. As shown in Fig. R5e, a 30V voltage was applied to the sample for durations ranging from 0.1s to 20s, in which damage only occurred when accumulated time is too long (Fig. R5f). Notably, multiple IR spectrum method usually takes several tens of seconds for data acquisition, thus is more likely to damage the sample.

In this regard, when the voltage is modulated at 50 ms for 30 V during the IR-E measurements, the sample will not be damaged, and its polarization can be stably established.

Fig. R5. Evaluation of the sample damage. **a** Topography before the application of the triangle-squared bias sequence. The maximum voltage of the sequence is 35 V. The white circle denoted the position where the bias was applied. **b** Topography after the application of the triangle-squared bias sequence. **c** Topography before the application of the triangle-squared bias sequence. The maximum voltage of the sequence is 40 V. **d** Topography after the application of the triangle-squared bias sequence. **e** Topography before the application of DC bias. The white circle denoted the position where the bias was applied. **f** Topography before the application of DC bias.

Point 5:

Finally, the reviewer also asks how modulation parameters (frequency, duty cycle) of the bias voltage impact the IR signal, and whether there is an optimal setting to obtain the best sensitivity and what are the limits not to damage the sample

On one hand, IR signal acquisition takes some time because the photothermal vibration is usually quite small. Thus, the frequency cannot be too high, and the duty cycle cannot be too short. On the other hand, low frequency and long duty cycle may increase the loading time of bias, resulting in sample damage and disrupting the IR signal, especially when the bias is too high.

The optimal setting for better sensitivity and limits for no damage is dependent on various factors. The amplitude of the voltage is the first thing to be concerned. According to our experience, to start setting optimization, one can slowly increase the voltage and find a voltage that just begins to damage the sample, then slightly reduce the voltage and optimize the IR parameters. One should ensure that the frequency and duty cycle are sufficient for IR signal acquisition.

Responses to Reviewers' comments and corresponding changes

Reviewer #5:

Summary:

As I understand, I am asked to assess authors' Response to the original reviewer as well as a followup referee. I have read the Response carefully, and it appears to me that the authors Response is adequate. One of the key issues is whether the technique damages the sample, and the authors articulated that the technique they proposed, cycling under a fixed wavenumber, does not damage the sample, while the suggested checkup, cycling under the full spectrum, could damage the sample. The response is quite convincing. Other issues raised are pretty minor to me, and the authors response is satisfactory. The technique proposed is quite promising to me. As the authors explained in their motivation, STEM does damage the organic, or organic-inorganic sample, even under low doses; see for example Nature Communications volume 9, Article number: 4807 (2018); and the sample preparation for STEM could alter polar topology as well; see for example Nature Communications volume 12, Article number: 4620 (2021). The technique proposed thus provide a valuable alternative to probe both polar and chemical structures at nanoscale.

Response:

We sincerely thank the reviewer for taking the time and effort to review our work. The STEM examples are very useful for us to demonstrate our point. It has been cited as ref. 23, 25.

Revision:

Please refer to lines 68-70 in the manuscript highlighted in red

“For other approaches, although transmission electron microscopy (TEM) may offer an alternative method for nanoscale chemical and polar analyses, radiation damage to organics can be a concern^{23,24}. The preparation procedure for thin-film samples, which involves slicing and ion thinning, can potentially alter the original structure of the organic film²⁵, even when using cryo-TEM.”

Responses to Reviewers' comments and corresponding changes

Reviewer #6:

I believe this manuscript represents a significant advance and is appropriate in its current form for publication in Nature Communications. The authors demonstrate the ability to dynamically modulate the polarization of ferroelectric materials while measuring corresponding changes in photothermally detected IR absorption with nanoscale spatial resolution.

In particular, the ability to perform chemically specific measurements of ferroelectric hysteresis loops at multiple IR bands is very compelling. This enables the authors to track specific molecular vibrational modes during polarization switching, offering insight into the molecular mechanisms behind ferroelectric behavior—including conformational transitions that are otherwise difficult to resolve.

The latest revision of the manuscript has satisfactorily addressed previous reviewer concerns. In particular, the negative control experiment using non-ferroelectric PMMA is well designed and convincingly rules out spurious contributions to the IR–E signals.

I expect the publication of this manuscript will attract broad interest, particularly among researchers in ferroelectric materials, nanoscale spectroscopy, and organic electronics. I strongly recommend publication, either as-is or with attention to the following minor suggestions for clarity:

Response:

We sincerely thank the reviewer for their thorough evaluation and recommendation of our work. We are grateful for the time and effort invested in reviewing our manuscript, as well as for the constructive feedback. Below we will respond point by point regarding reviewer suggestions. The reviewer's suggestions are highlighted in light blue, while our responses and subsequent revisions are noted in black and red, respectively.

Minor Suggestions for Improvement:

1. Figure 1a (Lock-in Amplifier Detail):

The diagram includes internal lock-in amplifier components such as mixers, low-pass filters, and X/Y-to-R conversion, which are standard and well understood. These details may distract from the core technical innovation being illustrated. I suggest simplifying this part of the figure to a single “Lock-in Amplifier” block, allowing the reader to focus on the ePTFM configuration and its conceptual novelty.

Response:

We thank the reviewer's helpful suggestion. This will be helpful in enhancing the clarity of our article.

Revisions:

As the reviewer suggested, we have simplified the diagram of the lock-in amplifier and rearranged the circuit block. Please refer to Figure R1 and Figure 1a in the main text. Meanwhile, the relative description in the main text has been revised. Please refer to lines 103-105 in the manuscript.

“The circuit first divides the deflection of the probe into two equivalent parts and multiplies them with their reference signals, which are produced by the OSC with a 0° or 90° phase shift. Following lock-in amplifier, the results yield the amplitude R and phase θ .”

2. Figure 1c (Molecular Representation Clarity):

a. The yellow glow around one molecule is unexplained and somewhat obscures the “+” charge marker. Consider either removing the glow or clarifying its intended meaning (if any).

Response:

The reviewer’s comment and reminder are appreciated. The yellow glow denoted the molecular structure that is selectively observed through infrared absorption. The meaning of the yellow glow was added in the caption of Figure 1.

Revisions:

Please refer to the caption of Figure 1 highlighted in red.

“The yellow glow indicates the molecule that is selectively observed.”

b. The charge indicators (+/–) are subtle and could be made more prominent.

Response:

We thank the reviewer for their suggestion. The charge indicators layer was brought to the front of the image. Meanwhile, the charge indicators have been scaled up for enhanced visibility.

Revision:

The charge indicators have been modified to enhance visibility. Please refer to Figure R1 and Figure 1c.

c. Adding a small vector arrow to indicate the dipole orientation of the selected molecule may help communicate the principle more intuitively.

Response:

Thanks for the reviewer’s useful comment. A small vector arrow was added to indicate the dipole orientation of selected molecule.

Revision:

Please refer to Figure R1 and Figure 1c in the manuscript.

d. The “IR electric field” label and arrow currently appear only on the right-hand panel. Initially, this caused some confusion—was IR illumination present only in that frame? To clarify, I suggest either centering the label between the two panels or moving it to

Figure 1a, where the system schematic could establish the IR field's consistent direction across all panels.

Response:

Thanks for the reviewer's suggestion. The "IR electric field" label and arrow in Figure 1c are strategically positioned to provide interpretive guidance for readers. Thus, after careful consideration, we maintain them in Figure 1c. To avoid the confusion that the reviewer was concerned about, the position of "IR electric field" label and arrow was adjusted in Figure 1c. Additionally, an "IR electric field" arrow was added in the schematic diagram of the laser beam in Figure 1a. Meanwhile, the "polarized pulsed laser" label was replaced by "pulsed polarized IR laser".

Revision:

Please refer to Figure R1 and Figure 1a in the main text.

Fig. R1. Setup and working principle of electrically modulated photothermal force microscopy. **a** Schematic of electrically modulated photothermal force microscopy. A polarized pulsed laser illuminates the sample below the AFM tip, generating photothermal expansion. A modulated bias is applied through the AFM tip for manipulation of the polarization. The cantilever deflection signal is obtained under the influence of pulse infrared excitation, photothermal force and electric modulation. A tuning circuit is used to collect and analyse the cantilever deflection signal. **b** Time-dependent variation in the amplitude and electric bias. A more specific result is shown in Supplementary Fig. 1. IR absorption is recorded under a triangle–square bias. The amplitude trace (red), which indicates the IR absorption, varies with the modulated bias (blue), reflecting the corresponding electrodriven evolution. **c** Schematic showing the correlation between the IR absorption and rotation of the molecular structure under a vertically polarized IR beam. The red and blue regions represent the polarizations of the entire species. The yellow glow indicates the molecule that is selectively observed.

The specific molecular dipolar moment can be tracked by selecting the IR wavenumber. The amplitude reaches the maximum or minimum value if the selective transition dipolar moment is parallel or perpendicular to the IR electric field, respectively.

3. Terminology: "Recently Developed" AFM-IR

The manuscript describes AFM-IR as "recently developed" but the first seminal paper on AFM-IR was published 20 years ago (citation below) and AFM-IR has been commercialized for at least 15 years. At this point AFM-IR is more accurately described as an established and widely used technique. I suggest modifying the wording accordingly.

A. Dazzi, R. Prazeres, F. Glotin, and J. M. Ortega, "Local infrared microspectroscopy with subwavelength spatial resolution with an atomic force microscope tip used as a photothermal sensor," *Opt. Lett.* 30, 2388-2390 (2005)

Response:

We thank the reviewer for pointing out the mistake. Relative terminology has been revised.

Revision:

Please refer to lines 70-72 in the manuscript highlighted in red.

“The **widely used** atomic force microscopy-infrared spectroscopy (AFM-IR) has **established itself** as a powerful tool for nanoscale chemical imaging, utilizing a combination of mechanical and optical techniques²⁶⁻²⁸”